# AUTOMATED STRUCTURED RADIOLOGY REPORT GENERATION WITH RICH CLINICAL CONTEXT

## ABSTRACT

Automated *structured radiology report generation* (SRRG) from chest X-ray images offers significant potential to reduce workload of radiologists by generating reports in structured formats that ensure clarity, consistency, and adherence to clinical reporting standards. While radiologists effectively utilize available clinical contexts in their diagnostic reasoning, existing SRRG systems *overlook* these essential elements. This fundamental gap leads to critical problems including *temporal hallucinations* when referencing non-existent clinical contexts. To address these limitations, we propose *contextualized SRRG* (**C-SRRG**) that comprehensively incorporates rich clinical context for SRRG. We curate C-SRRG dataset by integrating comprehensive clinical context encompassing 1) multi-view X-ray images, 2) clinical indication, 3) imaging techniques, and 4) prior studies with corresponding comparisons based on patient histories. Through extensive benchmarking with diverse multimodal large language models, we demonstrate that incorporating clinical context with the proposed C-SRRG significantly improves report generation quality, as summarized in Fig. 1. We will publicly release dataset, code, and checkpoints to facilitate future research for clinically-aligned automated RRG.

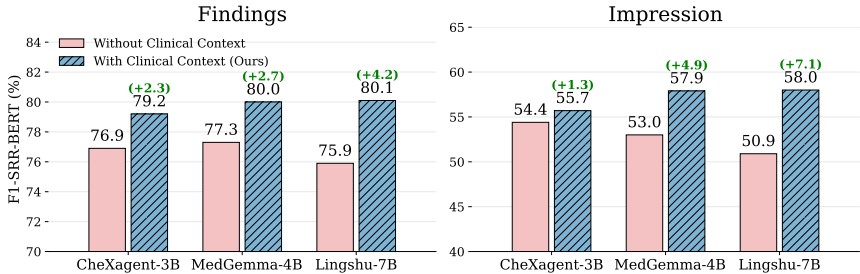

Figure 1: **Clinical context consistently and significantly improves medical MLLMs**—including CheXagent-3B (Chen et al., 2024b), MedGemma-4B (Sellergren et al., 2025), and Lingshu-7B (Team et al., 2025)—on both the findings and impression tasks for SRRG, as measured by F1-SRR-BERT metric (Delbrouck et al., 2025).

## 1 INTRODUCTION

Writing a radiology report requires radiologists to accurately interpret images and synthesize them into two main components: 1) detailed *findings* that systematically document anatomical structures and pathological observations, and 2) concise *impressions* that provide clinical interpretations for subsequent decision-making (Wallis & McCoubrie, 2011; Pahadia et al., 2020; Haygood et al., 2018; Trinh et al., 2019; ESR, 2011). However, generating such comprehensive reports is both cognitively demanding and time-consuming for radiologists. Given the high volume of imaging studies and the time-intensive nature of report writing, there is a critical need for automated systems that can assist radiologists by generating accurate, structured reports while reducing radiologists' workload and improving diagnostic efficiency (Markotić et al., 2021; Alexander et al., 2022).

*Automated radiology report generation* (**RRG**) has emerged as a crucial task to address these challenges by assisting radiologists in the diagnostic workflow (Esteva et al., 2019; Sloan et al., 2024; Tanno et al., 2025). Deep learning has accelerated the development of automated RRG frameworks that generate reports directly from medical images (Shin et al., 2016; Jing et al., 2018; Li et al.,

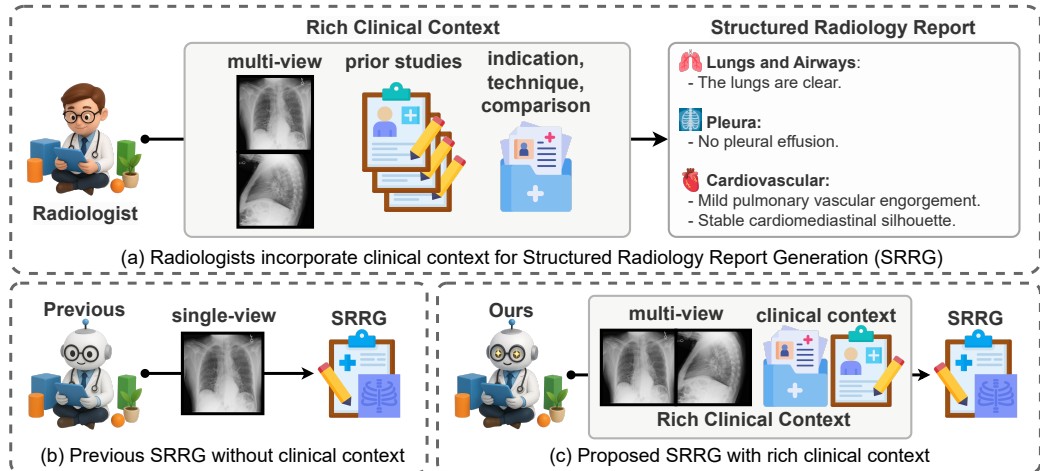

Figure 2: **A conceptual illustration of the proposed C-SRRG**. **(a)** Radiologists routinely use clinical context, while **(b)** existing SRRG frameworks do not. Motivated by this gap, **(c)** C-SRRG leverages multi-view images, indication, technique, and variable-length prior studies/comparisons to generate structured radiology reports.

2018; Wang et al., 2018; Jing et al., 2020; Chen et al., 2020b; 2022; Wang et al., 2022a). Recent advances in multimodal large language models (MLLMs) further enhanced this capability by integrating vision foundation models with large language models capable of generating coherent and clinically relevant text (Lee et al., 2025; Li et al., 2023a; Hyland et al., 2023; Bannur et al., 2024; Chen et al., 2024b; Sellergren et al., 2025; Team et al., 2025; Wang et al., 2025).

Despite the progress, most automated RRG frameworks overlook *essential clinical context* such as imaging indication, technique, and prior studies that radiologists use to generate reports (Kahn et al., 2009; ESR, 2011). Ignoring the clinical context leads to systematic errors (Liu et al., 2019; Ramesh et al., 2022) as the models fail to capture patient-specific properties and longitudinal changes essential for accurate diagnosis, including *temporal hallucinations* where the models generate references to nonexistent priors or fabricate temporal comparisons (Figs. 22 to 24). Although some work injected partial context—*e.g.*, multi-view images (Yuan et al., 2019; Chen et al., 2022), historical images (Hou et al., 2023a; Zhu et al., 2023b), and prior reports or indications (Miao et al., 2024; Wang et al., 2024)—these are still *limited*, *e.g.*, only consider partial clinical context, rely only on the immediately preceding image or report (Bannur et al., 2024; Liu et al., 2025b;a), with all approaches targeting free-form RRG rather than *structured* RRG (SRRG) (Delbrouck et al., 2025).

To this end, we first present **C-SRRG**, a framework for *contextualized structured radiology report generation* (Fig. 2), built upon the recently introduced SRRG paradigm (Delbrouck et al., 2025). We curate a *large-scale* dataset for *structured* report generation with *rich clinical context*, by leveraging MIMIC-CXR (Johnson et al., 2019) and CheXpert Plus (Chambon et al., 2024). Specifically, our **C-SRRG dataset** provides 1) **multi-view images** (frontal and lateral), 2) clinical **indication**, 3) imaging **technique**, and 4) variable-length **prior studies** with corresponding comparisons, which models can incorporate depending on their architecture.

We evaluate the effectiveness of the proposed C-SRRG with diverse medical MLLMs—including CheXagent-3B (Chen et al., 2024b), MedGemma-4B (Sellergren et al., 2025), and Lingshu-7B (Team et al., 2025)—and find that incorporating clinical context **substantially and consistently improves report quality** (summarized in Fig. 1 and detailed in Tabs. 3 and 4) measured by various metrics (Papineni et al., 2002; Lin, 2004; Zhang et al., 2019; Delbrouck et al., 2022; 2025). grows from 3B to 7B. We also provide a comprehensive analysis, including extensive ablation studies on clinical context (Tabs. 5 to 8), temporal hallucination mitigation (Tab. 9), and organ-level performance (Tab. 10). We will *publicly release* the 1) **dataset**, 2) **code**, and 3) **checkpoints** of benchmarked models to facilitate further research in C-SRRG and benefit the community.

Our contributions and empirical findings are summarized as follows:

- We identify a key limitation of existing SRRG frameworks, *i.e.*, the neglect of essential *clinical context*, which induces systematic errors, most notably *temporal hallucinations* about nonexistent prior studies. To address this, we introduce a clinically contextualized SRRG framework (**C-SRRG**) that explicitly integrates clinical context into the generation process.

- We curate *the largest structured* radiology report generation dataset with *rich clinical context*, namely, **C-SRRG dataset**, which includes 1) multi-view images, 2) indication, 3) technique, and 4) prior studies/comparisons, for training and evaluation of the proposed C-SRRG framework.

- As summarized in Fig. 1, we provide a *comprehensive benchmark* of **diverse MLLM**-based SRRG models, demonstrating that clinical context becomes **increasingly critical as models scale up**—enhancing report quality (*e.g.*, +2.3∼4.2/+1.3∼7.1 on findings/impression for F1-SRR-BERT) while reducing *temporal hallucinations* (Tab. 9; *e.g.*, 12.2%/18.0% on findings/impression).

## 2 RELATED WORK

**Automated radiology report generation (RRG).** Automated RRG has emerged as a promising approach to reduce radiologists' workload and improve reporting efficiency (Yang et al., 2023; Sloan et al., 2024; Esteva et al., 2019; Sirshar et al., 2022; Tanno et al., 2025; Singh & Singh, 2025). While early approaches simply combined vision encoders with language decoders for visual feature extraction (He et al., 2015; Dosovitskiy et al., 2020) and text generation (Shin et al., 2016; Jing et al., 2018; Li et al., 2018; Wang et al., 2018; Jing et al., 2020; Chen et al., 2020a; Yan & Pei, 2022; Miura et al., 2020), architectural innovations have significantly improved report quality, such as memory-driven transformers (Chen et al., 2020b; Liu et al., 2024b), specialized architectures for medical domain knowledge (Yang et al., 2021; Wang et al., 2022b; Kong et al., 2022), cross-modal learning for improved alignment (Chen et al., 2022; Wang et al., 2022a; Li et al., 2023b), and region-guided frameworks for anatomically relevant features (Tanida et al., 2023; Li et al., 2023c; Hou et al., 2023b). In this work, we focus on extending the recently proposed structured RRG (SRRG; Delbrouck et al., 2025)—improving clarity, consistency, and interpretability through standardized structure (Weiss & Langlotz, 2008; Kahn et al., 2009; Bosmans et al., 2012; 2015)—by incorporating rich clinical context aligned with radiologists' workflow.

**Multimodal large language models (MLLMs).** Building on recent advances in LLMs (Bai et al., 2023; Achiam et al., 2023; Touvron et al., 2023a;b; Yang et al., 2025), MLLMs have shown strong performance across many domains, including medical applications (Achiam et al., 2023; Team et al., 2023; Yang et al., 2025; Comanici et al., 2025; Wang et al., 2025; Zhu et al., 2025). They integrate visual understanding with natural-language generation, enabling effective tools for medical image analysis and clinical text generation (Li et al., 2023a; Chen et al., 2024a; He et al., 2024; Hurst et al., 2024; Lai et al., 2025; Pan et al., 2025). Medical-specific MLLMs further improve performance by incorporating domain knowledge and clinical expertise through specialized training procedures, including CheXagent (Chen et al., 2024b), MedGemma (Sellergren et al., 2025), and Lingshu (Team et al., 2025). These foundation models are particularly promising for comprehensive RRG frameworks (Lee et al., 2023; Zhu et al., 2023a; Liu et al., 2024c; Wang et al., 2023; Hyland et al., 2023; Bannur et al., 2024; Lee et al., 2025; Chen et al., 2024b; Dai et al., 2025; Sellergren et al., 2025; Team et al., 2025), where their ability to accept flexible inputs and produce coherent clinical text is especially valuable. Accordingly, we benchmark medical MLLMs for contextualized SRRG.

**Clinical context.** Radiologists routinely leverage clinical context when drafting reports, drawing upon patient history, prior studies, and clinical indications (Kahn et al., 2009; ESR, 2011; Wallis & McCoubrie, 2011; Haygood et al., 2018; Trinh et al., 2019; Pahadia et al., 2020; Castillo et al., 2020; Nguyen et al., 2021), motivating various approaches to integrate such clinical context into automated RRG frameworks. Multi-view image analysis utilizes complementary imaging perspectives, such as frontal and lateral views, to provide comprehensive anatomical coverage (Yuan et al., 2019; Miao et al., 2024; Chen et al., 2022; Nooralahzadeh et al., 2021; Serra et al., 2023; Liu et al., 2024d; Nicolson et al., 2024). Indication and clinical history integration approaches incorporate patient-specific clinical information (Hou et al., 2023a; Zhu et al., 2023b; Wang et al., 2024; Liu et al., 2024a; Miao et al., 2024). Previous studies enable temporal comparison and disease progression tracking (Hou et al., 2023a; Zhu et al., 2023b; Serra et al., 2023; Wang et al., 2024; Liu et al., 2021). Recent works such as MLRG (Liu et al., 2025b), PriorRG (Liu et al., 2025a), and MAIRA-2 (Bannur et al., 2024) have attempted to incorporate clinical context for more comprehensive report generation. However, these approaches have limitations: they either 1) consider only partial clinical context, 2) are restricted to specific input configurations, 3) have narrow temporal scope (only the previous prior study), with all 4) focusing on unstructured free-form report generation. We add detailed distinction between prior works and ours in terms of clinical context integration in §B.

> **Structured Report (Excerpt):**
> History: A male patient with hep C cirrhosis and large right pleural effusion status post thoracocentesis.
>
> Comparison: Prior portable AP chest radiograph
>
> Findings:
> Pleura:
> - Moderate pleural effusion within the right pleural space.
> - Moderate right pneumothorax, **new from prior exam**.
> - No left pleural effusion or pneumothorax.
> Impression:
> 1. Moderate right-sided pneumothorax.
> 2. Moderate right pleural effusion.
>
> > **Hallucination:** The phrase "**new from prior exam**" represents temporal information that cannot be verified from the current study alone, if not with previous history.

Figure 3: **An example of temporal hallucinations**. This report contains "**new from prior exam**" even though any prior studies are not provided. Please see examples of full structured reports in Figs. 22 to 24.

## 3 METHOD

In this section, we first elaborate on the dataset curation process for contextualized radiology report generation (C-SRRG) in §3.1 and then detail the proposed C-SRRG framework in §3.2.

### 3.1 CURATION OF CONTEXTUALIZED CLINICAL CONTEXT

**Motivation.** Our design principle is to reflect the clinical workflow of radiologists that incorporates a diverse diagnostic context such as indication, technique, and comparison (Wallis & McCoubrie, 2011; Trinh et al., 2019; Pahadia et al., 2020; Nguyen et al., 2021), supported by empirical evidence showing improvement in report quality (Castillo et al., 2020; Liu et al., 2021). This emphasis on a comprehensive clinical context aligns with recent work advocating that AI systems must move beyond narrow task-specific approaches that lack the ability to incorporate multimodal data and provide comprehensive interpretation assistance (Dogra et al., 2025). Most importantly, without this context, existing automated systems are prone to *temporal hallucinations*: ground truth reports frequently contain temporal statements such as "**new from prior exam**" (as shown in Fig. 3), which leads models to hallucinate by referencing nonexistent prior examination (Ramesh et al., 2022). When trained on such data, the SRRG frameworks learn to generate these temporal phrases even when no prior studies are available, as demonstrated in Figs. 25 to 30.

**Clinical context.** To address this limitation, we incorporate rich clinical context into automatic SRRG frameworks. Specifically, we consider **four clinical elements** that radiologists routinely use (refer to §D for detailed clinical interpretation of each clinical context element):

1. **Multi-view images** (*e.g.*, posteroanterior, anteroposterior, and lateral) provide complementary perspectives from different angles, enabling comprehensive assessment and detection of abnormalities that may be obscured in single views. Multi-view fusion captures richer information through cross-view consistency, improves pathological localization accuracy, and reduces diagnostic uncertainty (Yuan et al., 2019; Miao et al., 2024).

2. **Indication** conveys the clinical rationale for imaging, providing context about patient symptoms, suspected conditions, or clinical questions. This enables models to focus on specific diagnostic questions, tailor findings to physician concerns, and avoid clinically insignificant findings.

3. **Technique** documents examination parameters and limitations including imaging protocols, contrast use, and factors affecting image quality. It helps models note technical caveats, avoid mistaking artifacts for pathology, and prevent duplicate exams.

4. **Prior studies**, when available, enable temporal **comparison** by providing a history to detect disease progression, treatment response, and interval changes. Radiologists routinely consult such a history (Haygood et al., 2018; Liu et al., 2025a), which supports accurate change detection and prevents hallucinations referencing nonexistent prior exams.

Table 1: **Dataset statistics** for C-SRRG-Findings and C-SRRG-Impression.

| Tasks | Train | Valid | Test | Test-reviewed | Total |
|---|---|---|---|---|---|
| Findings | 181,874 | 976 | 1,459 | 233 | 184,542 |
| Impression | 405,972 | 1,505 | 2,219 | 231 | 409,927 |

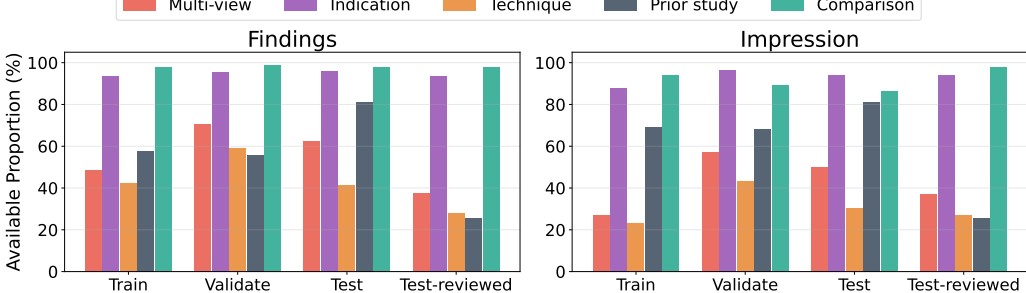

Figure 4: **Available proportion of clinical context** for each split in findings and impression.

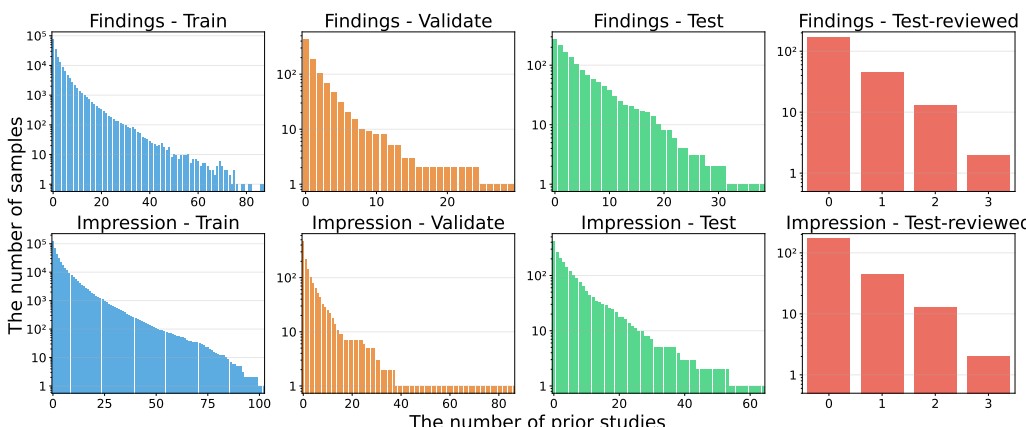

Figure 5: **Distribution of the number of prior studies** available per sample for findings and impression.

**Dataset curation.** We build on the recently proposed SRRG dataset (Delbrouck et al., 2025), which includes both MIMIC (Johnson et al., 2019) and CheXpert Plus (Chambon et al., 2024). We employ dataset-specific approaches to extract the necessary clinical context. When multiple views are available, we integrate multi-view images using `ViewPosition` for MIMIC and `frontal_lateral`, `ap_pa` for CheXpert Plus. For MIMIC, each patient is identified by a unique `subject_id` with associated `StudyDate` and `StudyTime` fields. We group patients by `subject_id`, then use temporally-ordered `StudyDate` and `StudyTime` to establish chronological sequences. For CheXpert Plus, each study contains a `deid_patient_id` and `patient_report_date_order` field. We group studies by patient and use the order of report dates to form longitudinal sequences. For other clinical contexts (indication, technique, comparison), we use SRRG components, parsed from free-form reports using GPT-4 (Achiam et al., 2023).

**Dataset analysis.** Accordingly, we curate two C-SRRG tasks that mirror clinical practice—**C-SRRG-Findings** and **C-SRRG-Impression** with **train**, **valid**, **test**, and **test-reviewed**[1] splits, as summarized in Tab. 1. The splits enforce the strict separation of patients between training and evaluation to prevent data leakage and properly assess generalization, as in Fig. 10. The availability of a clinical context varies across splits and tasks (Fig. 4). The availability of prior studies follows a

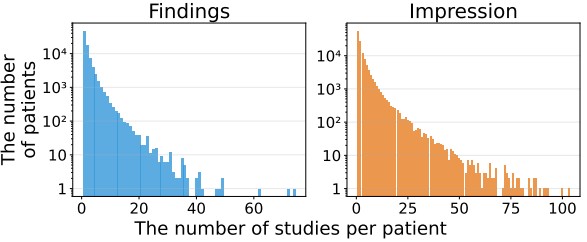

Figure 6: **Distribution of the number of studies per patient**.

---

[1]The *test-reviewed* split reports are reviewed by board-certified radiologists (Delbrouck et al., 2025).

**SYSTEM PROMPT:**
You are an expert radiologist.

**USER PROMPT:**
Analyze the current chest X-ray images and compare them with the previous studies to write the IM-PRESSION section of a radiology report. Provide a concise clinical summary and diagnosis, noting any changes from the prior studies, focusing on the most recent comparisons. Consider the available clinical contexts when formulating your impression.

Figure 7: **Prompt format for IMPRESSION** for C-SRRG.

**=== CURRENT CLINICAL HISTORY/INDICATION ===**
Male with end-stage renal disease on hemodialysis, multiple orthopedic hardware, bioprosthetic aortic valve replacement, coronary artery disease status post coronary artery bypass grafting, permanent pace-maker for sick sinus syndrome, admitted for MRSA bacteremia, transferred to the critical care unit for hypotension during anesthesia induction. Patient intubated for respiratory status assessment.

**=== CURRENT TECHNIQUE ===**
Chest single view

**=== CURRENT COMPARISON ===**
Prior imaging at an unspecified time.

**=== CURRENT IMAGES ===**
```
[Images:  Current chest X-ray images]
```

Figure 8: **An example of current study** for C-SRRG.

**ASSISTANT RESPONSE:**
IMPRESSION:
1. Worsened left basilar opacity, which may represent atelectasis or pneumonitis in the appropriate clinical context.
2. Mild improvement in pulmonary vascularity.

Figure 9: **An example of ground-truth assistant responses** in the C-SRRG-Impression dataset.

long-tailed distribution (Fig. 5), alongside the long-tailed counts of studies per patient (Fig. 6)—from no history to extensive longitudinal sequences. This variability reflects *real-world clinical practice* and requires models to handle missing information while leveraging available context.

## 3.2 Contextualized Radiology Report Generation (C-SRRG)

**Prompt design.** We construct prompt templates for four core settings: 1) findings prompts with prior studies (Fig. 11), 2) findings prompts without prior studies (Fig. 12), 3) impression prompts with prior studies (Fig. 7), and 4) impression prompts without prior studies (Fig. 13). Each prompt consists of the clinical context (*i.e.*, indication, technique, and comparison for the current study), and the associated images (Fig. 8). When available, it incorporates prior studies that also include indication, technique, comparison, and reports on findings or impression (Fig. 14). These structured components are concatenated to form a *single multimodal token sequence*. As shown in Figs. 9 and 15, the response format is standardized for the generation of structured reports. Detailed examples of prompt structures and integration of clinical context can be found in §G and §H.

**Training and inference.** We fine-tune medical MLLMs on these contextualized prompt–response pairs for both findings and impression tasks. Models receive prompt and clinical context to form a unified multimodal input sequence. The training objective is then the next-token prediction task under an autoregressive language modeling loss: $\frac{1}{T} \sum_{t=1}^{T} -\log p_\theta(y_t|x, y_{<t})$, where $x$ is the multimodal token sequence comprising the prompt (Figs. 7 and 11 to 13), the clinical context of the current study (Fig. 8) and any prior studies (Fig. 14), and $y_{1:T}$ is the target token sequence (*e.g.*, reports on findings or impression; Figs. 9 and 15). Here, $p_\theta$ denotes the MLLM parameterized by $\theta$. We minimize the negative log-likelihood with respect to $\theta$, *i.e.*, standard cross-entropy over the vocabulary. If prior studies are available, they are inserted into designated slots; otherwise, the model receives only the clinical context of the current study. This design allows the model to adapt to *heterogeneous clinical contexts* (Figs. 4 to 6), to produce context-aware reports when there is prior information, and to avoid hallucinated temporal comparisons when not.

## 4 EXPERIMENTS

### 4.1 EXPERIMENTAL SETUPS

**Implementation details.** We evaluate **CheXagent-3B** (Chen et al., 2024b), **MedGemma-4B** (Sellergren et al., 2025), and **Lingshu-7B** (Team et al., 2025). We first train baseline models without clinical context, generating reports directly from single image. When training with C-SRRG, we use all available clinical context (*e.g.*, indication, prior studies). The only exception is CheXagent-3B on the C-SRRG-Impression, where we use only *indication* due to training failure (detailed in §C.2). We consider the two most recent prior studies with limited number of images (2/3/2 for CheXagent-3B/MedGemma-4B/Lingshu-7B) due to computational constraints. We apply LoRA (Hu et al., 2022) for fine-tuning, optimizing with Adam (Kingma & Ba, 2014), and use vLLM (Kwon et al., 2023) for inference. Greedy decoding is adopted for *reproducibility* consistent with benchmarking purpose. All experiments run on a single NVIDIA H100 GPU. Detailed hyperparameter settings are in Tab. 2.

Table 2: **Summary of hyperparameters**.

| Name | Value |
|---|---|
| *LoRA* | |
| Rank $r$ | 32 |
| $\alpha$ | 64 |
| Dropout $p$ | 0.1 |
| *Training* | |
| Batch size | 128 |
| Optimizer | Adam |
| Epochs | 1 |
| Learning rate | 2e-4 |
| LR scheduler | Cosine |
| Warmup ratio | 3% |
| *Inference* | |
| Package | vLLM |
| Strategy | Greedy |

**Evaluation metrics.** We use standard metrics, such as **BLEU** (Papineni et al., 2002), **ROUGE-L** (Lin, 2004), and **BERTScore** (Zhang et al., 2019), to assess text quality. For clinical accuracy, we report **F1-RadGraph** (Delbrouck et al., 2022) and SRRG-specific metrics (Delbrouck et al., 2025): **F1-SRRG-BERT**, built on CXR-BERT (Boecking et al., 2022) for structured evaluation, and **Category Score** (only for findings) for the correctness of organ-section headers.

### 4.2 EXPERIMENTAL RESULTS

Table 3: **Results on the C-SRRG-Findings**. Clinical context is incorporated with our C-SRRG framework.

| Model | Clinical Context | Split | Traditional Metrics | | | | F1-SRR-BERT | | | Category Score | | |
|---|---|---|---|---|---|---|---|---|---|---|---|---|
| | | | BLEU | ROUGE-L | BERT Score | F1-RadGraph | Precision | Recall | F1-Score | Precision | Recall | F1-Score |
| CheXagent-3B | ✗ | Valid | 1.97 | 20.63 | 30.33 | 13.07 | 44.67 | 45.16 | 43.46 | 73.61 | 81.17 | 75.54 |
| | | Test | 2.08 | 20.09 | 31.91 | 12.99 | 43.73 | 42.54 | 41.70 | 74.47 | 85.26 | 77.74 |
| | | Test-reviewed | 2.13 | 20.38 | 32.73 | 12.96 | 44.94 | 42.78 | 42.31 | 72.84 | 87.35 | 77.55 |
| | ✓ | Valid | 2.31 | 23.01 | 33.46 | 15.76 | 48.73 | 48.20 | 46.79 | 77.58 | 83.46 | 78.73 |
| | | Test | 1.89 | 20.92 | 33.28 | 13.58 | 45.18 | 44.10 | 43.07 | 75.79 | 85.69 | 78.82 |
| | | Test-reviewed | 1.98 | 21.64 | 34.32 | 14.05 | 47.50 | 45.09 | 44.59 | 76.08 | 88.87 | 79.93 |
| MedGemma-4B | ✗ | Valid | 1.51 | 20.95 | 30.83 | 13.98 | 42.93 | 45.50 | 42.12 | 78.48 | 78.00 | 76.26 |
| | | Test | 1.58 | 19.69 | 31.52 | 13.30 | 42.32 | 41.38 | 40.19 | 76.31 | 82.36 | 77.44 |
| | | Test-reviewed | 1.60 | 20.11 | 32.61 | 13.42 | 44.49 | 42.94 | 41.92 | 75.39 | 86.56 | 78.24 |
| | ✓ | Valid | 4.98 | 27.22 | 37.87 | 20.44 | 50.52 | 49.68 | 48.42 | 80.38 | 83.73 | 80.35 |
| | | Test | 3.05 | 23.17 | 35.65 | 15.91 | 45.84 | 44.24 | 43.43 | 78.28 | 84.67 | 79.59 |
| | | Test-reviewed | 4.29 | 24.37 | 36.60 | 17.01 | 47.90 | 45.17 | 44.96 | 76.73 | 87.84 | 80.04 |
| Lingshu-7B | ✗ | Valid | 1.42 | 17.68 | 27.20 | 10.56 | 40.15 | 41.45 | 39.29 | 74.37 | 75.97 | 73.57 |
| | | Test | 1.40 | 17.71 | 29.65 | 11.14 | 40.60 | 39.41 | 38.65 | 75.86 | 81.47 | 76.84 |
| | | Test-reviewed | 1.60 | 18.62 | 31.09 | 12.09 | 42.85 | 40.82 | 40.37 | 74.32 | 85.20 | 77.39 |
| | ✓ | Valid | 6.02 | 28.70 | 38.85 | 21.67 | 51.16 | 50.50 | 49.20 | 81.97 | 83.03 | 80.87 |
| | | Test | 3.16 | 23.53 | 35.60 | 16.07 | 45.96 | 44.42 | 43.63 | 79.80 | 83.20 | 79.68 |
| | | Test-reviewed | 4.42 | 23.70 | 35.76 | 16.09 | 47.48 | 44.80 | 44.54 | 77.57 | 86.71 | 79.83 |

**Results on the C-SRRG-Findings.** Tab. 3 demonstrates **substantial improvements** achieved by C-SRRG on the C-SRRG-Findings across all evaluation metrics, except for slight BLEU decreases for CheXagent-3B on the test/test-reviewed splits (-0.19/-0.15). For example, F1-SRR-BERT scores improve by **+3.33/+1.37/+2.28** (CheXagent-3B), **+6.30/+3.24/+3.04** (MedGemma-4B), and **+9.91/+4.98/+4.17** (Lingshu-7B), with **larger models consistently showing greater gains**. Category Score performance likewise improves by **+3.19/+0.99/+2.38** (CheXagent-3B), **+4.09/+2.15/+1.80** (MedGemma-4B), and **+7.30/+2.84/+2.44** (Lingshu-7B).

**Results on the C-SRRG-Impression.** Tab. 4 also shows **significant gains** achieved by our C-SRRG on the C-SRRG-Impression. F1-SRR-BERT improves by **+0.8/+3.12** (CheXagent-3B), **+5.5/+4.43/+4.69** (MedGemma-4B), and **+7.42/+7.68/+6.16** (Lingshu-7B) except for CheXagent-3B on the valid split (-0.06). CheXagent-3B also exhibits similar BLEU score decreases on the C-SRRG-Findings (Tab. 3), indicating that rich clinical context may compromise text generation fluency in smaller models. We observe that performance **increases as the number of parameters grows from 3B to 7B**; however, this does not imply that clinical context is always helpful, as there may be **confounding factors**.

Table 4: **Results on the C-SRRG-Impression**. Clinical context is incorporated with our C-SRRG framework.

| Model | Clinical Context | Split | Traditional Metrics | | | | F1-SRR-BERT | | |
|---|---|---|---|---|---|---|---|---|---|
| | | | BLEU | ROUGE-L | BERT Score | F1-RadGraph | Precision | Recall | F1-Score |
| CheXagent-3B | ✗ | Valid | 9.44 | 34.03 | 61.82 | 19.30 | 63.80 | 63.48 | 59.10 |
| | | Test | 7.83 | 29.40 | 59.82 | 16.13 | 57.18 | 59.18 | 54.27 |
| | | Test-reviewed | 7.42 | 28.60 | 58.35 | 13.71 | 51.32 | 56.34 | 49.74 |
| | ✓ | Valid | 7.52 | 32.99 | 60.93 | 17.90 | 66.28 | 61.75 | 59.04 |
| | | Test | 7.03 | 29.18 | 59.66 | 16.07 | 59.69 | 58.42 | 55.07 |
| | | Test-reviewed | 6.92 | 29.04 | 58.91 | 14.84 | 55.42 | 58.26 | 52.86 |
| MedGemma-4B | ✗ | Valid | 8.92 | 41.24 | 60.94 | 17.80 | 62.19 | 60.77 | 56.81 |
| | | Test | 7.15 | 37.84 | 59.09 | 15.35 | 56.27 | 57.01 | 52.69 |
| | | Test-reviewed | 7.57 | 35.91 | 58.35 | 14.57 | 51.69 | 54.42 | 49.51 |
| | ✓ | Valid | 11.76 | 46.26 | 64.28 | 24.25 | 65.78 | 66.77 | 62.31 |
| | | Test | 10.58 | 41.92 | 61.85 | 19.23 | 59.45 | 61.89 | 57.12 |
| | | Test-reviewed | 11.21 | 40.15 | 61.12 | 19.16 | 55.02 | 60.71 | 54.20 |
| Lingshu-7B | ✗ | Valid | 8.15 | 32.17 | 59.15 | 17.23 | 63.82 | 57.10 | 55.06 |
| | | Test | 6.65 | 27.27 | 57.18 | 13.87 | 56.03 | 51.55 | 49.33 |
| | | Test-reviewed | 7.04 | 27.70 | 57.37 | 13.49 | 52.34 | 52.85 | 48.37 |
| | ✓ | Valid | 11.77 | 38.46 | 64.82 | 25.29 | 69.42 | 63.57 | 62.48 |
| | | Test | 10.58 | 32.86 | 62.07 | 19.85 | 63.04 | 58.39 | 57.01 |
| | | Test-reviewed | 11.61 | 33.66 | 62.04 | 21.28 | 57.48 | 58.80 | 54.53 |

Table 5: **Effect of clinical context for train/eval on the C-SRR-Findings** using MedGemma-4B.

| Clinical Context | | Split | F1-SRR-BERT | | |
|---|---|---|---|---|---|
| Train | Eval | | Precision | Recall | F1-Score |
| ✗ | ✗ | Valid | 42.93 | 45.50 | 42.12 |
| | | Test | 42.32 | 41.38 | 40.19 |
| | | Test-reviewed | 44.49 | 42.94 | 41.92 |
| ✓ | ✗ | Valid | 47.00 | 47.09 | 45.35 |
| | | Test | 42.76 | 41.55 | 40.64 |
| | | Test-reviewed | 44.79 | 43.12 | 42.45 |
| ✗ | ✓ | Valid | 45.28 | 45.25 | 43.56 |
| | | Test | 43.02 | 40.94 | 40.44 |
| | | Test-reviewed | 44.40 | 41.50 | 41.36 |
| ✓ | ✓ | Valid | 50.52 | 49.68 | 48.42 |
| | | Test | 45.84 | 44.24 | 43.43 |
| | | Test-reviewed | 47.90 | 45.17 | 44.96 |

Table 6: **Effect of clinical context for train/eval on the C-SRR-Impression** using MedGemma-4B.

| Clinical Context | | Split | F1-SRR-BERT | | |
|---|---|---|---|---|---|
| Train | Eval | | Precision | Recall | F1-Score |
| ✗ | ✗ | Valid | 62.19 | 60.77 | 56.81 |
| | | Test | 56.27 | 57.01 | 52.69 |
| | | Test-reviewed | 51.69 | 54.42 | 49.51 |
| ✓ | ✗ | Valid | 63.87 | 61.86 | 58.45 |
| | | Test | 54.42 | 56.53 | 51.64 |
| | | Test-reviewed | 51.45 | 57.86 | 51.17 |
| ✗ | ✓ | Valid | 62.60 | 64.23 | 59.10 |
| | | Test | 53.34 | 59.11 | 52.59 |
| | | Test-reviewed | 49.35 | 58.68 | 50.66 |
| ✓ | ✓ | Valid | 65.78 | 66.77 | 62.31 |
| | | Test | 59.45 | 61.89 | 57.12 |
| | | Test-reviewed | 55.02 | 60.71 | 54.20 |

Table 7: **Ablation study on clinical context for the C-SRRG-Findings** using MedGemma-4B.

| Configuration | Split | F1-SRR-BERT | | |
|---|---|---|---|---|
| | | Precision | Recall | F1-Score |
| Single-view | Valid | 47.00 | 47.09 | 45.35 |
| | Test | 42.76 | 41.55 | 40.64 |
| | Test-reviewed | 44.79 | 43.12 | 42.45 |
| Multi-view | Valid | 47.46 | 47.21 | 45.80 |
| | Test | 44.44 | 42.57 | 41.92 |
| | Test-reviewed | 45.49 | 42.69 | 42.39 |
| + Indication | Valid | 46.95 | 46.06 | 44.85 |
| | Test | 44.62 | 42.56 | 41.98 |
| | Test-reviewed | 45.92 | 43.14 | 42.79 |
| + Technique | Valid | 50.35 | 49.27 | 48.24 |
| | Test | 45.50 | 43.89 | 43.15 |
| | Test-reviewed | 47.48 | 44.60 | 44.42 |
| + Comparison + Prior studies | Valid | 50.52 | 49.68 | 48.42 |
| | Test | 45.84 | 44.24 | 43.43 |
| | Test-reviewed | 47.90 | 45.17 | 44.96 |

Table 8: **Ablation study on clinical context for C-SRRG-Impression** using MedGemma-4B.

| Configuration | Split | F1-SRR-BERT | | |
|---|---|---|---|---|
| | | Precision | Recall | F1-Score |
| Single-view | Valid | 63.87 | 61.86 | 58.45 |
| | Test | 54.42 | 56.53 | 51.64 |
| | Test-reviewed | 51.45 | 57.86 | 51.17 |
| Multi-view | Valid | 65.74 | 62.92 | 59.89 |
| | Test | 55.70 | 58.11 | 53.36 |
| | Test-reviewed | 51.78 | 59.37 | 52.25 |
| + Indication | Valid | 66.91 | 65.02 | 61.67 |
| | Test | 58.47 | 59.11 | 55.00 |
| | Test-reviewed | 53.32 | 60.47 | 52.86 |
| + Technique | Valid | 65.91 | 65.65 | 61.78 |
| | Test | 58.65 | 60.24 | 55.88 |
| | Test-reviewed | 54.66 | 60.05 | 53.39 |
| + Comparison + Prior studies | Valid | 65.78 | 66.77 | 62.31 |
| | Test | 59.45 | 61.89 | 57.12 |
| | Test-reviewed | 55.02 | 60.71 | 54.20 |

**Effect of clinical context on training/evaluation.** We next conduct ablation studies with four settings: 1) train+eval without context (baseline); 2) train with, eval without; 3) train without, eval with; and 4) train+eval with context. Tabs. 5 and 6 report F1-SRR-BERT on C-SRRG-Findings and C-SRRG-Impression using MedGemma-4B. We find incorporating clinical context in only one phase provides limited improvement or slight degradation: *e.g.*, +3.23/+0.45/+0.53 (train ✓), or +1.44/+0.25/-0.56 (test ✓) in findings and +1.65/-1.05/+1.66 (train ✓), or +2.29/-0.1/+1.15 (test ✓) in impression, which shows the **benefit of using context in both phases** for SRRG performance.

**Impact of Each Clinical Context Component.** We ablate four clinical-context components, 1) multi-view images, 2) indication, 3) technique, and 4) prior studies with comparison, to isolate their contributions on the performance. Tabs. 7 and 8 report F1-SRR-BERT for each variant on the C-SRRG-Findings, C-SRRG-Impression, respectively. **All components contribute incrementally** to both tasks (except for few cases), with **performance being highest when using all available context**, which shows the importance of incorporating clinical context in SRRG.

**Mitigation of temporal hallucinations.** To quantify temporal hallucinations, we train MedGemma-4B under two conditions: **without** clinical context (**baseline**) and **with** clinical context (**C-SRRG**). We evaluate both on evaluation sets **without clinical context**, and count reports that contain one of the following 33 indicators: 1) time

Table 9: **Mitigation effect of temporal hallucination**.

| Task | Split | Temporal Hallucination Rate | | Mitigation |
|------|-------|------------------------------|------|------------|
| | | Baseline (✗) | C-SRRG (✓) | |
| Findings | Valid | 146/976 (15.0%) | 70/976 (7.2%) | -7.8% |
| | Test | 416/1459 (28.5%) | 194/1459 (13.3%) | -15.2% |
| | Test-reviewed | 49/233 (21.0%) | 21/233 (9.0%) | -12.0% |
| | Overall | 611/2668 (22.9%) | 285/2668 (10.7%) | -12.2% |
| Impression | Valid | 630/1505 (41.9%) | 364/1505 (24.2%) | -17.7% |
| | Test | 1012/2219 (45.6%) | 599/2219 (27.0%) | -18.6% |
| | Test-reviewed | 92/231 (39.8%) | 58/231 (25.1%) | -14.7% |
| | Overall | 1734/3955 (43.8%) | 1021/3955 (25.8%) | -18.0% |

references ('new', 'newly', 'recent', 'recently', 'previous', 'prior', 'interval', 'compared to', 'since', 'from prior'), 2) stability indicators ('unchanged', 'stable', 'persistent', 'persisting'), and 3) change indicators ('improved', 'improvement', 'worsened', 'worsening', 'increased', 'decreased', 'enlarging', 'reducing', 'progression', 'regression', 'evolving', 'evolve', 'developing', 'developed', 'resolving', 'resolved', 'temporal change', 'compare', 'comparison'). By this, we can detect whether the generated reports **contained hallucinations** by identifying inappropriate temporal references in the **absence of clinical context**. Tab. 9 shows that **clinical context substantially mitigates hallucinations**: Findings drop from 22.9% to 10.7% (–12.2%) and Impression from 43.8% to 25.8% (–18.0%). This shows that C-SRRG effectively handles *heterogeneous clinical context availability*, *i.e.*, the absence of clinical context, while successfully mitigating temporal hallucinations. We also conduct temporal hallucination analysis on CheXagent-3B and Lingshu-7B and find that CheXagent-3B C-SRRG-Findings model tends to over-generate specific temporal terms such as 'stable' after training with clinical context even when no prior studies are available, increasing overall temporal hallucinations as detailed in §C.1.

**Anatomical region analysis.** We compare organ-level performance for findings task against the baseline using the SRRG anatomical categories (Delbrouck et al., 2025). Tab. 10 reports the Category Score on the validation split using MedGemma-4B, with abbreviations: P = pleura, A = abdominal, H/M = hila/mediastinum, O = Other, L/A = lungs/airways, C = cardiovascular, M/C = musculoskeletal/chest

Table 10: **Organ-level Category Score** on the Valid split.

| Region | Baseline (✗) | | | C-SRRG (✓) | | |
|--------|-----------|--------|----------|-----------|--------|----------|
| | Precision | Recall | F1-Score | Precision | Recall | F1-Score |
| P | 47.69 | 39.62 | 41.72 | 58.2 | 52.39 | 52.77 |
| A | 8.0 | 8.0 | 8.0 | 17.86 | 17.86 | 17.86 |
| H/M | 37.91 | 36.55 | 36.97 | 34.25 | 33.07 | 33.41 |
| O | 9.21 | 7.94 | 8.2 | 12.17 | 10.13 | 10.6 |
| L/A | 40.67 | 64.67 | 45.68 | 57.25 | 62.24 | 55.69 |
| C | 71.77 | 68.6 | 68.99 | 73.28 | 70.26 | 70.51 |
| M/C | 26.22 | 25.26 | 25.55 | 43.2 | 41.97 | 42.3 |
| T/C/S | 57.04 | 64.18 | 58.45 | 59.81 | 64.39 | 60.35 |

wall, T/C/S = tubes/catheters/support devices. We observe that incorporating clinical context with proposed C-SRRG **improves the performance across all the anatomical regions**, except for H/M.

## 5 CONCLUSION, LIMITATIONS, AND FUTURE WORK

We introduced *contextualized structured radiology report generation* (**C-SRRG**), a framework that aligns with radiologists' diagnostic workflow by integrating rich clinical context including multi-view images, indication, imaging technique, and prior studies with comparisons. Through comprehensive evaluation of diverse medical MLLMs, we demonstrate that clinical context integration consistently enhances text quality, diagnostic accuracy, and reduces temporal hallucinations. We will publicly release our dataset, code, and model checkpoints to foster further research in C-SRRG and benefit the broader community.

**Limitations.** The C-SRRG dataset relies on synthetic LLM annotations from reformulated reports, which may introduce biases and subtle hallucinations. Our supervised fine-tuning approach with greedy decoding may limit the full capture of clinical reasoning processes. Computational and architectural constraints limited our evaluation to 7B parameter models with restricted multiple image processing capabilities (*e.g.*, Lingshu-7B and CheXagent-3B limited to 2 images) and context windows that constrain comprehensive longitudinal history integration. Additionally, our recency-based selection strategy for prioritizing recent studies, while capturing clinically relevant temporal information, may occasionally omit important historical context.

**Future work** should explore scaling to larger foundation models with extended-context capabilities, developing intelligent clinical context selection policies through learned strategies or retrieval-augmented approaches, and incorporating preference learning techniques with radiologists' feed-

back. Expanding to comprehensive clinical modalities also presents promising avenues for enhanced diagnostic accuracy, with detailed discussions provided in §A.

## REPRODUCIBILITY STATEMENT

We ensure reproducibility by building C-SRRG entirely from publicly available MIMIC-CXR (Johnson et al., 2019) and CheXpert-Plus (Chambon et al., 2024) datasets, with detailed documentation of our data processing pipeline including longitudinal patient history extraction and multi-view image integration in §3.1. All experimental configurations are specified in §4.1, including model hyperparameters (learning rate 2e-4, batch size 128, LoRA rank 32), exact data splits with patient-level separation, and standard evaluation metrics. We commit to publicly releasing our complete codebase, the C-SRRG dataset with clinical context annotations, trained model checkpoints, and documentation for dataset recreation. All experiments use reproducible libraries (Hugging Face PEFT (Mangrulkar et al., 2022), vLLM (Kwon et al., 2023)) on a single NVIDIA H100 GPU with fixed random seeds.

## ETHICS STATEMENT

Our work presents no new ethical concerns as C-SRRG is built entirely from existing de-identified public datasets (MIMIC-CXR and CheXpert Plus) that have undergone rigorous de-identification and received appropriate IRB approvals. No additional patient data was collected for this work, and all privacy protections from the source datasets are maintained. We acknowledge that automated report generation systems may produce hallucinations when referencing non-existent prior studies, which our work specifically addresses by incorporating comprehensive clinical context. Our dataset and models are intended solely for research purposes and should not be used for clinical decision-making without appropriate validation and regulatory approval. The computational requirements are modest (single GPU training), minimizing environmental impact while maintaining research accessibility.

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

# APPENDIX

This appendix provides comprehensive supplementary materials including limitations and future work discussions (§A), discussion on clinical context integration in prior work (§B), specific limitations of CheXagent-3B with long clinical contexts (§C), detailed interpretation of each clinical context component (§D), baseline performance evaluation of medical MLLMs before instruction tuning (§E), detailed dataset statistics with patient-level data splits (§F), prompt design examples for findings and impression generation (§G), complete instruction fine-tuning examples with clinical context integration (§H), and analysis of temporal hallucinations in radiology report generation (§I).

## A  LIMITATIONS AND FUTURE WORKS

### A.1  SYNTHETIC DATASET AND ANNOTATIONS

Our C-SRRG dataset builds upon the SRRG dataset (Delbrouck et al., 2025), which was generated through reformulation of free-form radiology reports using large language models. This synthetic generation process introduces several potential limitations that warrant careful consideration. The use of LLM-generated content may introduce subtle hallucinations or inconsistencies that could propagate through our training pipeline. Additionally, the reformulation process may inadvertently introduce biases present in the underlying language models, potentially affecting the diversity and clinical accuracy of the generated reports.

While these limitations are inherent to synthetic data generation, several factors mitigate their impact on our work. Prior work validated the structured report generation task through a rigorous reader study with five board-certified radiologists (Delbrouck et al., 2025). Five board-certified radiologists reviewed 464 reports sampled from the MIMIC-CXR test set and CheXpert Plus validation set, assessing the clinical validity of GPT-4 generated structured reports on the test-reviewed split. Their analysis shows high average similarity ratios, computed as 2 × matches / total tokens in original and edited texts: 0.77 for impressions and 0.88 for findings, indicating that modifications were generally minor and focused on enhancing clarity rather than altering fundamental diagnostic content. The utterance-level label validation achieved a 72% exact match rate and 74% average Jaccard similarity between GPT-generated consensus labels and radiologist-reviewed labels, confirming the reliability of the automated label extraction approach. We note that this expert evaluation was conducted specifically on the **test-reviewed** split rather than the full dataset, which was necessary given the resource-intensive nature of comprehensive radiologist review at this scale. Large-scale data may average out such errors across diverse examples, and the clinical validation from prior work suggests the error rate is manageable.

Furthermore, our evaluation protocol is designed to assess clinical validity rather than exact textual matching. F1-SRR-BERT captures semantic similarity appropriate for medical reporting where multiple valid phrasings exist, while organ-level breakdowns provide fine-grained anatomical assessment. Temporal hallucination analysis was conducted by detecting temporal keywords solely from model-generated structured reports, not from synthetically generated ground truth, naturally avoiding potential bias when comparing generated outputs with synthetic targets.

Nevertheless, our approach lacks regulatory clearance for clinical deployment, and practitioners must retain final authority over all diagnostic decisions. Given the inherent risk of errors or hallucinations in any AI system, rigorous clinical validation and expert oversight remain essential when exploring or extending this work.

### A.2  MODEL ARCHITECTURE AND SCALE LIMITATIONS

Our experimental evaluation faces several architectural constraints that limit the full potential of our approach. First, we are constrained to backbone models with parameters up to 7B, which likely

underestimates achievable performance. The computational and memory requirements of larger models present practical limitations for comprehensive evaluation across multiple architectures and clinical contexts. While exploring the effectiveness of C-SRRG across a broader range of model scales would provide valuable insights into how contextual signals scale with model capacity, several practical constraints limit our evaluation scope.

As noted in the §4.1, all experiments were conducted on a single NVIDIA H100 GPU, which fundamentally constrains the scale of models we could evaluate. More importantly, publicly available medical MLLMs offer only limited size options: MedGemma (Sellergren et al., 2025) provides 4B and 27B variants, while Lingshu (Team et al., 2025) provides 7B and 32B versions. These larger variants (e.g., MedGemma-27B or Lingshu-32B) not only exceed our available computational capacity but also surpass the model sizes commonly studied in academic fine-tuning and evaluation settings. Consequently, our current evaluation does not demonstrate how our findings generalize to larger MLLMs, representing an important direction for future work as computational resources and model accessibility continue to improve.

Second, the multimodal large language models employed were not originally designed to handle multiple images simultaneously. In our experiments, we encountered specific limitations requiring tailored approaches. **Lingshu-7B** (Team et al., 2025) was limited to 2 images due to computational efficiency constraints, while **CheXagent-3B** (Chen et al., 2024b) was similarly restricted to 2 images due to model constraints. In contrast, **MedGemma-4B** (Sellergren et al., 2025) demonstrated multi-image processing capabilities, requiring fewer tokens per image and enabling the use of more images in our longitudinal analysis setting.

These parameter and architectural constraints particularly impact the models' ability to effectively integrate complex temporal relationships and multi-modal clinical information across current and prior studies. However, ongoing research trends in long-context LLM development (Dao et al., 2022; Kwon et al., 2023) suggest that future model architectures will naturally address these limitations. Scaling to larger foundation models, exploring mixture-of-experts variants, and advances in multimodal attention mechanisms represent promising directions for substantial improvements.

### A.3 Clinical Context Integration and Selection Methodology

Our approach faces constraints in both the scope and selection of clinical contexts. We impose limits on the number of images and prior studies included per clinical case, with prioritization given to the most recent studies. This limitation stems from current multimodal architectures' context window constraints and computational overhead of processing extensive longitudinal histories. While this recency-based selection strategy captures the most clinically relevant temporal information, it may occasionally omit important historical context that could inform diagnostic reasoning. Additionally, our current implementation primarily utilizes publicly available datasets such as MIMIC-CXR (Johnson et al., 2019) and CheXpert-Plus (Chambon et al., 2024), limiting the diversity of clinical contexts.

Our design choice to use the two most recent prior studies is motivated by both clinical practice and technical constraints. Radiologists typically reference a limited number of relevant prior studies when writing reports. Empirical evidence shows that radiologists consult an average of 3.2 prior imaging studies per report (Haygood et al., 2018), as clinical workflows prioritize the most recent and pertinent examinations for interval change detection. Our default setting of two most recent studies aligns with this standard practice in radiology while remaining within the practical constraints of current medical MLLMs. Current publicly available medical MLLMs also face significant context-length limitations. Some patients have more than 60-100 longitudinal studies as shown in Fig. 7, which exceeds the practical capacity of existing medical MLLMs.

While our approach uses the most recent prior studies, patients with extensive longitudinal histories may have clinically important examinations from earlier studies. This challenge could be addressed through two complementary approaches: advances in efficient attention mechanisms (Dao et al., 2022; Kwon et al., 2023) that enable processing longer temporal contexts, and retrieval-augmented generation methods (Lewis et al., 2021; Gao et al., 2024) that intelligently select the most informative prior studies based on clinical relevance, potentially integrated within agentic systems for dynamic study selection. Our design choice to truncate longitudinal contexts is motivated by both

clinical practice and technical limitations, but exploring longer patient histories remains an important research direction.

The scope could be significantly expanded to include additional clinical modalities such as CT imaging, Electronic Health Records (EHR) (Häyrinen et al., 2008), and comprehensive patient histories. Future work should explore learned selection policies that intelligently identify the most informative clinical contexts and optimize longitudinal coverage. Retrieval-augmented generation approaches over Picture Archiving and Communication Systems (PACS) (Andriole, 2023) and EHR systems could dynamically surface the most relevant historical information for each case, unlocking the untapped potential for richer clinical context integration.

Our focus on imaging-centric clinical contexts (multi-view images, indication, technique, and prior studies) is motivated by established clinical practice and empirical evidence. Prior works demonstrate that providing additional clinical information enhances interpretation accuracy and reporting confidence (Castillo et al., 2020), with best practice guidelines (Kahn et al., 2009) establishing indication, technique, and comparison as standard components of high-quality radiology reports. The structured radiology reporting framework (Delbrouck et al., 2025) explicitly defines these elements (Exam Type, History, Technique, Comparison, Findings, Impression) as standard sections, reflecting clinical practice. Our empirical results further demonstrate the effectiveness of these chosen imaging-centric clinical contexts and their substantial impact on reducing temporal hallucinations.

While additional patient information such as lab tests, demographics, and comprehensive EHR records could provide complementary value, integrating such diverse sources introduces additional complexity. Data heterogeneity, privacy considerations, availability across clinical settings, and computational requirements present practical challenges. The chosen clinical contexts are motivated by established clinical studies and supported by our empirical observations, while investigating the effectiveness of additional patient information remains an important direction for future work.

## A.4    TRAINING METHODOLOGY AND DECODING STRATEGY

Our training approach is restricted to supervised fine-tuning with greedy decoding for reproducibility and computational efficiency. This methodology, while providing stable and consistent results, may not fully capture the nuanced decision-making processes that characterize expert radiological interpretation. The supervised learning paradigm limits the model's ability to learn from comparative feedback and iterative refinement that occurs in clinical practice. Incorporating preference learning techniques and Reinforcement Learning (RL)-based methods with radiologists' feedback, such as Proximal Policy Optimization (PPO) (Schulman et al., 2017) or Direct Preference Optimization (DPO) (Rafailov et al., 2023), could enhance the fidelity and clinical appropriateness of generated reports. Furthermore, exploring retrieval-conditioned decoding strategies could improve temporal consistency and reduce hallucinations by grounding generation in verified clinical contexts.

## A.5    RADIOLOGIST-IN-THE-LOOP EVALUATION

Our evaluation framework relies exclusively on automated metrics rather than radiologist-in-the-loop assessment. While radiologist evaluation would provide crucial insights beyond automated metrics, including clinical appropriateness of terminology, coherence of diagnostic reasoning, and the impact of reduced temporal hallucinations on clinical decision-making, our work establishes technical foundations through comprehensive automated evaluation. Nevertheless, several factors support the validity of our automated approach: the underlying dataset quality has been validated through expert assessment in prior work (Delbrouck et al., 2025) as discussed in §A, our automated metrics (F1-SRR-BERT, F1-RadGraph) correlate with human judgment (Delbrouck et al., 2025; 2022), and our findings remain consistent across multiple model architectures, suggesting robust improvements from clinical context integration.

Conducting radiologist evaluation presents significant resource challenges, including funding for expert compensation, coordination of radiologist availability. Therefore, given these constraints, radiologist-in-the-loop evaluation represents an important direction for future work as resources become available, particularly when translating these methods toward clinical deployment where regulatory approval and clinical validation become essential requirements.

## A.6 GENERALIZATION TO OTHER IMAGING MODALITIES

The proposed C-SRRG framework is fundamentally motivated by the diagnostic workflow of radiologists, where clinical context plays a crucial role in report generation across all imaging modalities, not limited to chest X-rays. We believe C-SRRG demonstrates strong potential to generalize to other imaging modalities such as computed tomography (CT), magnetic resonance imaging (MRI), ultrasound, and mammography, as these modalities similarly rely on contextual information for accurate interpretation (Hattori et al., 2021). The core components (*e.g.*, multi-view images, clinical indication, imaging technique, and prior studies) are naturally applicable across modalities, though specific instantiations may vary (*e.g.*, CT involves multiple slices rather than views, MRI includes diverse sequences with different contrasts).

However, each imaging modality presents unique characteristics and reporting conventions that may require modality-specific adaptations. Empirical validation on other imaging modalities would be valuable to confirm generalization and identify the specific adaptations required for optimal performance in each domain. Systematic evaluation of C-SRRG across diverse imaging modalities represents an important direction for exploring its broader clinical utility beyond chest radiography.

## B CLINICAL CONTEXT INTEGRATION IN PRIOR WORK

Prior works have explored incorporating clinical context into radiology report generation, though primarily considering partial contexts in isolation or limited combinations. Some works (Yuan et al., 2019; Miao et al., 2024) incorporate multi-view images, while others explore longitudinal data (Zhu et al., 2023b; Wang et al., 2024; Serra et al., 2023). Recent attempts to combine multiple contexts still face fundamental constraints: fixed input structures (*e.g.*, MAIRA-2 (Bannur et al., 2024) is constrained to exactly three images), limited temporal scope (typically only the immediate previous study), and complex multi-stage training procedures (Liu et al., 2025b;a). In contrast, our work systematically investigates comprehensive clinical context integration with variable numbers of multi-view images, indication text, technique specifications, and multiple prior studies with their complete reports. The comprehensive ablation studies (Tabs. 5 and 6) demonstrate that each context component provides complementary benefits, yielding substantial improvements (2–7% F1-SRR-BERT) and significant reductions in temporal hallucinations (12–18%).

Critically, previous works focused exclusively on free-form report generation, overlooking structured reports despite their clinical importance for clarity, consistency, and adherence to reporting standards (Kahn et al., 2009; Weiss & Langlotz, 2008; Delbrouck et al., 2025). The structured report generation task, recently introduced by prior work (Delbrouck et al., 2025), has not been studied in the context of comprehensive clinical contextualization.

Furthermore, prior works employ specialized architectural components that fundamentally limit their applicability to evaluation on the proposed benchmark. These include multi-view fusion with cross-view consistency (Yuan et al., 2019), multi-positive contrastive learning (Miao et al., 2024; Liu et al., 2025b), posterior and prior knowledge distillation (Liu et al., 2021), graph-based disease progression reasoning (Hou et al., 2023a), group causal transformers for longitudinal aggregation (Wang et al., 2024), and multi-stage contrastive learning (Liu et al., 2025a). These dedicated architectural modifications are specifically designed for their respective context integration approaches and are fundamentally incompatible with recent medical multimodal large language models. Consequently, only recent medical MLLMs (*e.g.*, CheXagent-3B (Chen et al., 2024b), MedGemma-4B (Sellergren et al., 2025), Lingshu-7B (Team et al., 2025)) can be evaluated on the proposed datasets, leveraging their flexibility in handling diverse multimodal inputs.

## C LIMITATION IN CHEXAGENT-3B

### C.1 INCREASED TEMPORAL HALLUCINATION ON C-SRRG-FINDINGS

We conduct additional temporal hallucination analysis with CheXagent-3B on C-SRRG-Findings. We can only conduct this analysis on the C-SRRG-Findings, as the C-SRRG-Impression model was trained solely with indication inputs without clinical history due to the training failure behavior

described in §4.1. Consequently, the impression model lacks access to temporal information from prior reports, making temporal hallucination analysis infeasible for this task.

Table 11: **Temporal hallucination rates with all temporal keywords for CheXagent-3B.**

| Task | Split | Temporal Hallucination Rate | | Mitigation |
|------|-------|-----------------------------|--|------------|
| | | **Baseline (✗)** | **C-SRRG (✓)** | |
| Findings | Valid | 170/976 (17.4%) | 280/976 (28.7%) | +11.3% |
| | Test | 462/1459 (31.7%) | 614/1459 (42.1%) | +10.4% |
| | Test-reviewed | 65/233 (27.9%) | 77/233 (33.0%) | +5.1% |
| | Overall | 697/2668 (26.1%) | 971/2668 (36.4%) | +10.3% |

Table 12: **Temporal keyword distribution comparison for CheXagent-3B.** The table shows the frequency of temporal keywords in generated findings for the baseline model (trained without clinical context) and C-SRRG (trained with clinical context), both evaluated on test sets without clinical context. Only keywords that appear at least once in either model are included.

| Keyword | Temporal Keyword Counts | | Difference | Change (%) |
|---------|-------------------------|--|------------|------------|
| | **Baseline (✗)** | **C-SRRG (✓)** | | |
| stable | 307 | 583 | +276 | +89.9% |
| unchanged | 319 | 355 | +36 | +11.3% |
| interval | 14 | 23 | +9 | +64.3% |
| worse | 20 | 21 | +1 | +5.0% |
| previous | 0 | 1 | +1 | +100.0% |
| similar | 0 | 1 | +1 | +100.0% |
| decreased | 1 | 1 | 0 | 0.0% |
| recent | 3 | 2 | -1 | -33.3% |
| better | 1 | 0 | -1 | -100.0% |
| improved | 9 | 7 | -2 | -22.2% |
| persistent | 28 | 13 | -15 | -53.6% |
| from prior | 42 | 0 | -42 | -100.0% |
| prior | 45 | 1 | -44 | -97.8% |
| new | 137 | 88 | -49 | -35.8% |
| increased | 81 | 26 | -55 | -67.9% |
| Overall | 1007 | 1122 | +115 | +11.4% |

While CheXagent-3B shows improvements in the F1-SRR-BERT metric when trained with clinical context on the C-SRRG-Findings task (Tab. 3), the model exhibits unexpected increases in temporal hallucinations as detailed in Tab. 11. Across all evaluation splits (validate, test, and test-reviewed), the model trained with clinical context exhibits higher rates of temporal keyword usage compared to the baseline, with an overall hallucination rate of 36.4% versus 26.1%, representing a 10.3 percentage point increase. On the validation split, the hallucination rate increases from 17.4% to 28.7% (+11.3%); on the test split, from 31.7% to 42.1% (+10.4%); and on the test-reviewed split, from 27.9% to 33.0% (+5.1%).

To understand the nature of these hallucinations, we further analyzed the distribution of specific temporal keywords in Tab. 12. The analysis reveals that the model trained with clinical context demonstrates a marked preference for specific temporal terms, with "stable" increasing by 89.9% (from 307 to 583 occurrences) and "unchanged" increasing by 11.3% (from 319 to 355 occurrences). This shift suggests that when trained with clinical context, the model develops an over-reliance on specific temporal terms while inappropriately applying them even when no prior information is available, rather than generating more nuanced temporal comparisons.

Importantly, such hallucination increases were not observed in other models. As shown in Tab. 9 and Tab. 13, temporal hallucination rates were substantially reduced for both MedGemma-4B and LingShu-7B when trained with clinical context. Furthermore, as demonstrated in Tab. 14 and Tab. 15, the tendency to overuse temporal keywords was also not observed in these models, with both showing significant reductions in total temporal keyword usage.

Table 13: **Mitigation effect of temporal hallucination for LingShu-7B**. The table shows temporal hallucination rates across different data splits for the baseline model (trained without clinical context) and the model trained with clinical context (C-SRRG). The hallucination effect of MedGemma-4B is shown in Table 9.

| Task | Split | Temporal Hallucination Rate | | Mitigation |
|------|-------|------------------|------------------|------------|
| | | **Baseline (✗)** | **C-SRRG (✓)** | |
| Findings | Valid | 174/976 (17.8%) | 140/976 (14.3%) | -3.5% |
| | Test | 491/1459 (33.7%) | 369/1459 (25.3%) | -8.4% |
| | Test-reviewed | 64/233 (27.5%) | 42/233 (18.0%) | -9.4% |
| | Overall | 729/2668 (27.3%) | 551/2668 (20.7%) | -6.7% |
| Impression | Valid | 692/1505 (46.0%) | 382/1505 (25.4%) | -20.6% |
| | Test | 1153/2219 (52.0%) | 622/2219 (28.0%) | -23.9% |
| | Test-reviewed | 116/231 (50.2%) | 50/231 (21.6%) | -28.6% |
| | Overall | 1961/3955 (49.6%) | 1054/3955 (26.6%) | -22.9% |

Table 14: **Temporal keyword distribution for MedGemma-4B**. The table shows the frequency of temporal keywords in generated findings (left) and impressions (right) for the baseline model (trained without clinical context) and the model trained with clinical context. Only keywords that appear at least once in either model are included.

| Keyword | Temporal Keyword Counts | | Difference | Change (%) | Keyword | Temporal Keyword Counts | | Difference | Change (%) |
|---------|------------|-----------|------------|------------|---------|------------|-----------|------------|------------|
| | **Baseline (✗)** | **C-SRRG (✓)** | | | | **Baseline (✗)** | **C-SRRG (✓)** | | |
| increased | 12 | 52 | +40 | +333.3% | new | 149 | 166 | +17 | +11.4% |
| interval | 13 | 19 | +6 | +46.2% | worse | 93 | 74 | -19 | -20.4% |
| new | 62 | 66 | +4 | +6.5% | progression | 29 | 1 | -28 | -96.6% |
| persistent | 6 | 7 | +1 | +16.7% | increased | 52 | 19 | -33 | -63.5% |
| compare | 1 | 0 | -1 | -100.0% | compare | 39 | 0 | -39 | -100.0% |
| compared to | 1 | 0 | -1 | -100.0% | compared to | 39 | 0 | -39 | -100.0% |
| worsened | 1 | 0 | -1 | -100.0% | from prior | 46 | 5 | -41 | -89.1% |
| better | 1 | 0 | -1 | -100.0% | recent | 6 | 1 | -5 | -83.3% |
| similar | 1 | 0 | -1 | -100.0% | decreased | 6 | 0 | -6 | -100.0% |
| worse | 6 | 2 | -4 | -66.7% | resolving | 1 | 0 | -1 | -100.0% |
| decreased | 5 | 1 | -4 | -80.0% | previous | 21 | 1 | -20 | -95.2% |
| improved | 7 | 0 | -7 | -100.0% | prior | 136 | 27 | -109 | -80.1% |
| stable | 86 | 46 | -40 | -46.5% | interval | 196 | 79 | -117 | -59.7% |
| from prior | 112 | 33 | -79 | -70.5% | persistent | 217 | 46 | -171 | -78.8% |
| prior | 115 | 34 | -81 | -70.4% | unchanged | 457 | 202 | -255 | -55.8% |
| unchanged | 500 | 113 | -387 | -77.4% | stable | 1487 | 900 | -587 | -39.5% |
| **Total** | 929 | 373 | -556 | -59.8% | **Total** | 2974 | 1521 | -1453 | -48.9% |

Given this keyword distribution pattern, we conducted additional analyses excluding specific temporal terms to isolate their contribution to the hallucination problem. Tab. 16 presents results when excluding the keyword "stable," revealing a remarkable reversal: C-SRRG now achieves lower hallucination rates than the baseline across all splits (validate: 10.8% vs 11.9%, -1.1%; test: 21.0% vs 23.7%, -2.7%; test-reviewed: 20.2% vs 21.0%, -0.8%). The overall hallucination rate drops from 36.4% to 17.2%, demonstrating that the single keyword "stable" accounts for over half of the model trained with clinical context's apparent temporal hallucinations.

When we further exclude both "stable" and "unchanged", this advantage becomes even more pronounced (Tab. 17). The model trained with clinical context achieves substantially lower hallucination rates across all splits: validate (4.9% vs 7.7%, -2.8%), test (7.1% vs 14.1%, -7.0%), and test-reviewed (7.3% vs 13.7%, -6.4%). The overall hallucination rate drops dramatically from 36.4% to 6.3%, while the baseline decreases from 26.1% to 11.7%. This analysis reveals that 83% of the model trained with clinical context's temporal hallucinations (30.1 percentage points out of 36.4%) are attributable solely to these two terms, whereas they account for only 55% of the baseline's hallucinations (14.4 percentage points out of 26.1%). Critically, when these terms are excluded, the model trained with clinical context demonstrates superior performance with 5.4 percentage points lower hallucination rate than the baseline, indicating that it actually learned more conservative and appropriate usage of other temporal language.

These findings suggest that CheXagent-3B's apparent temporal hallucination problem is not a fundamental failure of temporal reasoning, but rather the model learning to incorporate certain temporal terms ("stable," "unchanged") as stylistic markers during training on longitudinal data. However, this learned behavior is problematic from a clinical perspective: assertions like "effusion is stable"

Table 15: **Temporal keyword distribution for LingShu-7B**. The table shows the frequency of temporal keywords in generated findings (left) and impressions (right) for the baseline model (trained without clinical context) and the model trained with clinical context. Only keywords that appear at least once in either model are included.

| Keyword | Temporal Keyword Counts | | Difference | Change (%) | Keyword | Temporal Keyword Counts | | Difference | Change (%) |
|---|---|---|---|---|---|---|---|---|---|
| | Baseline (✗) | C-SRRG (✓) | | | | Baseline (✗) | C-SRRG (✓) | | |
| interval | 11 | 37 | +26 | +236.4% | worse | 77 | 95 | +18 | +23.4% |
| new | 27 | 54 | +27 | +100.0% | improved | 1 | 0 | -1 | -100.0% |
| increased | 27 | 48 | +21 | +77.8% | worsened | 1 | 0 | -1 | -100.0% |
| worse | 5 | 12 | +7 | +140.0% | decreased | 1 | 0 | -1 | -100.0% |
| worsened | 0 | 1 | +1 | +100.0% | resolving | 2 | 0 | -2 | -100.0% |
| better | 0 | 1 | +1 | +100.0% | progression | 2 | 0 | -2 | -100.0% |
| decreased | 1 | 0 | -1 | -100.0% | recent | 15 | 4 | -11 | -73.3% |
| compare | 2 | 0 | -2 | -100.0% | new | 104 | 71 | -33 | -31.7% |
| compared to | 2 | 0 | -2 | -100.0% | from prior | 35 | 0 | -35 | -100.0% |
| improved | 5 | 0 | -5 | -100.0% | increased | 57 | 3 | -54 | -94.7% |
| previous | 5 | 0 | -5 | -100.0% | interval | 179 | 102 | -77 | -43.0% |
| similar | 5 | 0 | -5 | -100.0% | prior | 130 | 31 | -99 | -76.2% |
| stable | 68 | 60 | -8 | -11.8% | previous | 135 | 1 | -134 | -99.3% |
| persistent | 11 | 0 | -11 | -100.0% | compare | 178 | 1 | -177 | -99.4% |
| from prior | 129 | 33 | -96 | -74.4% | compared to | 178 | 1 | -177 | -99.4% |
| prior | 131 | 34 | -97 | -74.0% | persistent | 245 | 47 | -198 | -80.8% |
| unchanged | 653 | 391 | -262 | -40.1% | unchanged | 591 | 209 | -382 | -64.6% |
| | | | | | stable | 1618 | 960 | -658 | -40.7% |
| **Total** | 1082 | 671 | -411 | -38.0% | **Total** | 3549 | 1525 | -2024 | -57.0% |

Table 16: **Temporal hallucination rates without "stable" for CheXagent-3B**.

| Task | Split | Temporal Hallucination Rate | | Mitigation |
|---|---|---|---|---|
| | | Baseline (✗) | C-SRRG (✓) | |
| Findings | Valid | 116/976 (11.9%) | 105/976 (10.8%) | -1.1% |
| | Test | 346/1459 (23.7%) | 306/1459 (21.0%) | -2.7% |
| | Test-reviewed | 49/233 (21.0%) | 47/233 (20.2%) | -0.8% |
| | Overall | 511/2668 (19.2%) | 458/2668 (17.2%) | -2.0% |

create false confidence by implying that prior comparison occurred when no prior study is available, which is clinically misleading.

## C.2 TRAINING FAILURE ON C-SRRG-IMPRESSION

While most models show improvement with clinical context, CheXagent-3B exhibits a **critical failure** in following the structured report format instructions when provided with full clinical context. Instead of generating properly formatted impression sections with numbered findings, the model frequently produces single-word outputs or generic phrases. For instance, when the expected format is a multi-point structured impression such as "1. Slight decrease in size of the right apicolateral pneumothorax with chest tube in place. 2. Unchanged multifocal right-sided pulmonary opacities...", CheXagent-3B often generates only "Pneumothorax" or "Pneumonia". This format degradation is widespread, with the model generating non-structured outputs like "No acute cardiopulmonary process" or "Pulmonary edema" rather than detailed clinical impressions.

The performance metrics in Tab. 18 reveal the severity of this issue: when provided with full clinical context, traditional metrics plummet dramatically (BLEU: 9.44→2.57, ROUGE-L: 34.03→21.76, BERTScore: 61.82→40.10 on validation set). This catastrophic degradation suggests that CheXagent-3B, likely trained primarily on shorter sequence lengths, struggles to process and integrate the extensive clinical context while maintaining adherence to the structured output format. The model's inability to handle long input sequences effectively undermines its utility for clinical applications requiring comprehensive context integration.

## C.3 SPECIFIC LIMITATIONS IN CHEXAGENT-3B

We hypothesize that these failures of CheXagent-3B when trained with rich clinical context stems from two fundamental limitations inherent to the model design.

Table 17: **Temporal hallucination rates without "stable" and "unchanged" for CheXagent-3B.**

| Task | Split | Temporal Hallucination Rate | | Mitigation |
|------|-------|-----------------------------|-----------------------|-----------|
| | | Baseline (✗) | C-SRRG (✓) | |
| Findings | Valid | 75/976 (7.7%) | 48/976 (4.9%) | -2.8% |
| | Test | 206/1459 (14.1%) | 104/1459 (7.1%) | -7.0% |
| | Test-reviewed | 32/233 (13.7%) | 17/233 (7.3%) | -6.4% |
| | Overall | 313/2668 (11.7%) | 169/2668 (6.3%) | -5.4% |

Table 18: **Performance degradation of CheXagent-3B on C-SRRG-Impression with full clinical context.**
The model shows dramatic drops across all metrics when provided with complete clinical context.

| Model | Full Clinical Context | Split | Traditional Metrics | | | | F1-SRR-BERT | | |
|-------|----------------------|-------|------|---------|---------------|----------------|-----------|--------|---------------|
| | | | BLEU | ROUGE-L | BERT Score | F1-RadGraph | Precision | Recall | F1-Score |
| CheXagent-3B | ✗ | Valid | 9.44 | 34.03 | 61.82 | 19.30 | 63.80 | 63.48 | 59.10 |
| | | Test | 7.83 | 29.40 | 59.82 | 16.13 | 57.18 | 59.18 | 54.27 |
| | | Test-reviewed | 7.42 | 28.60 | 58.35 | 13.71 | 51.32 | 56.34 | 49.74 |
| | ✓ | Valid | 2.57 | 21.76 | 40.10 | 13.10 | 74.48 | 49.40 | 54.05 |
| | | Test | 2.40 | 17.54 | 33.79 | 9.78 | 66.56 | 41.04 | 45.99 |
| | | Test-reviewed | 2.89 | 19.61 | 37.88 | 11.87 | 64.18 | 41.27 | 46.44 |

**Architectural and Capacity Constraints.** CheXagent-3B is built upon Phi-2 (Li et al., 2023d), a 2.7 billion parameter decoder-only transformer that was pretrained with a maximum sequence length of 2,048 tokens. While CheXagent-3B extends this to 4,096 tokens during fine-tuning (Chen et al., 2024b), this capacity remains insufficient for processing rich clinical context that combines visual tokens (ViT sequence length of 1,024), imaging techniques, clinical indications, and extensive prior study narratives. The relatively small model capacity (3.1 billion parameters total, including the vision encoder) may be further insufficient to simultaneously encode rich visual features, process extensive clinical text, maintain attention over long contexts, and generate structured outputs conforming to specific format constraints. Furthermore, the vision-language projector consists of only a two-layer multi-layer perceptron (MLP) projecting visual features from dimension 1,024 to 2,560 (Chen et al., 2024b), which may create a bottleneck when the model must integrate complex visual semantics with extensive textual context.

**Training Data Characteristics and Template Overfitting.** The CheXinstruct dataset, while large-scale with 8.5 million samples across 35 tasks, predominantly consists of short, task-specific instruction-response pairs using ten manually-defined templates for each task (Chen et al., 2024b). The dataset composition emphasizes perception tasks with concise outputs: view classification with three-choice answers (AP/PA/Lateral), disease identification with binary Yes/No responses, and findings generation from CXR images. For example, a typical training sample for disease classification might consist of: instruction ("Identify if pneumothorax is present in this image"), a single CXR image, and a short response ("Yes" or "No"). Critically, none of the 35 tasks in CheXinstruct explicitly require processing long clinical context alongside images to generate structured radiology reports. This creates a fundamental gap: the model learns to generate reports from visual input with minimal textual context, but fails when required to integrate extensive clinical narrative while maintaining structured output format. The temporal hallucination increase (§C.1) and the training failure (§C.2) suggest that CheXagent-3B has difficulty generalizing beyond the relatively constrained instruction formats encountered during training.

# D CLINICAL INTERPRETATION OF EACH CONTEXT COMPONENT

In this section, we provide detailed clinical interpretation of each context component introduced in §3.1, explaining their distinct roles in radiological interpretation and the clinical implications of their omission.

**Multi-view Images.** Multi-view chest radiography provides complementary anatomical information essential for comprehensive diagnostic assessment. For instance, lateral views enable visualiza-

tion of retrocardiac and retrosternal regions obscured by the cardiac silhouette in frontal projections. Without multi-view information, pathologies such as middle lobe pneumonia or retrosternal masses may remain undetected. Multiple views allow radiologists to triangulate spatial locations and distinguish true pathology from overlapping normal structures.

**Clinical Indication.** Clinical indication provides essential context regarding patient symptoms and clinical history that guide radiological interpretation. For example, fever and productive cough direct attention toward infectious etiologies, while trauma history shifts focus to skeletal injuries and pneumothorax. Without clinical indication, models cannot appropriately integrate clinical significance of findings, potentially emphasizing incidental observations while overlooking relevant pathologies.

**Imaging Technique.** Technical parameters document the imaging protocol and inherent limitations crucial for proper interpretation. These include patient positioning, projection type, inspiratory effort, and whether the study was portable. Omitting technique information can lead models to misinterpret technical artifacts as pathological findings. for instance, limited inspiratory effort produces apparent cardiomegaly, while portable anteroposterior examinations magnify cardiac silhouette. Recognition of these factors prevents false-positive interpretations.

**Prior Studies.** Temporal comparison with prior examinations enables detection of interval changes often more clinically significant than static findings. Prior study information allows tracking disease progression, assessing treatment response, and distinguishing stable chronic findings from acute changes. For example, serial nodule measurements differentiate stable benign lesions from growing malignancies, directly impacting management decisions. Without prior studies, models may generate temporal hallucinations or fail to detect critical patterns such as progressive infiltrates or worsening effusions.

In summary, each clinical context component used in our setting serves a distinct role in radiological interpretation. Context omission not only reduces model performance quantitatively (Tabs. 5 and 6), but fundamentally limits the clinical utility of generated reports by removing essential diagnostic information.

## E   EVALUATION BEFORE INSTRUCTION TUNING

Medical MLLMs possess inherent capabilities to generate free-form radiology reports even without task-specific fine-tuning, as these models are typically pretrained on diverse medical image-text pairs and general medical knowledge. However, the structured radiology report generation (SRRG) task imposes specific format requirements that fundamentally affect whether pretrained models can be meaningfully evaluated. In this section, we investigate the baseline performance of pretrained medical MLLMs before instruction tuning to establish their zero-shot capabilities and demonstrate the necessity of task-specific fine-tuning.

### E.1   TASK-SPECIFIC FORMAT REQUIREMENTS

While existing medical MLLMs have been exposed to diverse radiology reports during pretraining, they have not encountered the specific structured format required by the SRRG task (Delbrouck et al., 2025). The evaluation challenges that arise from this format mismatch vary significantly depending on which report section is being generated.

For the **findings generation** task, models must produce reports in a specific structured format with predefined anatomical sections such as "Lungs and Airways:", "Pleura:", "Cardiovascular:", "Hila and Mediastinum:", and "Tubes, Catheters, and Support Devices:". Each section must contain bullet-pointed observations organized by anatomical region. However, pretrained models without instruction fine-tuning cannot generate this structured format; instead, they produce free-form narrative text rather than adhering to the required section structure. Since the evaluation metrics and clinical utility depend critically on this structured organization, findings generation cannot be meaningfully evaluated without fine-tuning.

In contrast, the **impression generation** task requires only numbered diagnostic conclusions without strict section headers, making it feasible to evaluate pretrained models. While the quality and clinical appropriateness may vary, pretrained models can generate numbered lists that align with the expected output format, enabling quantitative evaluation.

## E.2 BASELINE PERFORMANCE WITHOUT FINE-TUNING

We evaluate the baseline performance of pretrained medical MLLMs on the impression generation task without any instruction fine-tuning. Models are evaluated both with (✓) and without (✗) clinical context provided at inference time to assess whether clinical information improves zero-shot report generation capabilities.

Table 19: **Baseline performance on impression generation without instruction fine-tuning**. Models are evaluated with (✓) and without (✗) clinical context at inference time.

| Model | Clinical Context | Split | Traditional Metrics | | | | F1-SRR-BERT | | |
|---|---|---|---|---|---|---|---|---|---|
| | | | BLEU | ROUGE-L | BERT Score | F1-RadGraph | Precision | Recall | F1-Score |
| CheXagent-3B | ✗ | Valid | 1.03 | 18.71 | 30.00 | 13.80 | 67.65 | 37.64 | 44.71 |
| | | Test | 0.98 | 15.19 | 21.22 | 12.27 | 60.07 | 28.65 | 36.14 |
| | | Test-reviewed | 1.13 | 16.02 | 22.56 | 13.20 | 59.20 | 32.76 | 39.70 |
| | ✓ | Valid | 1.44 | 19.13 | 30.23 | 13.98 | 67.66 | 38.60 | 45.47 |
| | | Test | 1.45 | 15.75 | 25.83 | 12.47 | 59.77 | 30.66 | 37.40 |
| | | Test-reviewed | 1.32 | 15.91 | 24.93 | 12.38 | 57.44 | 31.81 | 38.65 |
| MedGemma-4B | ✗ | Valid | 1.23 | 14.26 | 34.10 | 9.11 | 68.13 | 53.97 | 55.06 |
| | | Test | 1.21 | 13.44 | 32.86 | 9.60 | 60.07 | 48.46 | 48.89 |
| | | Test-reviewed | 0.70 | 11.73 | 29.99 | 7.58 | 55.20 | 48.68 | 47.50 |
| | ✓ | Valid | 1.80 | 15.04 | 37.60 | 11.29 | 66.18 | 61.74 | 58.88 |
| | | Test | 1.85 | 15.02 | 38.06 | 11.76 | 59.21 | 57.54 | 53.54 |
| | | Test-reviewed | 1.17 | 13.16 | 34.88 | 8.94 | 56.39 | 54.78 | 50.78 |
| Lingshu-7B | ✗ | Valid | 1.46 | 14.11 | 37.18 | 9.33 | 64.96 | 41.23 | 46.26 |
| | | Test | 1.58 | 13.90 | 39.37 | 10.40 | 55.45 | 34.80 | 39.02 |
| | | Test-reviewed | 1.27 | 13.23 | 37.38 | 9.34 | 49.93 | 33.17 | 36.56 |
| | ✓ | Valid | 4.44 | 21.29 | 50.03 | 12.54 | 61.45 | 51.64 | 51.44 |
| | | Test | 4.70 | 21.92 | 52.44 | 12.80 | 54.50 | 46.05 | 45.11 |
| | | Test-reviewed | 2.22 | 14.81 | 39.97 | 9.12 | 51.69 | 36.37 | 39.23 |

## E.3 ANALYSIS AND OBSERVATIONS

The results in Tab. 19 reveal several important observations about pretrained model capabilities and the role of clinical context.

**Substantial Performance Gap Between Pretrained and Fine-Tuned Models.** Pretrained models achieve substantially lower F1-SRR-BERT scores compared to all fine-tuned models reported in the main paper (Tab. 4): CheXagent-3B (36.14–45.47), MedGemma-4B (47.50–58.88), and Lingshu-7B (36.56–51.44). The only exception is MedGemma-4B when evaluated with clinical context on the validation set, which achieved 58.88, surpassing the same model trained without clinical context (56.81 from Tab. 4). However, when MedGemma-4B is trained with clinical context as shown in the ablation study (Tab. 5), it achieves 59.10, which still exceeds the pretrained baseline of 58.88. This demonstrates that fine-tuning with appropriate clinical context consistently yields superior performance.

**Model-Specific Baseline Capabilities and Improvement Potential.** MedGemma-4B exhibits stronger baseline performance than Lingshu-7B across all settings before fine-tuning (47.50–58.88 vs. 36.56–51.44). However, after fine-tuning, Lingshu-7B demonstrates more substantial improvements, ultimately achieving competitive or superior performance (Tab. 4). This suggests that larger models may benefit more from task-specific fine-tuning despite weaker initial capabilities, possibly due to greater capacity to learn structured output formats and integrate complex clinical reasoning patterns.

**Clinical Context Improves Zero-Shot Performance.** Notably, even without fine-tuning, providing clinical context at evaluation time consistently improves baseline performance across most settings, underscoring its fundamental importance. Specifically, clinical context improves F1-SRR-

BERT across splits: on validation by +0.76 (CheXagent-3B), +3.82 (MedGemma-4B), and +5.18 (Lingshu-7B); on test by +1.26 (CheXagent-3B), +4.65 (MedGemma-4B), and +6.09 (Lingshu-7B); and on test-reviewed by +2.67 (Lingshu-7B) and +3.28 (MedGemma-4B). While only one exception exists (CheXagent-3B on test-reviewed split showing -1.05 degradation), it demonstrates that clinical context enhances report generation quality even for models not explicitly trained to utilize such information. This zero-shot improvement provides additional evidence that clinical context carries inherent diagnostic value that medical MLLMs can leverage through their general pretraining.

**Necessity of Instruction Fine-Tuning.** These results demonstrate the necessity of instruction fine-tuning for the SRRG task. While pretrained models possess general medical knowledge and basic report generation capabilities, they lack the task-specific understanding required for high-quality structured report generation. For this reason, our evaluation focuses on comparing fine-tuned models, examining whether they are trained with or without clinical context to isolate the effect of comprehensive clinical information on structured report generation.

## F    DETAILED DATASET STATISTICS

We provide detailed statistics of our clinical context chest X-ray dataset, focusing on patient distribution across splits.

**Patient Distribution Across Splits.** Our dataset maintains strict patient-level separation across training, validation, and test splits to prevent data leakage. As shown in the patient overlap heatmaps, the training set contains 83,147 unique patients for findings and 125,947 unique patients for impression tasks. The validation sets include 434 patients for findings and 477 patients for impression, while the test sets contain 274 patients for findings and 423 patients for impression. The test-reviewed splits comprise 173 patients for findings and 172 patients for impression, with 106 and 108 patients respectively shared with the test split. This patient-level split ensures that clinical studies from the same patient do not appear across different evaluation splits, with zero patient overlap between training and evaluation sets. The distribution maintains clinical diversity while preserving the integrity of comprehensive clinical contexts within patient histories.

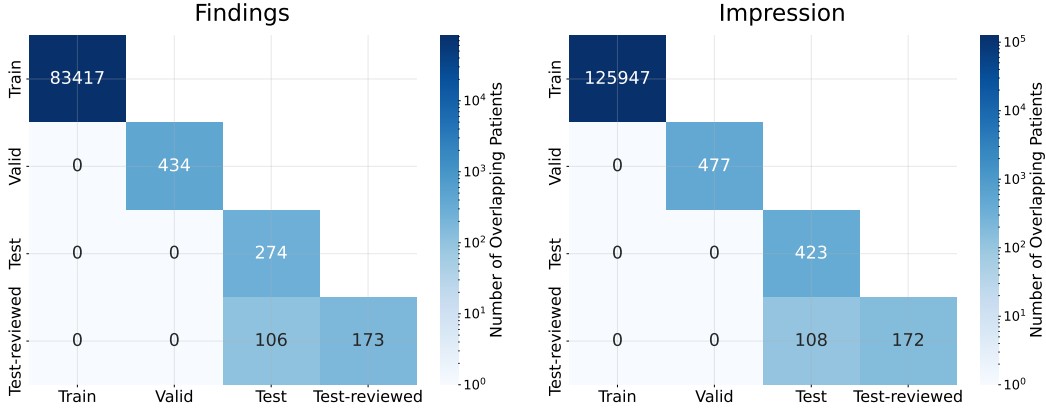

Figure 10: **Patient overlap heatmaps** across train, valid, test, and test-reviewed splits.

## G    PROMPT DESIGNS

In this section, beyond formats for impression with prior studies (Fig. 7), current study (Fig. 8), we provide examples of our design choices for prompts, *e.g.*, formats for findings with/without prior studies (Figs. 11 and 12), impression without prior studies (Fig. 13), prior studies (Fig. 14), and response format for findings and impression (Figs. 9 and 15), used for training and evaluation.

> **SYSTEM PROMPT:**
> You are an expert radiologist.
>
> **USER PROMPT:**
> Analyze the current chest X-ray images and compare them with the previous studies to write the FIND-INGS section of a radiology report. Use standard medical terminology and note any changes from the prior studies, focusing on the most recent comparisons. Consider the available clinical contexts when formulating your findings.

Figure 11: **Prompt format for FINDINGS** for C-SRRG.

> **SYSTEM PROMPT:**
> You are an expert radiologist.
>
> **USER PROMPT:**
> Analyze the chest X-ray images and write the FINDINGS section of a radiology report. Use standard medical terminology and organize findings by anatomical regions. Consider the available clinical contexts when formulating your findings.

Figure 12: **Prompt format for FINDINGS without previous history** for C-SRRG.

> **SYSTEM PROMPT:**
> You are an expert radiologist.
>
> **USER PROMPT:**
> Analyze the chest X-ray images and write the IMPRESSION section of a radiology report. Provide a concise clinical summary and diagnosis based on the imaging findings. Consider the available clinical contexts when formulating your impression.

Figure 13: **Prompt format for IMPRESSION without previous history** for C-SRRG.

## H    INSTRUCTION TUNING DATASET PROMPT EXAMPLE

We provide detailed instruction fine-tuning examples that showcase the comprehensive clinical context utilized in our approach. These examples demonstrate how all available clinical information is systematically integrated into our instruction tuning dataset, including patient medical history, imaging techniques, previous study findings, and temporal comparisons. The following multi-part examples illustrate the complete structure of our training data, highlighting how comprehensive clinical contexts including temporal, multi-view, and metadata information are preserved and leveraged for clinical reasoning in radiology report generation.

### H.1    FINDINGS GENERATION EXAMPLE

The first example demonstrates the generation of the FINDINGS section, which requires detailed anatomical observation and temporal comparison across multiple studies (Figs. 16 to 18):

### H.2    IMPRESSION GENERATION EXAMPLE

The second example demonstrates the generation of the IMPRESSION section, which requires clinical synthesis and diagnostic reasoning (Figs. 19 to 21):

## I    HALLUCINATION ANALYSIS

In this section, we examine a critical limitation of radiology report generation models trained without clinical context, specifically their tendency to hallucinate temporal comparisons when referencing non-existent prior studies. We first demonstrate that dataset ground truth reports contain temporal statements that become hallucinations when clinical context is absent, as radiologists naturally write these comparisons when they have access to prior studies. We then analyze how models trained

```
=== PREVIOUS STUDY N ===
Previous Clinical History/Indication:
Status post coronary artery bypass grafting (CABG), post chest tube removal.

Previous Technique:
Standard posteroanterior and lateral chest radiographs.

Previous Findings:
Lungs and Airways:
- No pneumothorax visualized
- Low lung volumes
- Increased infiltrate in the left upper lung

Pleura:
- Retrocardiac opacity
- Layering left pleural effusion

Cardiovascular:
- Vascular plethora

Other:
- Patchy areas of alveolar edema on the right
- Endotracheal (ET) tube positioned 4.3 cm above the carina
- Nasogastric (NG) tube within the stomach

Previous Impression:
1. Worsened infiltrate in the left upper lung suggestive of an acute process.
2. Presence of retrocardiac opacity and layering left pleural effusion.
3. Vascular plethora and patchy alveolar edema on the right, which may indicate fluid overload or cardiac
etiology.
4. Proper positioning of the ET and NG tubes.

Previous Images:
[Images:  Previous chest X-ray images]
```

Figure 14: **An example of prior study** for C-SRRG.

```
ASSISTANT RESPONSE:
FINDINGS:
Lungs and Airways:
- No pneumothorax or focal consolidation.

Pleura:
- Enlarged small right pleural effusion.

Cardiovascular:
- Mildly enlarged heart.

Hila and Mediastinum:
- Normal hilar and mediastinal contours.

Tubes, Catheters, and Support Devices:
- Right internal jugular (IJ) catheter terminates at the superior cavoatrial junction.
```

Figure 15: **An example of ground-truth assistant responses** in the C-SRRG-Findings dataset.

without such clinical context systematically produce these hallucinations, even for patients with no imaging history. Finally, we quantify these hallucinations by detecting the frequency of temporal statements on the generated reports on the evaluation set without clinical context.

**Dataset Hallucination.**  Ground truth radiology reports in clinical datasets frequently contain temporal statements such as "new from prior exam," "unchanged," or "stable compared to previous study." These temporal references are clinically appropriate when radiologists have access to prior imaging studies for comparison. However, when language models are trained on these reports without access to the corresponding clinical context and prior studies, they learn to replicate these temporal language patterns indiscriminately. This training paradigm creates a systematic hallucination problem where models generate temporal comparison statements even for patients with no prior imaging history. The following examples demonstrate these temporal hallucinations present

**Findings Example - Part 1: Current Study Context**

**USER PROMPT:**
Analyze the current chest X-ray images and compare them with the previous studies to write the FIND-INGS section of a radiology report. Use standard medical terminology and note any changes from the prior studies, focusing on the most recent comparisons. Consider the available clinical contexts when formulating your findings.

**=== CURRENT CLINICAL HISTORY/INDICATION ===**
Evaluation for fluid overload.

**=== CURRENT TECHNIQUE ===**
Standard frontal chest radiography protocol.

**=== CURRENT COMPARISON ===**
Prior radiographs and CT scans.

**=== CURRENT IMAGES ===**
[Images: Current chest X-ray images]

Figure 16: **Findings generation example (Part 1)** in the C-SRRG-Findings dataset.

**Findings Example - Part 2: Previous Study 1**

**=== PREVIOUS STUDY 1 (Most Recent) ===**
Previous Clinical History/Indication:
Patient with a history of multifocal after CABG, currently presenting with symptoms suggestive of CHF or pneumonia.

Previous Technique:
A single frontal chest radiograph was obtained.

Previous Comparison:
Multiple prior radiographs

Previous Findings:
Lungs and Airways:
- No definitive consolidation observed on this examination; however, subsequent CT confirms presence at the right base
- Mild pulmonary edema

Pleura:
- Moderate right pleural effusion, unchanged
- No pneumothorax

Cardiovascular:
- Moderate cardiomegaly noted
- Aortic tortuosity present

Tubes, Catheters, and Support Devices:
- Status post median sternotomy with CABG and valve replacements

Previous Impression:
1. Mild pulmonary edema with right pleural effusion.
2. Right base consolidation not definitive on radiograph but confirmed on subsequent CT.

Previous Images:
[Images: Previous chest X-ray images]

Figure 17: **Findings generation example (Part 2)** in the C-SRRG-Findings dataset.

in ground truth radiology reports from the dataset, showing how temporal comparison statements appear without proper clinical context (Figs. 22 to 24):

**Trained Model Hallucination.** Our trained models also exhibit hallucination patterns, particularly when generating temporal comparisons despite having access only to the current study. The following examples from MedGemma-4B model trained without clinical history demonstrate how

**Findings Example - Part 3: Previous Study 2 & Response**

**=== PREVIOUS STUDY 2 ===**
Previous Clinical History/Indication:
The patient presents with tachycardia.

Previous Technique:
Posteroanterior (PA) and lateral chest radiographs.

Previous Findings:
Cardiovascular:
- Status post sternotomy with mitral and aortic valve replacements
- Moderate cardiomegaly, unchanged

Hila and Mediastinum:
- Stable mediastinal and hilar contours

Lungs and Airways:
- New mild-to-moderate elevation of the right hemidiaphragm suggesting volume loss
- Increased patchy opacity in the right lower lobe, concerning for pneumonic consolidation
- Probable new opacification in the right middle lobe
- New vague opacity in the right suprahilar region of the right upper lobe, potentially indicating early pneumonia

Pleura:
- No definite pleural effusion

Previous Impression:
1. Findings suggest multifocal pneumonia involving the right lung.

**ASSISTANT RESPONSE:**
FINDINGS:
Lungs and Airways:
- No pneumothorax or focal consolidation.

Pleura:
- Enlarged small right pleural effusion.

Cardiovascular:
- Mildly enlarged heart.

Hila and Mediastinum:
- Normal hilar and mediastinal contours.

Tubes, Catheters, and Support Devices:
- Right internal jugular (IJ) catheter terminates at the superior cavoatrial junction.

Figure 18: **Findings generation example (Part 3)** in the C-SRRG-Findings dataset.

models can inappropriately generate comparative language even when no prior studies are available for comparison (Figs. 25 to 30):

### Example Case with Clinical Context - Part 1: Current Study

**USER PROMPT:**
Analyze the current chest X-ray images and compare them with the previous studies to write the IM-PRESSION section of a radiology report. Provide a concise clinical summary and diagnosis, noting any changes from the prior studies, focusing on the most recent comparisons. Consider the available clinical contexts when formulating your impression.

**=== CURRENT CLINICAL HISTORY/INDICATION ===**
Male with end-stage renal disease on hemodialysis, multiple orthopedic hardware, bioprosthetic aortic valve replacement, coronary artery disease status post coronary artery bypass grafting, permanent pacemaker for sick sinus syndrome, admitted for MRSA bacteremia, transferred to the critical care unit for hypotension during anesthesia induction. Patient intubated for respiratory status assessment.

**=== CURRENT TECHNIQUE ===**
Chest single view

**=== CURRENT COMPARISON ===**
Prior imaging at an unspecified time.

**=== CURRENT IMAGES ===**
```
[Images:  Current chest X-ray images]
```

Figure 19: **Impression generation example (Part 1)** in the C-SRRG-Impression dataset.

### Example Case with Clinical Context - Part 2: Previous Study 1

**=== PREVIOUS STUDY 1 (Most Recent) ===**
Previous Clinical History/Indication:
Status post coronary artery bypass grafting (CABG), post chest tube removal.

Previous Technique:
Standard posteroanterior and lateral chest radiographs.

Previous Findings:
Lungs and Airways:
- No pneumothorax visualized
- Low lung volumes
- Increased infiltrate in the left upper lung

Pleura:
- Retrocardiac opacity
- Layering left pleural effusion

Cardiovascular:
- Vascular plethora

Other:
- Patchy areas of alveolar edema on the right
- Endotracheal (ET) tube positioned 4.3 cm above the carina
- Nasogastric (NG) tube within the stomach

Previous Impression:
1. Worsened infiltrate in the left upper lung suggestive of an acute process.
2. Presence of retrocardiac opacity and layering left pleural effusion.
3. Vascular plethora and patchy alveolar edema on the right, which may indicate fluid overload or cardiac etiology.
4. Proper positioning of the ET and NG tubes.

Previous Images:
```
[Images:  Previous chest X-ray images]
```

Figure 20: **Impression generation example (Part 2)** in the C-SRRG-Impression dataset.

**Example Case with Clinical Context - Part 3: Previous Study 2 & Response**

**=== PREVIOUS STUDY 2 ===**
Previous Clinical History/Indication:
Patient presents with portions and exertion x.

Previous Technique:
The examination was performed using frontal and lateral views of the chest.

Previous Findings:
Tubes, Catheters, and Support Devices:
- Dual lead left-sided pacemaker extending to the expected positions of the right atrium and right ventricle.

Pleura:
- Slight blunting of the posterior costophrenic angles, which may suggest very trace pleural effusions.

Lungs and Airways:
- No focal consolidation observed.

Cardiovascular:
- The cardiac silhouette appears stable.

Hila and Mediastinum:
- The mediastinal silhouette is stable.

Musculoskeletal and Chest Wall:
- Right proximal humerus hardware is present but not well evaluated due to the imaging technique.

Other:
- Slight prominence of the interstitium, which may indicate minimal interstitial edema.

Previous Impression:
1. Possible very trace pleural effusions.
2. Minimal interstitial edema.
3. Stable cardiac and mediastinal silhouettes.
4. Presence of dual lead left-sided pacemaker in expected positions.
5. Right proximal humerus hardware present but not adequately assessed.

**ASSISTANT RESPONSE:**
IMPRESSION:
1. Worsened left basilar opacity, which may represent atelectasis or pneumonitis in the appropriate clinical context.
2. Mild improvement in pulmonary vascularity.

Figure 21: **Impression generation example (Part 3)** in the C-SRRG-Impression dataset.

### Dataset Hallucination Example 1: Temporal Information Fabrication

**Structured Report:**

Exam Type: Chest radiograph.

Technique: Portable anteroposterior (AP) chest radiography was performed.

History: A male patient with hep C cirrhosis and large right pleural effusion status post thoracocentesis. Evaluate for resolution of pleural effusion.

Comparison: Prior portable AP chest radiograph

Findings:

Lungs and Airways:
- Mild inflation of the right upper lobe
- Collapsed right lower lobe
- No consolidation in the left lung

Pleura:
- Moderate pleural effusion within the right pleural space.
- Moderate right pneumothorax, new from prior exam.
- No left pleural effusion or pneumothorax.

Cardiovascular:
- No significant mediastinal shift observed.

Hila and Mediastinum:
- Mediastinum appears unremarkable

Impression:
1. Moderate right-sided pneumothorax.
2. Moderate right pleural effusion.
3. Inflation of the right upper lobe with collapse of the right lower lobe.
4. No mediastinal shift.

> **Hallucination:** The phrase "**new from prior exam**" represents temporal information that cannot be verified from the current study alone, if not with previous history.

Figure 22: **Dataset hallucination example 1** in SRRG dataset.

---

**Dataset Hallucination Example 2: Stability Assumption Without Comparison**

**Structured Report:**
Exam Type: Chest radiograph

Technique: Standard frontal and lateral chest radiographic views were performed.

History: Atrial fibrillation (AF), coronary artery disease (CAD), congestive heart failure (CHF).

Comparison: Prior chest radiographs

Findings:

Cardiovascular:
- Mild to moderate cardiomegaly, unchanged.
- Tortuous but stable aorta.

Tubes, Catheters, and Support Devices:
- Transvenous pacemaker/AICD with leads terminating in the right atrium and right ventricle.
- Median sternotomy wires are aligned and intact.

Lungs and Airways:
- Lungs are clear with no evidence of consolidation, pleural effusion, pneumothorax, or overt pulmonary edema.

Impression:
1. No radiographic evidence for acute cardiopulmonary process.

> **Hallucination:** The term "**unchanged**" implies comparison with prior studies, which is problematic without previous history.

Figure 23: **Dataset hallucination example 2** in SRRG dataset.

---

**Dataset Hallucination Example 3: Generic Temporal Statement**

**Structured Report:**

Exam Type: Chest radiograph

Technique: Single AP upright portable chest radiograph.

History: Shortness of breath.

Comparison: Prior chest radiograph

Findings:

Lungs and Airways:
- Low lung volumes
- Minimal bibasilar atelectasis
- Subcentimeter left lower lung rounded calcification, stable, possibly representing a calcified granuloma
- No focal consolidation
- No overt pulmonary edema

Pleura:
- No pleural effusion
- No evidence of pneumothorax

Cardiovascular:
- Stable cardiac and mediastinal silhouettes

Hila and Mediastinum:
- Ovoid calcification over the left mediastinum, stable

Tubes, Catheters, and Support Devices:
- Right-sided Port-A-Cath in place, terminating at the cavoatrial junction, without significant change in position

Musculoskeletal and Chest Wall:
- Status post median sternotomy

Impression:

1. No significant interval change.

> **Hallucination:** The phrase "**No significant interval change**" is a generic temporal statement that, if not with prior studies, represents a non-specific conclusion that could mask important clinical details.

Figure 24: **Dataset hallucination example 3** in SRRG dataset.

---

**Trained Model Hallucination Example 1: Temporal Comparisons in Findings**

**Sample ID:** 12
**Generated Report:**
FINDINGS:
Tubes, Catheters, and Support Devices:
- Endotracheal tube terminates 4.5 cm above the carina.
- Right internal jugular central venous catheter terminates in the right atrium.
- Left pectoral pacemaker with leads in the right atrium and right ventricle.
- Median sternotomy wires are intact.

Cardiovascular:
- Moderate cardiomegaly, unchanged.

Lungs and Airways:
- Low lung volumes.
- No evidence of pneumothorax.

Pleura:
- Small left pleural effusion, unchanged.

Other:
- Retrocardiac opacity, likely representing atelectasis, unchanged.

> **Hallucination:** Multiple instances of "**unchanged**" indicate temporal comparisons when only analyzing a single study.

Figure 25: **Trained model hallucination example 1 on test-reviewed split** using MedGemma-4B.

---

**Trained Model Hallucination Example 2: Interval and Persistent Language**

**Sample ID:** 60
**Generated Report:**
FINDINGS:
Pleura:
- Interval placement of a right-sided pigtail catheter.
- Decreased size of the right pleural effusion.
- No evidence of pneumothorax.

Lungs and Airways:
- Persistent right basilar opacity, likely representing atelectasis.
- Left lung is clear.

Cardiovascular:
- Stable cardiomediastinal silhouette.

> **Hallucination:** Terms like "**Interval placement**", "**Decreased size**", and "**Persistent**" all imply temporal comparisons.

Figure 26: **Trained model hallucination example 2 on test-reviewed split** using MedGemma-4B.

**Trained Model Hallucination Example 3: Explicit Temporal Language**

**Sample ID:** 37
**Generated Report:**
FINDINGS:
Lungs and Airways:
- No significant interval change in the appearance of the lungs.

Pleura:
- Small right pleural effusion, unchanged.

Cardiovascular:
- Stable cardiomediastinal silhouette.

Tubes, Catheters, and Support Devices:
- Tracheostomy tube in place.
- Right internal jugular (IJ) dialysis catheter in place.
- Vascular stent in the right subclavian region.

Other:
- No evidence of pneumothorax.

> **Hallucination:** Explicit temporal language ("**No significant interval change**", "**unchanged**") when no prior studies are available.

Figure 27: **Trained model hallucination example 3 on test-reviewed split** using MedGemma-4B.

**Trained Model Hallucination Example 4: Unchanged Position in Impression**

**Sample ID:** 0
**Generated Report:**
IMPRESSION:
1. No evidence of pneumothorax.
2. Stable small bilateral pleural effusions.
3. Unchanged position of the right chest tube.
4. Normal cardiomediastinal silhouette.
5. Presence of a vagal nerve stimulator.

> **Hallucination:** The term "**Unchanged position**" suggests comparison with prior imaging that is not available.

Figure 28: **Trained model hallucination example 4 on test-reviewed split** using MedGemma-4B.

**Trained Model Hallucination Example 5: Interval and Persistent Terms**

**Sample ID:** 36
**Generated Report:**
IMPRESSION:
1. Interval increase in bilateral pleural effusions.

2. Persistent left lower lobe atelectasis.
3. Mild pulmonary edema.
4. Stable cardiomegaly.

> **Hallucination:** Terms "**Interval increase**" and "**Persistent**" indicate temporal comparisons without access to prior studies.

Figure 29: **Trained model hallucination example 5 on test-reviewed split** using MedGemma-4B.

**Trained Model Hallucination Example 6: New vs Unchanged Terminology**

**Sample ID:** 40
**Generated Report:**
IMPRESSION:
1. New right lower lobe opacity , which may represent pneumonia or aspiration.

2. Small right pleural effusion.
3. Stable mild cardiomegaly.

4. Unchanged position of the right-sided central venous catheter.

> **Hallucination:** The term "**Unchanged position**" implies knowledge of prior imaging studies that the model does not have access to. **New** also suggests temporal comparison but may be acceptable if referring to acute findings.

Figure 30: **Trained model hallucination example 6 on test-reviewed split** using MedGemma-4B.

# J    THE USE OF LLMS

We used LLMs solely for light editing such as correcting grammatical errors and polishing some words. They did not contribute to research ideation, experiments, analysis, or substantive writing.

