# OpenReview forum: "Automated Structured Radiology Report Generation with Rich Clinical Context"
_ICLR.cc/2026/Conference — Submitted to ICLR 2026_

### Official Review · Reviewer_4zws · 2025-10-29

**Soundness:** 3
**Presentation:** 3
**Contribution:** 2
**Rating:** 4
**Confidence:** 4

**Summary:**

The paper focuses on how clinical context (such as patient history, imaging indication, technique, and prior studies) affect the performance of radiology report generation. The key contribution is a curated dataset that include several types of context. This paper conduct extensive experiments to benchmark the performance of several MLLMs and analyze how including context affect the performance of radiology report generation.

**Strengths:**

1. Considering context information for radiology report generation is a reasonable technical design that worth exploring.
2. Compared to previous work, this paper conducts a very comprehensive empirical analysis on how context information affect radiology report generation performance.

**Weaknesses:**

1. The contribution of the newly curated datasets are limited.

1.1 The dataset in this paper is primarily based on an existing resource, with only minor modifications. Most samples are drawn from MIMIC-CXR, which already contains all the context information considered by the authors. As a result, the additional effort required to construct this dataset is minimal, and the contribution in terms of dataset creation is limited.

1.2 It is not a novel idea to consider context information in radiology report generation. Previous works have already extensively studied the influence of the context information considered in this paper. Or maybe you can provide some experiments on how these methods perform on the proposed benchmark datasets.

1.2 There are lots of other types of context information that haven't been extensively studies by previous works that can potentially benefit report generation. For example, results of other exams or lab tests, patient's demographic information, patient's EHR. Lots of the information is actually also included in MIMIC datasets, but the authors missed them.

2. For part of the samples in the new dataset, the context information is generated by GPT-4. I think there should be more detailed analysis on the quality of the generated data.
2.1 what prompts are used to generated the synthetic samples?

2.2 How the authors make sure the generated context is aligned with the content of the target radiology study  and will there be contradiction between them.

2.3 What are the quality control measures?

2.4 Can you provide some evaluation on the quality of the generated data?

**Questions:**

Please address my concerns in the Weaknesses.

---

> ### Author Response · Authors · 2025-11-23
> **Response to Reviewer 4zws (1/6)**
>
> We sincerly appreciate you for your time and constructive comments which improve our paper. We respectfully address your concerns as following:
>
> >**[Q1-1]** The dataset in this paper is primarily based on an existing resource, with only minor modifications. Most samples are drawn from MIMIC-CXR, which already contains all the context information considered by the authors.
>
> - We sincerely appreciate the reviewer for raising this important concern and also agree that the additional effort for dataset curation may appear minimal when viewed solely from a data processing perspective, as MIMIC-CXR already contains much of the contextual information we utilize.
>
> - We first would like to clarify that our benchmark integrates both MIMIC-CXR [1] and CheXpert Plus [2].
>
>     - For findings generation: **126,955 samples (68.8%)** from MIMIC-CXR and **57,587 samples (31.2%)** from CheXpert Plus, totaling **184,542**.
>     - For impressions generation: **188,455 samples (46.0%)** from MIMIC-CXR and **221,472 samples (54.0%)** from CheXpert Plus, totaling **409,927**.
>
> - We emphasize that CheXpert Plus [2] was introduced recently and takes a significant portion of our dataset (**31.2~54.0%** depending on the task).
> - More crucially, even though both MIMIC-CXR [1] and CheXpert Plus [2] contain **rich clinical context information**, the usage of such contexts for report generation remains **largely underexplored in the literature** as we elaborate in following response to **[Q1-2]**.
>
> ---
>
> >**[Q1-2]** As a result, the additional effort required to construct this dataset is minimal, and the contribution in terms of dataset creation is limited.
>
> - We sincerely appreciate the reviewer for this valuable feedback regarding the contribution. We would first like to clarify that we **do not claim novelty** in introducing a **fundamentally new dataset**.
>
> - Rather, we respectfully believe that as noted in `L104-115`, our primary contribution lies in **identifying a key limitation of SRRG**, **effciently curating a dataset with essential clinical context**, and **providing empirical evidence** through benchmarking to facilitate future work on SRRG for practitioners:
>
>     - Our work systematically analyzes **the clinical workflow of radiologists** to identify relevant context that are routinely consulted during report writing, and demonstrates empirically that omitting such context leads to **systematic issues like temporal hallucinations** as depicted in `L187-198` in `§3.1`.
>
>     - A key distinction from prior work is the **comprehensive incorporation of multiple clinical contexts simultaneously**, multi-view chest radiographs, imaging technique specifications, clinical indications, and comparison with prior studies, rather than limiting analysis to isolated contextual elements as depicted in `L200-215` in `§3.1`.
>
>     - To the best of our knowledge, although the required effort may appear minimal, this study is the **first to curate a large-scale structured radiology report generation dataset** that aligns diverse clinical contexts with structured radiology reports, enabling a systematic investigation of their effects as shown in `Tabs. 5, 6, 7, and 8`.
>
>     - Furthermore, we provide the **first comprehensive empirical study** examining how these clinical contexts influence structured report generation across **multiple state-of-the-art medical MLLMs (e.g., CheXagent-3B, MedGemma-4B, Lingshu-7B)**, revealing consistent improvements (**+2–7% F1-SRR-BERT**) and substantial reductions in temporal hallucinations (**12–18%**).
>
>
> - Notably, investigating the **effect of clinical contexts in medical MLLMs remains largely underexplored**, despite **the significant importance of report writing in the clinical workflow** described in `L43-47` (i.e. report written by radiologists is read by clinicians to do desicion-making on the patient).
> - This is partly due to the limited capability MLLMs in handling long context, as evidenced by the difficulties of CheXagent-3B with comprehensive contextual information (`L331-333`, `Appendix A.2, A.3`, and `Appendix C`).
>
> - We believe that the above discussion clarifies that we do **not** claim to introduce a new dataset, and it also clarifies the contributions presented in `L104–115`.
>
>
> ---

---

> ### Author Response · Authors · 2025-11-23
> **Response to Reviewer 4zws (2/6)**
>
> ---
>
> >**[Q2-1]** It is not a novel idea to consider context information in radiology report generation. Previous works have already extensively studied the influence of the context information considered in this paper.
>
> - We sincerely agree that incorporating clinical context into radiology report generation is not a novel concept, as prior works have explored various contextual elements. However, as outlined in `§2`, we respectfully emphasize **three key distinctions**: 1) the use of **full clinical context**, 2) the focus on **structured report generation**, and 3) the investigation of **medical MLLMs using comprehensive clinical contexts**.
>
>
> - First, prior work has only considered **partial contexts** in isolation or limited combinations:
>     - For example, [5, 6] incorporate multi-view images, [7] focus on indication, and [9] explore longitudinal data.
>     - Even recent attempts to combine multiple contexts (e.g., MLRG [10], PriorRG [11], MAIRA-2 [12]) face **fundamental constraints**: **fixed input structures** (e.g., MAIRA-2 restricted to fixed 3 images; current frontal, current lateral, and prior frontal image), **limited temporal scope** (only immediate previous study), and **complex multi-stage training restricting input flexibility** [10, 11].
>     - Empirically, the comprehensive ablation studies in `Tabs. 3, 4, 5, 6, 7, and 8` systematically demonstrate **complementary benefits** of each context component, which has **not been thoroughly investigated**.
>
> - Secondly, all previous work has focused exclusively on **free-form report generation**:
>     - This overlooks structured reports, despite their clinical importance for clarity, consistency, and adherence to reporting standards [3, 4, 13].
>     - The structured report generation task, recently introduced in prior work [13], has **never been studied** in the context of clinical contextualization.
>
> - Lastly, to our knowledge, previous works have not systematically investigated how **recent medical MLLMs leverage comprehensive clinical contexts** in practice:
>     - Our work examines how **MLLMs perform when provided with the full range of clinical information** available in real radiological workflows—including multi-view images, indication text, technique specifications, and multiple prior studies with reports.
>     - Our comprehensive evaluation shows that **clinical contextualization in structured report generation** leads to **substantial improvements (2–7% F1-SRR-BERT)** and mitigates **systemtatic errors** such as temporal hallucinations (**12–18% reduction**), offering important insights for context-aware structured reporting that reflects actual clinical practice.
>
>
> ---
>
> >**[Q2-2]** Or maybe you can provide some experiments on how these methods perform on the proposed benchmark datasets.
>
> - We sincerely appreciate the reviewer for this constructive suggestion. Consistent to our response to **[Q2-1]**, we respectfully note that **existing methods can not be evaluated** on our proposed benchmark.
> - Prior works [5, 6, 7, 8, 9, 10, 11] employ **specialized architectural components**:
>
>     - Multi-view fusion with cross-view consistency [5]
>     - Multi-positive contrastive learning with knowledge-guided generation [6]
>     - Posterior/prior knowledge exploration with medical knowledge graphs [8]
>     - Graph-based disease progression reasoning [7]
>     - Group causal transformers for longitudinal aggregation [9]
>     - Multi-stage contrastive learning [10, 11]
> - The above components are **specifically designed for their respective context integration approaches**. These dedicated architectural modifications are fundamentally **incompatible with recent medical MLLMs**, therefore, cannot be directly applied to our study on **recent medical MLLMs with our proposed benchmark datasets**.
>
> - Thus, **only recent medical MLLMs** (e.g., CheXagent-3B [20], MedGemma-4B [21], Lingshu-7B [22]) can be **seamlessly evaluated and benchmarked** on the proposed datasets, thanks to their flexibility in handling multimodal inputs (e.g., images and text).
>
> - We believe the above discussion, together with our response to **[Q2-1]**, clarifies the key distinctions of our work as outlined in `§2` and detailed in `Appendix B`.
>
> ---

---

> ### Author Response · Authors · 2025-11-23
> **Response to Reviewer 4zws (3/6)**
>
> ---
>
> >**[Q3]** There are lots of other types of context information that haven't been extensively studies by previous works that can potentially benefit report generation. For example, results of other exams or lab tests, patient's demographic information, patient's EHR. Lots of the information is actually also included in MIMIC datasets, but the authors missed them.
>
> - We sincerely appreciate your point that **additional patient information** such as lab tests, demographics, and EHR records could potentially benefit report generation.
>
> - We would first like to clarify our **design choice for clinical context**. We identify the core clinical contexts (multi-view images, indication, technique, and prior studies) based on **previously established studies**:
>
>     - Prior works demonstrate that providing **additional clinical information** enhances interpretation accuracy and reporting confidence [14], with **best practice guidelines** [3, 15] establishing indication, technique, and comparison as standard components of high-quality radiology reports.
>
>     - The **structured radiology reporting framework** from [13] explicitly defines these elements (Exam Type, History, Technique, Comparison, Findings, Impression) as standard sections, reflecting **clinical practice**.
>
>     - More crucially, `Tabs. 3, 4, 5, 6, 7, and 8` empirically show the effectiveness the chosen **imaging-centric clinical contexts** and `Tab. 9` demonstrates their substantial impact on the reduction of **temporal hallucinations**.
>
> - However, we sincerely agree that **incorporating additional patient information** could provide complementary value:
>
>     - We have already discussed that extending beyond imaging-specific contexts to comprehensive patient data represents an **important future direction** as described in `Appendix A.3`.
>
>     - However, integrating such diverse sources introduces **additional complexity**: data heterogeneity, privacy considerations, availability across clinical settings, and computational requirements.
>
>     - Moreover, incorporating all these possible contexts (lab results, demographics, EHR records, and extensive longitudinal imaging) may exceed the **practical capacity of current medical MLLMs**. For example, some patients have more than **60-100 longitudinal studies** as shown in `Fig. 6`, and integrating such temporal depth alongside other patient information would require processing substantially longer sequences.
>
>     - Future extensions could integrate additional patient information through two complementary approaches: (1) advances in efficient attention mechanisms [16, 17] that enable processing longer temporal contexts and richer patient data, and (2) retrieval-augmented generation (RAG) methods [18, 19] that intelligently select the most informative contextual information based on clinical relevance, potentially integrated within agentic systems for dynamic context selection.
>
> - In summary, while the chosen clinical contexts are **motivated by established studies** and **supported by our empirical observations**, we believe that investigating the effectiveness of additional patient information remains an **important direction for future work**.
>
> - We have clarified the above discussion on top of `Appendix A.3`.
>
> ---

---

> ### Author Response · Authors · 2025-11-23
> **Response to Reviewer 4zws (4/6)**
>
> ---
>
> >**[Q4-1]** For part of the samples in the new dataset, the context information is generated by GPT-4. I think there should be more detailed analysis on the quality of the generated data.
>
> - We sincerely appreciate the reviewer for raising this important concern about the quality of GPT-4–generated context.
>
> - Our dataset builds on the structured report generation framework [13], which was **explicitly validated** through a rigorous reader study with **five board-certified radiologists**, and we rely on this clinically validated process as the primary quality control for the GPT-4–generated structured reports used in our benchmark.
>
> - Specifically, [13] asked the five radiologists to review **464 GPT-4–generated structured reports** (233 findings, 231 impressions) sampled from the MIMIC-CXR test set and CheXpert Plus validation set, edit them for any clinical inaccuracies, and quantify the edits using Python's `difflib.SequenceMatcher` to compute word-level insertions, deletions, and replacements.
> - Their analysis reports **average similarity ratios**, which are computed as (2 × matches / total tokens in original and edited texts).
>
> - This evaluation showed that GPT-4 maintains **high clinical fidelity**:
>     - For **impression sections**, **55.79%** of reports required edits, with an average similarity ratio of **0.77**, indicating that most changes were structural refinements rather than fundamental diagnostic corrections.
>     - For **findings sections**, **70.39%** of reports required edits, with an even higher average similarity ratio of **0.88**, suggesting strong preservation of clinical content.
>     - At the utterance level, **1,609 utterances** were reviewed for disease labels, achieving a **72% exact match rate** and **74% average Jaccard similarity** between GPT consensus labels and radiologist-reviewed labels.
> - The radiologist-edited subset defines the `test_reviewed` split, which we adopt as a **gold-standard evaluation set**, and the prompting pipeline in [13] is designed to **preserve semantic content** by restructuring existing free-form reports into standardized sections without adding new findings or extrapolating beyond the original study.
>
> - Taken together, the prior reader study and our empirical results suggest that the GPT-4–generated structured context achieves **clinically reliable quality** for both training and evaluation.
>
> - We have clarified these quality control and evaluation details at `L1052–1075` of `Appendix A.1` in the revised manuscript.
>
> ---
>
> >**[Q4-2]** what prompts are used to generated the synthetic samples?
>
> - We sincerely appreciate the reviewer for this valuable question. The structured radiology reports were generated by prior work [13] using GPT-4 Turbo 1106 Preview via Azure services:
>
>     - First of all, we kindly note that the comprehensive prompt documentation is provided in prior work [13].
>     - Specifically, **Structuring Prompt** (Prompt 2 in [13]) converts free-form radiology reports into standardized structured format following strict desiderata.
>     - It **enforces specific sections** (Exam Type, History, Technique, Comparison, Findings, Impression) with **anatomical headers** (Lungs and Airways, Pleura, Cardiovascular, Hila and Mediastinum, Tubes/Catheters/Support Devices, Musculoskeletal and Chest Wall, Abdominal, Other).
>     - Lastly, the prompt explicitly instructs the model to **exclude identifiers** (dates, names, institutions) while retaining patient sex/age, and to strictly adhere to current examination results without extrapolating beyond the original report.
>
> - These prompts ensure systematic conversion while maintaining **clinical fidelity**, as validated through their reader study with five board-certified radiologists (`Appendix B` in [13]).
> - We refer readers to the full prompt text provided in their appendix.
>
> ---

---

> ### Author Response · Authors · 2025-11-23
> **Response to Reviewer 4zws (5/6)**
>
> ---
>
> >**[Q5]** How the authors make sure the generated context is aligned with the content of the target radiology study and will there be contradiction between them.
>
> - Consistent to our response to **[Q4-1]**, we kindly note the structured report generation process from prior work [13] is specifically designed to **maintain semantic alignment** with original free-form reports:
>
>     - The GPT-4-based process converts free-form reports into structured format while **preserving clinical content**, reorganizing information under standardized anatomical section headers without adding or extrapolating beyond the original content.
>
>     - The extraction operates on the report of the same study, ensuring all clinical context elements are **derived from and aligned with** the target examination.
>
>     - As detailed in **[Q4-1]**, this alignment has been **rigorously validated** through a reader study in prior work [13] with **five board-certified radiologists**, showing **high similarity ratios** between GPT-4–generated and radiologist-edited reports (0.77 for impressions and 0.88 for findings).
>
> - Moreover, several aspects of our evaluation protocol further ensure that contradictions between generated context and target studies remain **minimal**:
>
>     - Our evaluation metrics (**F1-SRR-BERT**) measure **semantic similarity** using contextual embeddings rather than exact text matching, which naturally accounts for paraphrasing or minor variations while capturing clinical content alignment.
>
>     - With **large-scale training data** containing thousands of reports, models learn aggregate statistical patterns rather than memorizing individual examples, allowing random noise to be averaged out during training.
>
>     - Consistent performance trends across **three different model architectures** with varying scales (CheXagent-3B, MedGemma-4B, Lingshu-7B in `Table 2`) suggest that observed improvements stem from learning **genuine contextual signals** rather than exploiting dataset-specific artifacts, as architectural differences would likely respond inconsistently to systematic contradictions.
>
> - In summary, we ensure alignment between clinical context and target studies by inheriting the clinically validated framework [13] and adopting an evaluation protocol designed to minimize contradictions.
>
> - We have added above discussion at `L1052-1075` in `Appendix A.1`.
>
> ---
>
> >**[Q6]** What are the quality control measures?
>
> - We sincerely appreciate the reviewer for raising this important point.
> - Please refer to our comprehensive response to **[Q4-1]**, where we provide detailed discussion on quality control measures, including the expert review panel, systematic similarity analysis, and utterance-level validation conducted by prior work [13].
>
> ---
>
> >**[Q7]** Can you provide some evaluation on the quality of the generated data?
>
> - We sincerely appreciate the reviewer for raising this important point.
> - Please refer to our comprehensive response to **[Q4-1]**, where we provide detailed discussion on quality control measures, including the expert review panel, systematic similarity analysis, and utterance-level validation conducted by prior work [13].
>
> ---

---

> ### Author Response · Authors · 2025-11-23
> **Response to Reviewer 4zws (6/6)**
>
> ---
>
> ### Reference
>
> [1] Johnson, Alistair EW, et al. "MIMIC-CXR, a de-identified publicly available database of chest radiographs with free-text reports." Scientific data. 2019.
>
> [2] Chambon, Pierre, et al. "CheXpert plus: Hundreds of thousands of aligned radiology texts, images and patients." arXiv preprint. 2024.
>
> [3] Kahn Jr, Charles E., et al. "Toward best practices in radiology reporting." Radiology. 2009.
>
> [4] Weiss, David L., and Curtis P. Langlotz. "Structured reporting: patient care enhancement or productivity nightmare?." Radiology. 2008.
>
> [5] Yuan, Jianbo, et al. "Automatic radiology report generation based on multi-view image fusion and medical concept enrichment." MICCAI. 2019.
>
> [6] Miao, Qiguang, et al. "EVOKE: Elevating Chest X-ray Report Generation via Multi-View Contrastive Learning and Patient-Specific Knowledge." arXiv preprint. 2024.
>
> [7] Hou, Wenjun, et al. "RECAP: Towards precise radiology report generation via dynamic disease progression reasoning." arXiv preprint. 2023.
>
> [8] Liu, Fenglin, et al. "Exploring and distilling posterior and prior knowledge for radiology report generation." CVPR. 2021.
>
> [9] Wang, Fuying, Shenghui Du, and Lequan Yu. "Hergen: Elevating radiology report generation with longitudinal data." ECCV. 2024.
>
> [10] Liu, Kang, et al. "Enhanced contrastive learning with multi-view longitudinal data for chest x-ray report generation." CVPR. 2025.
>
> [11] Liu, Kang, et al. "Priorrg: Prior-guided contrastive pre-training and coarse-to-fine decoding for chest x-ray report generation." arXiv preprint. 2025.
>
> [12] Bannur, Shruthi, et al. "Maira-2: Grounded radiology report generation." arXiv preprint. 2024.
>
> [13] Delbrouck, Jean-Benoit, et al. "Automated structured radiology report generation." ACL. 2025.
>
> [14] Castillo, Chelsea, et al. "The effect of clinical information on radiology reporting: a systematic review." Journal of medical radiation sciences. 2021.
>
> [15] European Society of Radiology (ESR). "Good practice for radiological reporting. Guidelines from the European Society of Radiology (ESR)." Insights into imaging. 2011.
>
> [16] Dao, Tri, et al. "Flashattention: Fast and memory-efficient exact attention with io-awareness." NeurIPS. 2022.
>
> [17] Kwon, Woosuk, et al. "Efficient memory management for large language model serving with pagedattention." SOSP. 2023.
>
> [18] Lewis, Patrick, et al. "Retrieval-augmented generation for knowledge-intensive nlp tasks." NeurIPS. 2020.
>
> [19] Gao, Yunfan, et al. "Retrieval-augmented generation for large language models: A survey." arXiv preprint. 2023.
>
> [20] Chen, Zhihong, et al. "Chexagent: Towards a foundation model for chest x-ray interpretation." arXiv preprint. 2024.
>
> [21] Sellergren, Andrew, et al. "Medgemma technical report." arXiv preprint. 2025.
>
> [22] Xu, Weiwen, et al. "Lingshu: A Generalist Foundation Model for Unified Multimodal Medical Understanding and Reasoning." arXiv preprint. 2025.
>
> ---

---

> ### Author Response · Authors · 2025-11-27
> **Gentle Reminder**
>
> Dear Reviewer 4zws,
>
> We sincerely appreciate your time and consideration. We respectfully believe that our response has thoroughly addressed the concerns raised. If you have any remaining concerns or questions, please feel free to contact us and we would be happy to discuss and clarify them.
>
> Best,
>
> The Authors

---

### Official Review · Reviewer_3C92 · 2025-11-01

**Soundness:** 3
**Presentation:** 3
**Contribution:** 2
**Rating:** 4
**Confidence:** 2

**Summary:**

This paper introduces C-SRRG, a framework for contextualized structured radiology report generation that integrates multi-view X-rays, imaging indications, techniques, and prior studies into the report generation process. Authors curate C-SRRG derived from MIMIC-CXR and CheXpert Plus, providing structured radiology reports aligned with these contextual elements. They did comprehensive benchmarking with CheXagent-3B, MedGemma-4B, and Lingshu-7B, and demonstrate that incorporating clinical context improves report quality, mitigates temporal hallucinations, and yields stronger clinical accuracy, especially as model scale increases.

**Strengths:**

- Strong motivation: lack of contextual grounding is a real limitation
- Dataset design is well thought-out
- Multiple model scales and ablations isolate the effects of each context component
- Hallucination analysis is good as it provides a practical diagnostic for temporal reasoning failures

**Weaknesses:**

- Dataset relies partly on GPT-4–parsed elements which could introduce noise or biases
- Evaluation limited to truncated longitudinal contexts so it's unclear how well it scales to full patient histories
- No human expert evaluation beyond the “test-reviewed” split; a reader study would strengthen this alot

**Questions:**

- Could the authors quantify how annotation noise from GPT-4 parsing affects downstream model performance?
- Would radiologist-in-the-loop evaluation (e.g., assessing factual correctness or usability) further validate C-SRRG’s clinical relevance?
- Can the framework generalize to other imaging modalities?
- How does context omission affect interpretability from a clinical perspective?

---

> ### Author Response · Authors · 2025-11-23
> **Response to Reviewer 3C92 (1/4)**
>
> We sincerly appreciate you for your time and constructive comments which improve our paper. We respectfully address your concerns as following:
>
> ---
>
> >**[Q1]** Dataset relies partly on GPT-4–parsed elements which could introduce noise or biases
>
> - We sincerely appreciate the reviewer for raising this important concern about the reliability of our synthetic structured labels. We fully agree that GPT-4–parsed elements which could introduce noise or biases as we discussed in `Appendix A.1`.
>
> - However, we repectfully note that the structured report generation task was introduced and **validated by prior work** through a **rigorous reader study** with **five board-certified radiologists** [1].
>
> - They assessed the clinical validity of GPT-4 generated structured reports on the `test_reviewed` split (`Appendix B` in [1]), demonstrating that this approach maintains **sufficient clinical fidelity** for practical use as follows:
>
>     - Specifically, **five board-certified radiologists reviewed 464 reports** sampled from the MIMIC-CXR test set and CheXpert Plus validation set.
>
>     - Their analysis shows the **high average similarity ratios** which is computed as 2 × matches / total tokens in original and edited texts; 0.77 for impressions and 0.88 for findings. It indicates that modifications were **generally minor**, focusing on enhancing clarity and precision rather than altering fundamental diagnostic content.
>
>     - Furthermore, the utterance-level label validation achieved a **72% exact match rate** and **74% average Jaccard similarity** between GPT-generated consensus labels and radiologist-reviewed labels, confirming the reliability of the automated label extraction approach.
>
>     - Lastly, **large-scale data may average out such errors** across diverse examples, and the clinical validation from prior work [1] **suggests the error is manageable**.
>
> - Additionally, our evaluation protocol is designed to assess **clinical validity rather than exact textual matching**:
>
>     - **F1-SRR-BERT captures semantic similarity** appropriate for medical reporting where multiple valid phrasings exist.
>
>     - Organ-level breakdowns provide **fine-grained anatomical assessment**.
>
>     - Temporal hallucination analysis was conducted by **detecting temporal keywords solely from model generated structured report**, not from synthetically generated ground truth structured report.
>
> - In summary, the synthetic supervision used in structured report generation is thoroughly **validated by certified radiologists** [1], and our evaluation protocol naturally **avoids potential bias** when comparing generated outputs with the synthetic targets.
>
> - Nevertheless, as discussed in `Appendix A.1`, we acknowledge that our approach **lacks regulatory clearance** for clinical deployment, and **practitioners must retain final authority** over all diagnostic decisions. Given the inherent risk of errors or hallucinations in any AI system, we strongly advocate for **rigorous clinical validation and expert oversight** when exploring or extending this work.
>
> - We have added a dedicated discussion section in the revised manuscript at `L1052-1075` of `Appendix A.1`.
>
> ---

---

> ### Author Response · Authors · 2025-11-23
> **Response to Reviewer 3C92 (2/4)**
>
> ---
>
> >**[Q2]** Evaluation limited to truncated longitudinal contexts so it's unclear how well it scales to full patient histories
>
> - We sincerely appreciate the reviewer for this important observation regarding the scope of longitudinal context.
> - Our design choice to use the two most recent prior studies was motivated by **both clinical practice** and **technical constraints**:
>
>     - Radiologists typically reference a limited number of relevant prior studies when writing reports. Empirical evidence shows that radiologists consult an average of **3.2 prior imaging studies per report** [2], as clinical workflows prioritize the most recent and pertinent examinations for interval change detection. Our default setting of two most recent studies **aligns with this standard practice in radiology**, while remaining within the practical constraints of current medical MLLMs.
>
>     - Current publicly available medical MLLMs face **context-length limitations**. As shown in `Figure 6` and discussed in `Appendix A.2` (`L1101–1106`), some patients have more than **60–100 longitudinal studies**, which exceeds the practical capacity of existing medical MLLMs.
>
> - Nonethless, we sincerely agree that exploring longer patient histories remains an **important research direction**:
>     - While our approach uses the most recent prior studies, patients with extensive longitudinal histories may have clinically important examinations from **earlier studies**.
>     - This challenge could be addressed through two complementary approaches: (a) advances in efficient attention mechanisms [3, 4] that enable processing longer temporal contexts, and (b) retrieval-augmented generation (RAG) methods [5, 6] that intelligently select the most informative prior studies based on clinical relevance, potentially integrated within agentic systems for dynamic study selection.
>
> - In summary, our design choice to truncate longitudinal contexts is motivated by both **clinical practice** and **technical limitations**, but it remains a promising direction for future research.
>
> - We have clarified the above discussion at `L1119-1135` of `Appendix A.3`.
>
> ---
>
> >**[Q3]** No human expert evaluation beyond the “test-reviewed” split; a reader study would strengthen this alot
>
> - We sincerely appreciate the reviewer for this comment, and we appreciate the opportunity to clarify the scope of human expert evaluation.
> - As discussed in the response to the **[Q1]**, a comprehensive human expert evaluation was **conducted on the `test_reviewed` split** by prior work [1].
> - We acknowledge that the reader study was limited to the `test_reviewed` split due to practical constraints:
>     - Conducting expert evaluation across the full dataset would be **prohibitively resource-intensive** given the large scale of our benchmark.
>     - This approach of using a **representative sample** for validation is common practice in **large-scale medical AI** research.
>
> - However, we respectfully agree that additional radiologist-in-the-loop evaluation of model-generated reports would further strengthen clinical relevance.
>
> - We have clarified the scope of existing human expert evaluation and note this important direction for future research at `L1061-1063` of `Appendix A.1`.
>
> ---

---

> ### Author Response · Authors · 2025-11-23
> **Response to Reviewer 3C92 (3/4)**
>
> ---
>
> >**[Q4]** Could the authors quantify how annotation noise from GPT-4 parsing affects downstream model performance?
>
> - We sincerely appreciate the reviewer for this insightful question. We acknowledge that we have not conducted a dedicated quantification study isolating the impact of annotation noise on downstream model performance.
> - Consistent to **[Q1]** and **[Q3]**, we respectfully argue that the **impact is limited**.
> - As mentioned in **[Q1]** and **[Q3]**, the GPT-4 parsing process was **clinically validated** through a rigorous reader study by prior work [1].
>
>     - **Five board-certified radiologists** assessed semantic equivalence and clinical validity of structured reports on the `test_reviewed` split.
>     - This validation provides evidence that the parsing maintains **high clinical fidelity**, though it does not directly measure downstream model impact.
>
> - Moreover, several factors and observations suggest the noise has **limited effect on model performance**:
>
>     - Our evaluation metrics (**F1-SRR-BERT**) measure **semantic similarity** using contextual embeddings rather than exact text matching, which naturally accounts for paraphrasing or minor variations that may arise from GPT-4 parsing.
>
>     - With **large-scale training data** containing thousands of reports, models learn aggregate statistical patterns rather than memorizing individual examples, allowing random noise to be averaged out during training.
>
>     - Consistent performance trends across **three different model architectures** with varying scales (CheXagent-3B, MedGemma-4B, Lingshu-7B in `Table 2`) suggest that observed improvements stem from learning **genuine contextual signals** rather than exploiting dataset-specific annotation artifacts, as architectural differences would likely respond inconsistently to systematic biases.
>
> ---
>
> >**[Q5]** Would radiologist-in-the-loop evaluation (e.g., assessing factual correctness or usability) further validate C-SRRG’s clinical relevance?
>
> - We sincerely appreciate the reviewer for this suggestion. We respectfully agree that radiologist-in-the-loop evaluation would significantly strengthen the validation of clinical relevance for the proposed C-SRRG:
>
>     - Such evaluation would provide **crucial insights beyond automated metrics**, including whether generated reports meet radiologists' quality standards, accurately reflect clinical findings, and are actionable in real clinical workflows.
>
>     - It would be particularly valuable for assessing subtle aspects such as **clinical appropriateness of terminology**, **coherence of diagnostic reasoning**, and the **impact of reduced temporal hallucinations** on clinical decision-making.
>
> - However, conducting radiologist evaluation presents **significant resource challenges**, including the need for substantial funding and expert time commitments.
> - We acknowledge this as an important direction for future work as resources become available, particularly for translating these methods toward clinical deployment.
> - We have clarified the above discussion on the limitation and futurework at `Appendix A.5`.
>
> ---
>
> >**[Q6]** Can the framework generalize to other imaging modalities?
>
> - We sincerely appreciate the reviewer for this thought-provoking question.
> - The proposed C-SRRG is motivated by the diagnostic workflow of radiologists, where clinical context (e.g., imaging indication, technique, and prior studies) plays a **crucial role in report generation across all imaging modalities**, not just chest X-rays.
> - We believe C-SRRG has **strong potential to generalize** to other modalities (e.g., CT, MRI, ultrasound, and mammography):
>
>     - These modalities similarly rely on **contextual information** for accurate interpretation [7].
>
>     - The core components (multi-view images, clinical indication, imaging technique, prior studies) are **naturally applicable across modalities**, though specific instantiations may vary. For example, CT scans involve **multiple slices rather than views**, and MRI includes **diverse sequences with different contrasts**.
>
> - However, we respectfully note that each imaging modality has **unique characteristics** and reporting conventions that may require **modality-specific adaptations**:
> - **Empirical validation on other modalities** would be necessary to confirm generalization, which we view as an important direction for **future research**.
>
> - We have clarified the above discussion in `Appendix A.6`.
>
> ---

---

> ### Author Response · Authors · 2025-11-23
> **Response to Reviewer 3C92 (4/4)**
>
> ---
>
> >**[Q7]** How does context omission affect interpretability from a clinical perspective?
>
> - We sincerely appreciate the reviewer for this insightful question. Each clinical context component plays a **distinct role in radiological interpretation**, and we believe that omitting these elements can significantly compromise **diagnostic accuracy and clinical utility**:
>
>     - **Multi-view images** [8]: Different views provide **complementary information** critical for accurate diagnosis. For instance, lateral chest X-ray views reveal retrocardiac or retrosternal abnormalities obscured in frontal views alone. Without multi-view information, models may miss pathologies hidden behind anatomical structures (e.g., middle lobe pneumonia, posterior rib fractures).
>
>     - **Indication** [9]: Clinical indication provides **essential context** about patient symptoms and clinical questions that guide diagnostic focus. For example, fever and cough direct attention to infectious etiologies, while trauma history focuses on fractures. Without indication, models cannot tailor findings to address referring physicians' specific concerns, potentially emphasizing irrelevant details while overlooking clinically significant findings.
>
>     - **Technique** [10, 11]: Technical parameters document **imaging quality, protocols, and limitations** crucial for proper interpretation. Omitting technique information can lead models to misinterpret artifacts as pathology. For instance, limited inspiratory effort may cause apparent cardiomegaly that resolves with proper technique, while portable examinations have inherent quality limitations requiring acknowledgment.
>
>     - **Prior studies** [2]: Temporal comparison with prior examinations enables **detection of interval changes** fundamental to radiological practice. For example, tracking pulmonary nodule size over time distinguishes stable benign lesions from growing malignancies, directly impacting clinical management. Without prior study information, models may generate **temporal hallucinations** by referencing non-existent exams or fail to detect critical interval changes (disease progression, treatment response).
>
> - In summary, context omission not only reduces model performance quantitatively (`Tabs. 7 and 8`) but also **fundamentally limits clinical utility and safety** of generated reports by removing essential information that radiologists rely on for accurate diagnosis and appropriate patient management.
>
> - We have included above clinical interpretations of each clinical context component in `Appendix D` for the better picture of how each component plays in radiology report generation.
>
> ---
>
> ### Reference
>
> [1] Delbrouck, Jean-Benoit, et al. "Automated structured radiology report generation." ACL. 2025.
>
> [2] Haygood, Tamara Miner, et al. "Consultation and citation rates for prior imaging studies and documents in radiology." Journal of Medical Imaging. 2018.
>
> [3] Dao, Tri, et al. "Flashattention: Fast and memory-efficient exact attention with io-awareness." NeurIPS. 2022.
>
> [4] Kwon, Woosuk, et al. "Efficient memory management for large language model serving with pagedattention." SOSP. 2023.
>
> [5] Lewis, Patrick, et al. "Retrieval-augmented generation for knowledge-intensive nlp tasks." NeurIPS. 2020.
>
> [6] Gao, Yunfan, et al. "Retrieval-augmented generation for large language models: A survey." arXiv preprint. 2023.
>
> [7] Hattori, Shinya, et al. "Impact of clinical information on CT diagnosis by radiologist and subsequent clinical management by physician in acute abdominal pain." European Radiology. 2021.
>
> [8] Yuan, Jianbo, et al. "Automatic radiology report generation based on multi-view image fusion and medical concept enrichment." MICCAI. 2019.
>
> [9] Castillo, Chelsea, et al. "The effect of clinical information on radiology reporting: a systematic review." Journal of medical radiation sciences. 2021.
>
> [10] Kahn Jr, Charles E., et al. "Toward best practices in radiology reporting." Radiology. 2009.
>
> [11] Mowery, Myles L., and Vikramjeet Singh. "X-ray production technical evaluation." 2020.
>
> ---

---

> ### Author Response · Authors · 2025-11-27
> **Gentle Reminder**
>
> Dear Reviewer 3C92,
>
> We sincerely appreciate your time and consideration. We respectfully believe that our response has thoroughly addressed the concerns raised. If you have any remaining concerns or questions, please feel free to contact us and we would be happy to discuss and clarify them.
>
> Best,
>
> The Authors

---

### Official Review · Reviewer_TNV1 · 2025-11-01

**Soundness:** 2
**Presentation:** 2
**Contribution:** 2
**Rating:** 2
**Confidence:** 4

**Summary:**

The paper tackles the structured radiology report generation (SRRG) task with a proposed contextualized SRRG (C-SRRG). The work contributes a new C-SRRG dataset and evaluates three state-of-the-art medical MLLMs with and without the clinical context.

**Strengths:**

- Temporal hallucination is introduced nicely, and the paper shows substantial mitigation.
- The ablation study is detailed and sound.

**Weaknesses:**

- The technical novelty seems lacking.
- The claim "the critical importance of clinical context in scaling up MLLMs for SRRG." (377) is questionable due to different model families (MedGemma vs Lingshu).
- While temporal hallucination seems problematic, it is not clearly shown how critical/large/common the issue is. Table 9 is performed on only MedGemma-4B. An experiment on the other two models would help giving a better picture of the issue.

**Questions:**

- Is the "curated" C-SRRG dataset mostly data-processing on top of the existing SRRG dataset from [1]?
- Does the "training failure" behavior on CheXagent-3B (333) exist before training as well?
- How is the baseline performance without any fine-tuning in the evaluation?

[1] Delbrouck, Jean-Benoit, et al. "Automated Structured Radiology Report Generation." (2025).

---

> ### Author Response · Authors · 2025-11-23
> **Response to Reviewer TNV1 (1/6)**
>
> We sincerly appreciate you for your time and constructive comments which improve our paper. We respectfully address your concerns as following:
>
> ---
>
> >**[Q1]** The technical novelty seems lacking.
>
> - We sincerely appreciate the reviewer for this valuable feedback regarding the technical novelty. However, we respectfully believe that as noted in `L104–115`, our primary contribution do not lie in **technical novelty** but rather in **identifying a key limitation of SRRG**, **curating a dataset with essential clinical context**, and **providing empirical evidence** through benchmarking to facilitate future work on SRRG:
>
>     - Our work systematically analyzes **the clinical workflow of radiologists** to identify relevant contextual signals that are routinely consulted during report writing, and demonstrates empirically that omitting such context leads to **systematic issues like temporal hallucinations** as depicted in `L187-197` in `§3.1`.
>
>     - A key distinction from prior work is the **comprehensive incorporation of multiple clinical contexts simultaneously**, multi-view chest radiographs, imaging technique specifications, clinical indications, and comparison with prior studies, rather than limiting analysis to isolated contextual elements as depicted in `L198-215` in `§3.1`.
>
>     - To the best of our knowledge, this study is the **first to curate a large-scale structured radiology report generation dataset** that aligns these diverse clinical contexts with structured radiology reports, enabling systematic investigation of their effects as in `Tabs. 5, 6, 7, and 8`.
>
>     - Furthermore, we provide the **first comprehensive empirical study** examining how these clinical contexts influence structured report generation across **multiple state-of-the-art medical MLLMs (e.g., CheXagent-3B, MedGemma-4B, Lingshu-7B)**, revealing consistent improvements (**+2–7% F1-SRR-BERT**) and substantial reductions in temporal hallucinations (**12–18%**).
>
> - Notably, investigating the **effect of clinical contexts in medical MLLMs remains largely underexplored**, despite **the significant importance of report writing in the clinical workflow** described in `L43-47` (i.e. report written by radiologists is read by clinicians to do desicion-making on the patient).
> - This is partly due to the limited capability MLLMs in handling long context, as evidenced by the difficulties of CheXagent-3B with comprehensive contextual information (`L331-333`, `Appendix A.2, A.3`, and `Appendix C`).
>
> - We believe that the above discussion clarifies the contributions presented in `L106–117`.
>
> ---
>
> >**[Q2]** The claim "the critical importance of clinical context in scaling up MLLMs for SRRG." (377) is questionable due to different model families (MedGemma vs Lingshu).
>
> - We sincerely agree that our current claim in `L377` overstates what can be concluded from our experimental design, and we appreciate this rebuttal as an opportunity to further clarify and discuss this point in context:
>
>     - As raised in this concern, comparing different model families (MedGemma [1] vs. Lingshu [2]) introduces **confounding factors**. Specifically, the observed differences could be attributed to **model architecture** and **training methodology** rather than purely to the interaction between **clinical context and model scale**.
>
>     - To clarify this point, **scaling up each model variant** is appropriate. However, our evaluation is constrained by computational resources and model availability. As noted in `L338-339`, all experiments were conducted on a **single H100 GPU**, which limits the scale of models we could evaluate.
>
>     - More crucially, **publicly available medical MLLMs have limited size options**: MedGemma [1] offers 4B and **27B** versions, while Lingshu [2] provides 7B and **32B** variants. We respectfully argue that these larger variants (e.g., MedGemma-27B or Lingshu-32B) are not only beyond our computational capacity but also **beyond what is reasonable for academic fine-tuning and evaluation**.
>
> - Accordingly, we have toned down the claim in `L377` to: "We observe that performance **increases as the number of parameters grows from 3B to 7B**; however, this does not imply that clinical context is always helpful, as there may be **confounding factors**."
>
> ---

---

> ### Author Response · Authors · 2025-11-23
> **Response to Reviewer TNV1 (2/6)**
>
> ---
>
> >**[Q3-1]** While temporal hallucination seems problematic, it is not clearly shown how critical/large/common the issue is. Table 9 is performed on only MedGemma-4B. An experiment on the other two models would help giving a better picture of the issue.
>
> - We sincerely appreciate the reviewer for pointing this out. We first kindly note that the reason why we initially focused on MedGemma-4B is its **representativeness**.
> - However, we sincerely agree that examining the temporal hallucination effect only for MedGemma-4B may weaken our conclusion. Therefore, we conducted additional experiments to evaluate **temporal hallucination for CheXagent-3B and Lingshu-7B** as well.
> - As noted in `L331–333`, we utilize only the clinical indication with CheXagent-3B on C-SRRG-impression, rather than the full clinical context including prior studies, due to training failure behavior. Therefore, we exclude it from the evaluation of temporal hallucination on C-SRRG-impression.
>
>     **[Table TNV1-1]** Results of temporal hallucination mitigation effect with **CheXagent-3B**.
>     | Task | Split | Temporal Hallucination Rate | | Mitigation |
>     |------|-------|-------------------|-----------------|------------|
>     | | | **Baseline (✗)** | **C-SRRG (✓)** | |
>     | Findings | Valid | 170/976 (17.4%) | 280/976 (28.7%) | **+11.3%** |
>     | | Test | 462/1459 (31.7%) | 614/1459 (42.1%) | **+10.4%** |
>     | | Test-reviewed | 65/233 (27.9%) | 77/233 (33.0%) | **+5.2%** |
>     | | Overall | 697/2668 (26.1%) | 971/2668 (36.4%) | **+10.3%** |
>
>     **[Table TNV1-2]** Results of temporal hallucination mitigation effect with **Lingshu-7B**.
>     | Task | Split | Temporal Hallucination Rate | | Mitigation |
>     |------|-------|-------------------|-----------------|------------|
>     | | | **Baseline (✗)** | **C-SRRG (✓)** | |
>     | Findings | Valid | 174/976 (17.8%) | 140/976 (14.3%) | **-3.5%** |
>     | | Test | 491/1459 (33.7%) | 369/1459 (25.3%) | **-8.4%** |
>     | | Test-reviewed | 64/233 (27.5%) | 42/233 (18.0%) | **-9.4%** |
>     | | Overall | 729/2668 (27.3%) | 551/2668 (20.7%) | **-6.7%** |
>     | Impression | Valid | 694/1505 (46.1%) | 382/1505 (25.4%) | **-20.7%** |
>     | | Test | 1156/2219 (52.1%) | 623/2219 (28.1%) | **-24.0%** |
>     | | Test-reviewed | 116/231 (50.2%) | 50/231 (21.6%) | **-28.6%** |
>     | | Overall | 1966/3955 (49.7%) | 1055/3955 (26.7%) | **-23.0%** |
>
> `Table TNV1-1` and `Table TNV1-2` show that temporal hallucinations are widespread across models trained without clinical context:
>     - C-SRRG-Findings: **26.1% (CheXagent-3B), 22.9% (MedGemma-4B), and 27.3% (Lingshu-7B)**;
>     - C-SRRG-Impression **43.8% (MedGemma-4B) and 49.7% (Lingshu-7B)**.
> - This demonstrates that **temporal hallucinations are not model-specific artifacts but a fundamental systematic problem** affecting generated reports.
>
> - In `Tab. 9` and `Table TNV1-2`, MedGemma-4B and Lingshu-7B show substantial hallucination reductions with C-SRRG (e.g., **−12.2%/−18.0%** and **−6.7%/−23.0%** for findings/impression respectively).
>
> - However, in `Table TNV1-1`, CheXagent-3B exhibits an **unexpected opposite pattern**, with temporal hallucinations increasing by **+10.3%** for findings generation.
> - We defer the explanation for the increased temporal hallucination of CheXagent-3B involving the terms 'stable and 'unchanged' to **[Q3-2]**, and we provide our hypothesis on why this issue occurs only for CheXagent-3B—given its limited context capacity and pretraining scheme—in **[Q3-3]**.
>
> - In summry, `Table TNV1-1 and TNV1-2` have two important implications:
>     - **Temporal hallucinations are pervasive** across current MLLM-based report generation systems, affecting clinical reliability, and
>     - Clinical context integration requires **appropriate architectural capacity**, simply providing context is insufficient without models designed to effectively utilize such information.
>
>
> ---

---

> ### Author Response · Authors · 2025-11-23
> **Response to Reviewer TNV1 (3/6)**
>
> ---
>
> >**[Q3-2]** Explanation for the **increased temporal hallucination** of CheXagent-3B.
>
> - We further investigate the unexpected behavior of CheXagent-3B in `Table TNV1-1` and identify its distinctive failure patterns.
> - Unlike other models, we observe that CheXagent-3B exhibits **excessive overuse of 'stable' and 'unchanged'** when trained **with clinical context**.
> - Specifically, `Table TNV1-3` shows dramatic increases in two particular keywords: **'stable' (+89.9%, from 307 to 583 occurrences)** and **'unchanged' (+11.3%, from 319 to 355 occurrences)** across all evaluation splits.
>
>     **[Table TNV1-3]** Temporal keyword usage comparison for CheXagent-3B on C-SRRG-Findings.
>     |Keyword|Without Clinical Context|With Clinical Context|Change|
>     |:-|:-:|:-:|:-:|
>     |**stable**|**307**|**583**|**+276 (+89.9%)**|
>     |**unchanged**|**319**|**355**|**+36 (+11.3%)**|
>     |new|137|88|-49 (-35.8%)|
>     |...|...|...|...|
>     |**Total Keywords**|**1,007**|**1,122**|**+115 (+11.4%)**|
>
>
> - Importantly, the behavior in `Table TNV1-3` appears **specific to CheXagent-3B**. As shown in `Table TNV1-4`, both MedGemma-4B and Lingshu-7B do **not exhibit excessive overuse** of 'stable' and 'unchanged', with substantial reductions in such terms when trained with clinical context.
>
>     **[Table TNV1-4]** Comparison of temporal keyword usage patterns across models.
>     |Model|Task|stable|unchanged|Total Keywords|
>     |:-|:-|:-:|:-:|:-:|
>     |**CheXagent-3B**|Findings|307 → 583 (+89.9%)|319 → 355 (+11.3%)|1,007 → 1,122 (+11.4%)|
>     |**MedGemma-4B**|Findings|86 → 46 (-46.5%)|500 → 113 (-77.4%)|929 → 373 (-59.8%)|
>     |**MedGemma-4B**|Impression|1,487 → 900 (-39.5%)|457 → 202 (-55.8%)|2,974 → 1,521 (-48.9%)|
>     |**Lingshu-7B**|Findings|68 → 60 (-11.8%)|653 → 391 (-40.1%)|1,082 → 671 (-38.0%)|
>     |**Lingshu-7B**|Impression|1,618 → 960 (-40.7%)|591 → 209 (-64.6%)|3,549 → 1,525 (-57.0%)|
>
> - To provide a more comprehensive view, we present detailed temporal keyword distributions for all models and tasks below, which clearly demonstrate that the overuse of 'stable' and 'unchanged' is unique to CheXagent-3B.
>
>     **[Table TNV1-5a]** Temporal keyword usage for MedGemma-4B on C-SRRG-Findings.
>     |Keyword|Without Clinical Context|With Clinical Context|Change|
>     |:-|:-:|:-:|:-:|
>     |**unchanged**|**500**|**113**|**-387 (-77.4%)**|
>     |prior|115|34|-81 (-70.4%)|
>     |from prior|112|33|-79 (-70.5%)|
>     |**stable**|**86**|**46**|**-40 (-46.5%)**|
>     |...|...|...|...|
>     |**Total Keywords**|**929**|**373**|**-556 (-59.8%)**|
>
>     **[Table TNV1-5b]** Temporal keyword usage for MedGemma-4B on C-SRRG-Impression.
>     |Keyword|Without Clinical Context|With Clinical Context|Change|
>     |:-|:-:|:-:|:-:|
>     |**stable**|**1,487**|**900**|**-587 (-39.5%)**|
>     |**unchanged**|**457**|**202**|**-255 (-55.8%)**|
>     |...|...|...|...|
>     |**Total Keywords**|**2,974**|**1,521**|**-1,453 (-48.9%)**|
>
>     **[Table TNV1-5c]** Temporal keyword usage for Lingshu-7B on C-SRRG-Findings.
>     |Keyword|Without Clinical Context|With Clinical Context|Change|
>     |:-|:-:|:-:|:-:|
>     |**unchanged**|**653**|**391**|**-262 (-40.1%)**|
>     |prior|131|34|-97 (-74.0%)|
>     |from prior|129|33|-96 (-74.4%)|
>     |**stable**|**68**|**60**|**-8 (-11.8%)**|
>     |...|...|...|...|
>     |**Total Keywords**|**1,082**|**671**|**-411 (-38.0%)**|
>
>     **[Table TNV1-5d]** Temporal keyword usage for Lingshu-7B on C-SRRG-Impression.
>     |Keyword|Without Clinical Context|With Clinical Context|Change|
>     |:-|:-:|:-:|:-:|
>     |**stable**|**1,618**|**960**|**-658 (-40.7%)**|
>     |**unchanged**|**591**|**209**|**-382 (-64.6%)**|
>     |...|...|...|...|
>     |**Total Keywords**|**3,549**|**1,525**|**-2,024 (-57.0%)**|
>
> - By comparing `Table TNV1-3` with `Table TNV1-5a, TNV1-5b, TNV1-5c, and TNV1-5d`, the detailed breakdowns reveal several critical insights:
>     - **CheXagent-3B is unique** in showing increased usage of 'stable' (+89.9%) and 'unchanged' (+11.3%), while all other model-task combinations show substantial decreases (ranging from -11.8% to -77.4%).
>     - MedGemma-4B and Lingshu-7B both demonstrate **proper learning of temporal context**, reducing overall temporal keyword usage by 38-60% as they generate more precise, context-appropriate descriptions rather than overusing generic temporal statements.
>     - The reduction patterns are **consistent across both findings and impression tasks** for MedGemma-4B and Lingshu-7B, suggesting robust learning of temporal reasoning capabilities.

---

> ### Author Response · Authors · 2025-11-23
> **Response to Reviewer TNV1 (4/6)**
>
> - To quantify the impact of this overuse pattern, we conduct an exclusion analysis where we **progressively remove 'stable' and 'unchanged' from temporal hallucination detection**.
>
>     **[Table TNV1-6a]** CheXagent-3B hallucination rates excluding 'stable' from detection.
>     |Split|Without Clinical Context|With Clinical Context|Difference|
>     |:-|:-:|:-:|:-:|
>     |Validate (976 samples)|11.9% (116)|10.8% (105)|-1.1%|
>     |Test (1,459 samples)|23.7% (346)|21.0% (307)|-2.7%|
>     |Test Reviewed (233 samples)|21.0% (49)|20.2% (47)|-0.9%|
>     |**Overall (2,668 samples)**|**19.2% (511)**|**17.2% (459)**|**-1.9%**|
>
>     **[Table TNV1-6b]** CheXagent-3B hallucination rates excluding both 'stable' and 'unchanged' from detection.
>     |Split|Without Clinical Context|With Clinical Context|Difference|
>     |:-|:-:|:-:|:-:|
>     |Validate (976 samples)|7.8% (76)|4.9% (48)|-2.9%|
>     |Test (1,459 samples)|14.1% (206)|7.2% (105)|-6.9%|
>     |Test Reviewed (233 samples)|13.7% (32)|7.3% (17)|-6.4%|
>     |**Overall (2,668 samples)**|**11.8% (314)**|**6.4% (170)**|**-5.4%**|
>
>
> - `Table TNV1-6a` shows that the hallucination rate of CheXagent-3B decreases when the term 'stable' is excluded, and `Table TNV1-6b` reveals that **83% of its temporal hallucinations (30.1% out of 36.4%) arise from overuse of just 'stable' and 'unchanged'.**
> - When these terms are excluded from evaluation, the temporal hallucination of CheXagent-3B is acutally **reduced with clinical context** (6.4% vs. 11.8%).
> - In summary, temporal hallucinations of CheXagent-3B are **dominated by the terms 'stable' and 'unchanged'**, a pattern **not observed in MedGemma-4B or Lingshu-7B**.
>
> - We have incorporated these expanded results and detailed discussion into the revised manuscript at `Appendix C.1`.
>
> ---
>
> >**[Q3-3]** Why the issue in **[Q3-2]** occurs only for CheXagent-3B?
>
> - We now explain why this issue arises only for CheXagent-3B, based on its limitations in **(1) context length** and **(2) pretraining scheme**.
> - First, we summarize the failures of CheXagent-3B observed in `Table TNV1-1`, `Tables 3 and 4`, and `Appendix A.2, A.3, and C` when clinical context is used:
>     - **Minimal improvements in traditional metrics** despite context availability;
>     - **Increased temporal hallucinations** in findings generation (+10.3%)** instead of the expected reduction observed in other models; and
>     - **Catastrophic training failure in impression generation** that produces malformed outputs (BLEU: 9.44→2.57, ROUGE-L: 34.03→21.76).
>
> - We hypothesize that these failures stem from fundamental limitations in **(1) context length**:
>
>     - CheXagent-3B is built upon Phi-2 (2.7B parameters), which was pretrained with a maximum sequence length of **2,048 tokens** [5].
>     - While CheXagent-3B extends this to **4,096 tokens during fine-tuning** [4], this capacity remains insufficient for processing rich clinical context that combines visual tokens, imaging techniques, clinical indications, and extensive prior study narratives.
>     - The relatively small model capacity (3.1B parameters total) may be further insufficient to simultaneously encode rich visual features, process extensive clinical text, maintain attention over long contexts, and generate structured outputs conforming to format constraints.
>
> - We also attribute the failures to its **(2) pretraining scheme**:
>
>     - The CheXinstruct [4] dataset predominantly consists of short, task-specific instruction-response pairs using ten manually-defined templates for each of the 35 tasks.
>     - Most tasks involve concise outputs: view classification with three-choice answers (AP/PA/Lateral), disease identification with binary responses (Yes/No), and findings generation from CXR images.
>     - Critically, **none of the 35 tasks require processing long clinical context alongside images to generate structured reports**.
>     - This creates a fundamental gap: the model learns to generate reports with minimal textual context but fails when required to integrate extensive clinical narrative while maintaining structured output format.
>     - The dramatic format violations (models generating unstructured report) and the overuse 'stable' and 'unchanged' (`Table TNV1-4`) suggest that CheXagent-3B may not generalize beyond the constrained instruction formats encountered during training.
>
> - The discussion throughout **[Q3-1]**, **[Q3-2]**, and **[Q3-3]** shows that **effective clinical context integration requires appropriate architectural capacity and a suitable training regimen**; simply providing additional context is insufficient when models are not designed to leverage such information effectively.
>
> - We have added a detailed discussion in the revised manuscript at `Appendix C.3`.
>
> ---

---

> ### Author Response · Authors · 2025-11-23
> **Response to Reviewer TNV1 (5/6)**
>
> ---
>
> > **[Q4]** Is the "curated" C-SRRG dataset mostly data processing on top of the existing SRRG dataset from [3]?
>
> - We sincerely appreciate the reviewer for raising this point and apologize for the unclear presentation. However, we have already acknowledged that our dataset **builds upon the SRRG task framework** proposed in prior work [3], as stated in `L248–249`.
>
> - This is because structured report generation is **a very recently introduced task**, and [3] provides a testbed that has been **thoroughly validated** through a **rigorous reader study** involving **five board-certified radiologists**.
>
> - More importantly, we respectfully believe that our contribution goes beyond simple data processing, as it addresses a **fundamental limitation in the current structured radiology report generation framework**, as noted in `L104-115`.
>
> - As detailed in our response to **[Q1]**, our work introduces the first **contextualized structured radiology report generation (C-SRRG)** framework that systematically incorporates **comprehensive clinical context** (multi-view images, imaging techniques, clinical indications, and prior studies), reflecting the actual workflow of radiologists. We further conduct extensive experiments analyzing **how these contextual signals influence report generation quality** across multiple state-of-the-art medical MLLMs.
>
> - The resulting **C-SRRG** dataset enables systematic investigation of how clinical context affects both report quality and temporal consistency, which was not possible with the original SRRG dataset.
>
> - We believe this contextualized benchmark will meaningfully facilitate future research on clinically aligned radiology report generation.
>
>
> ---
>
> >**[Q5]** Does the "training failure" behavior on CheXagent-3B (333) exist before training as well?
>
> - We sincerely appreciate the reviewer for rasing this point. However, we would like to note that "training failure" behavior on CheXagent-3B **always exists** before training:
>
>     - First of all, we refer the "training failure behavior" in `L333` to the **inability of model** to generate outputs in the **proper structured format**. This naturally occurs at the initialization of the pretrained model before fine-tuning, as it was **never trained to produce structured reports** in the SRRG format.
>
>     - We next provide an additional clarification on how this training failure behavior manifested in CheXagent-3B after training, which is already included in `Appendix C.2`.
>
>     - Specifically, we found that **even after fine-tuning with full clinical context** including prior studies, CheXagent-3B continued to **struggle with maintaining the proper structured format** for impression generation tasks.
>
>     - For example, CheXagent-3B often generates only "Pneumothorax" or "Pneumonia", while the expected format is "1. Slight decrease in size of the right apicolateral pneu-mothorax with chest tube in place. 2. Unchanged multifocal right-sided pulmonary opacities...".
>
>     - We attribute this to the incapability of models for handling long sequence and pre-training scheme as discussed in `Appendix A.2` and our response to **[Q3-3]**, thus only occured in CheXagent-3B as in `Appendix C`
>
> - Due to the above reasons, we only used clinical indication (without prior studies) for CheXagent-3B on C-SRRG-Impression experiments, while C-SRRG-Findings experiments could successfully generate the expected format, although it still failed to trained properly to integrate temporal relationships as shown in temporal hallucination effect in `Table TNV1-1` in our response to **[Q3-1]**.
>
> ---

---

> ### Author Response · Authors · 2025-11-23
> **Response to Reviewer TNV1 (6/6)**
>
> ---
>
> >**[Q6]** How is the baseline performance without any fine-tuning in the evaluation?
>
> - We sincerely appreciate the reviewer for raising this concern. However, we respectfully believe that it may stem from a misunderstanding.
> - As discussed in **[Q5]**, medical MLLMs **without any fine-tuning** cannot generate outputs in the expected or proper format. This is because SRRG is a very recent task [3], and existing models have **not been pretrained on any task similar to SRRG**:
>
>     - For example, in the **findings generation** task, models must produce reports in a **specific structured format with predefined anatomical sections** (e.g., "Lungs and Airways:"", "Pleura:"", "Cardiovascular:"").
>
>     - However, pretrained models cannot generate this structured format without instruction fine-tuning; instead, they produce free-form text rather than adhering to the required section structure. As a result, findings generation **cannot be meaningfully evaluated without fine-tuning**.
>
>     - Unlike findings, **impression generation** requires only **numbered elements without strict section headers**, making it feasible to evaluate pretrained models.
>
> - Below, we present the performance of pretrained models (**without fine-tuning**) on impression generation, evaluated both with (✓) and without (✗) clinical context at inference time.
>
>     **[Table TNV1-3]** Results of baseline performance on impression generation without any fine-tuning.
>     |Model|Clinical Context|Split|BLEU|ROUGE-L|BERT Score|F1-RadGraph|Precision|Recall|F1-Score|
>     |:-|:-:|:-:|:-:|:-:|:-:|:-:|:-:|:-:|:-:|
>     |**CheXagent-3B**|✗|Valid|1.03|18.71|30.00|13.80|67.65|37.64|44.71|
>     |||Test|0.98|15.19|21.22|12.27|60.07|28.65|36.14|
>     |||Test-reviewed|1.13|16.02|22.56|13.20|59.20|32.76|39.70|
>     ||✓|Valid|1.44|19.13|30.23|13.98|67.66|38.60|45.47|
>     |||Test|1.45|15.75|25.83|12.47|59.77|30.66|37.40|
>     |||Test-reviewed|1.32|15.91|24.93|12.38|57.44|31.81|38.65|
>     |**MedGemma-4B**|✗|Valid|1.23|14.26|34.10|9.11|68.13|53.97|55.06|
>     |||Test|1.21|13.44|32.86|9.60|60.07|48.46|48.89|
>     |||Test-reviewed|0.70|11.73|29.99|7.58|55.20|48.68|47.50|
>     ||✓|Valid|1.80|15.04|37.60|11.29|66.18|61.74|58.88|
>     |||Test|1.85|15.02|38.06|11.76|59.21|57.54|53.54|
>     |||Test-reviewed|1.17|13.16|34.88|8.94|56.39|54.78|50.78|
>     |**Lingshu-7B**|✗|Valid|1.46|14.11|37.18|9.33|64.96|41.23|46.26|
>     |||Test|1.58|13.90|39.37|10.40|55.45|34.80|39.02|
>     |||Test-reviewed|1.27|13.23|37.38|9.34|49.93|33.17|36.56|
>     ||✓|Valid|4.44|21.29|50.03|12.54|61.45|51.64|51.44|
>     |||Test|4.70|21.92|52.44|12.80|54.50|46.05|45.11|
>     |||Test-reviewed|2.22|14.81|39.97|9.12|51.69|36.37|39.23|
>
>
> - In `Table TNV1-3`, we observe that pretrained medical MLLMs (w/o fine-tuning) **perform substantially worse** than their fine-tuned variants in `Tab. 4` using F1-SRR-BERT:
>     - CheXagent-3B: **36.14~45.47** (w/o fine-tuning) vs. **49.74~59.10** (with finetuning)
>     - MedGemma-4B: **47.50~58.88** (w/o fine-tuning) vs. **49.51~62.31** (with fine-tuning)
>     - Lingshu-7B: **36.56~51.44** (w/o fine-tuning) vs **48.37~62.48**
>
> - Interestingly, MedGemma-4B **(w/o fine-tuning) with context** (**58.88**) shows better performance than MedGemma-4B **(with fine-tuning) w/o context** (**56.81**), but MedGemma-4B **(with fine-tuning) with context** still surpasses this (**62.31**).
>
> - Moreover, **without fine-tuning**, MedGemma-4B is stronger than Lingshu-7B (**47.50~58.88** vs. **36.56~51.44**). However, **Lingshu-7B benefits more from instruction tuning** and eventually reaches **comparable or superior performance** as shown in `Tab. 4`.
>
> - Lastly, **providing clinical context** at inference improves medical MLLMs even **without fine-tuning** using F1-SRR-BERT:
>     - On validation by +0.76 (CheXagent-3B), +3.82 (MedGemma-4B), and +5.18 (Lingshu-7B); on test by +1.26 (CheXagent-3B), +4.65 (MedGemma-4B), and +6.09 (Lingshu-7B); and on test-reviewed by +2.67 (Lingshu-7B) and +3.28 (MedGemma-4B).
>     - With only one exception exists (CheXagent-3B on test-reviewed split showing -1.05 degradation)
>
> - In summary, we observe that **instruction tuning is crucial for structured report generation**, and **clinical context provides meaningful benefits** even **without fine-tuning**.
>
> - A detailed discussion of pretrained baseline behavior without fine-tuning has been added in `Appendix E`.
>
> ---
>
> ### Reference
>
> [1] Sellergren, Andrew, et al. "Medgemma technical report." arXiv preprint. 2025.
>
> [2] Xu, Weiwen, et al. "Lingshu: A Generalist Foundation Model for Unified Multimodal Medical Understanding and Reasoning." arXiv preprint. 2025.
>
> [3] Delbrouck, Jean-Benoit, et al. "Automated structured radiology report generation." ACL. 2025.
>
> [4] Chen, Zhihong, et al. "Chexagent: Towards a foundation model for chest x-ray interpretation." arXiv preprint. 2024.
>
> [5] Li, Yuanzhi, et al. "Textbooks are all you need ii: phi-1.5 technical report." arXiv preprint. 2023.
>
> ---

---

> > ### Comment · Reviewer_TNV1 · 2025-11-26
> >
> > Thank you for the very detailed responses. I appreciate the insight on the temporal hallucinations and the additional evaluations. While the clinical context being critical as models scale up is a promising claim, the current experiment setup (due to limited resources, understandly) can't fully support the claim. I have adjusted my score, and I have a follow-up question.
> > Since the paper is claiming the need of clinical context for SoTA MLLM, the smaller models in the families are not actually SoTA. The SRRG project page indicates models such as CheXpert-Plus and MAIRA-2 which might be SoTA for the SRRG task. The current models in the experiments do not represent SoTA in my opinion, so the framing between the contribution and the experiments isn't fully aligned as it is in the current form.

---

> > > ### Author Response · Authors · 2025-11-27
> > > **Response to Reviewer TNV1 (1/4)**
> > >
> > > We sincerely appreciate your time and efforts to consider our paper, and espeically your proactive engagement during this rebuttal. We address your follow-up questions below:
> > >
> > > ---
> > >
> > > > **[Follow-Up-Q1]** Since the paper is claiming the need for clinical context for SoTA MLLMs, the smaller models in the families are not actually SoTA.
> > >
> > > - We sincerely appologize that our claim regarding the need for clinical context for "SoTA" medical MLLMs was misstated, as the smaller variants used in this study are not themselves SoTA models.
> > >
> > > - Our intention in referring to them as "SoTA" was to convey that their **model families** (e.g., the MedGemma family [1] and the Lingshu family [2]) represent the latest publicly available medical MLLMs and are considered as SoTA models in academic research settings.
> > >
> > > - More importantly, the use of these recent cutting-edge medical MLLMs serves a specific purpose: to demonstrate the **general effectiveness of clinical context** across **diverse medical MLLM architectures**. Recent advances in medical MLLMs have enabled integration of multi-modal, multi-source clinical information (e.g., multi-view images, imaging techniques, indications, and prior studies), making them appropriate for benchmarking the effectiveness of clinical context.
> > >
> > > - Nevertheless, we respectfully agree that the term "SoTA MLLMs" may have caused confusion. We have revised the manuscript and now refer to them as **"diverse recent medical MLLM families"**, as reflected in our revised revision.
> > >
> > > ---

---

> > > ### Author Response · Authors · 2025-11-27
> > > **Response to Reviewer TNV1 (3/4)**
> > >
> > > ---
> > >
> > > **[Follow-Up-Q2-2]** The SRRG project page indicates models such as CheXpert-Plus and MAIRA-2 which might be SoTA for the SRRG task. The current models in the experiments do not represent SoTA in my opinion, so the framing between the contribution and the experiments isn't fully aligned as it is in the current form.
> > >
> > > - Instead, we report the performance of CheXpert-Plus [3] and MAIRA-2 [4] on the SRRG without clinical context which is taken from `Table 5 and 6` in [6].
> > >
> > > **[Table TNV1-4]** Results on the **SRRG-Findings**.
> > > |model|context|split|BLEU|ROUGE-L|BERTScore|F1-RadGraph|Prec(SRR)|Rec(SRR)|F1(SRR)|CatPrec|CatRec|CatF1|
> > > |:-|:-:|:-:|:-:|:-:|:-:|:-:|:-:|:-:|:-:|:-:|:-:|:-:|
> > > |CheXagent-3B|✘|Valid|1.97|20.63|30.33|13.07|44.67|45.16|43.46|73.61|81.17|75.54|
> > > ||✘|Test|2.08|20.09|31.91|12.99|43.73|42.54|41.70|74.47|85.26|77.74|
> > > ||✘|Test-reviewed|2.13|20.38|32.73|12.96|44.94|42.78|42.31|72.84|87.35|77.55|
> > > |CheXagent-3B|✔|Valid|2.31|23.01|33.46|15.76|48.73|48.20|46.79|77.58|83.46|78.73|
> > > ||✔|Test|1.89|20.92|33.28|13.58|45.18|44.10|43.07|75.79|85.69|78.82|
> > > ||✔|Test-reviewed|1.98|21.64|34.32|14.05|47.50|45.09|44.59|76.08|88.87|79.93|
> > > |MedGemma-4B|✘|Valid|1.51|20.95|30.83|13.98|42.93|45.50|42.12|78.48|78.00|76.26|
> > > ||✘|Test|1.58|19.69|31.52|13.30|42.32|41.38|40.19|76.31|82.36|77.44|
> > > ||✘|Test-reviewed|1.60|20.11|32.61|13.42|44.49|42.94|41.92|75.39|86.56|78.24|
> > > |MedGemma-4B|✔|Valid|4.98|27.22|37.87|20.44|50.52|49.68|48.42|80.38|83.73|80.35|
> > > ||✔|Test|3.05|23.17|**35.65**|15.91|45.84|44.24|43.43|78.28|84.67|79.59|
> > > ||✔|Test-reviewed|4.29|**24.37**|**36.60**|**17.01**|**47.90**|**45.17**|**44.96**|76.73|87.84|**80.04**|
> > > |Lingshu-7B|✘|Valid|1.42|17.68|27.20|10.56|40.15|41.45|39.29|74.37|75.97|73.57|
> > > ||✘|Test|1.40|17.71|29.65|11.14|40.60|39.41|38.65|75.86|81.47|76.84|
> > > ||✘|Test-reviewed|1.60|18.62|31.09|12.09|42.85|40.82|40.37|74.32|85.20|77.39|
> > > |Lingshu-7B|✔|Valid|6.02|28.70|38.85|21.67|**51.16**|**50.50**|49.20|**81.97**|83.03|**80.87**|
> > > ||✔|Test|3.16|**23.53**|35.60|16.07|**45.96**|**44.42**|43.63|**79.80**|83.20|**79.68**|
> > > ||✔|Test-reviewed|**4.42**|23.70|35.76|16.09|47.48|44.80|44.54|**77.57**|86.71|79.83|
> > > |CheXpert-Plus|✘|Valid|4.12|20.90|31.58|16.95|44.28|43.19|42.08|72.10|85.45|76.52|
> > > ||✘|Test|**3.51**|18.97|31.50|14.99|42.79|40.08|39.85|72.84|86.17|77.18|
> > > ||✘|Test-reviewed|3.96|18.72|31.33|14.89|42.78|39.10|39.28|71.63|88.71|77.24|
> > > |MAIRA-2|✘|Valid|**6.32**|**29.00**|**39.38**|**25.62**|49.66|49.66|**49.66**|78.21|**86.24**|80.52|
> > > ||✘|Test|3.39|23.15|35.44|**19.03**|43.65|43.65|**43.65**|75.64|**86.23**|79.03|
> > > ||✘|Test-reviewed|2.26|20.55|32.87|16.90|42.36|42.36|42.36|72.25|**88.90**|77.79|
> > >
> > > **[Table TNV1-4]** Results on the **SRRG-Impression**.
> > > |model|context|split|BLEU|ROUGE-L|BERTScore|F1-RadGraph|Prec(SRR)|Rec(SRR)|F1(SRR)|
> > > |:-|:-:|:-:|:-:|:-:|:-:|:-:|:-:|:-:|:-:|
> > > |CheXagent-3B|✘|Valid|9.44|34.03|61.82|19.30|63.80|63.48|59.10|
> > > ||✘|Test|7.83|29.40|59.82|16.13|57.18|59.18|54.27|
> > > ||✘|Test-reviewed|7.42|28.60|58.35|13.71|51.32|56.34|49.74|
> > > |CheXagent-3B|✔|Valid|7.52|32.99|60.93|17.90|66.28|61.75|59.04|
> > > ||✔|Test|7.03|29.18|59.66|16.07|59.69|58.42|55.07|
> > > ||✔|Test-reviewed|6.92|29.04|58.91|14.84|55.42|58.26|52.86|
> > > |MedGemma-4B|✘|Valid|8.92|41.24|60.94|17.80|62.19|60.77|56.81|
> > > ||✘|Test|7.15|37.84|59.09|15.35|56.27|57.01|52.69|
> > > ||✘|Test-reviewed|7.57|35.91|58.35|14.57|51.69|54.42|49.51|
> > > |MedGemma-4B|✔|Valid|11.76|**46.26**|64.28|24.25|65.78|**66.77**|62.31|
> > > ||✔|Test|10.58|**41.92**|61.85|19.23|59.45|**61.89**|**57.12**|
> > > ||✔|Test-reviewed|11.21|**40.15**|61.12|19.16|55.02|**60.71**|54.20|
> > > |Lingshu-7B|✘|Valid|8.15|32.17|59.15|17.23|63.82|57.10|55.06|
> > > ||✘|Test|6.65|27.27|57.18|13.87|56.03|51.55|49.33|
> > > ||✘|Test-reviewed|7.04|27.70|57.37|13.49|52.34|52.85|48.37|
> > > |Lingshu-7B|✔|Valid|11.77|38.46|**64.82**|25.29|**69.42**|63.57|**62.48**|
> > > ||✔|Test|10.58|32.86|62.07|19.85|**63.04**|58.39|57.01|
> > > ||✔|Test-reviewed|11.61|33.66|**62.04**|**21.28**|**57.48**|58.80|**54.53**|
> > > |CheXpert-Plus|✘|Valid|**16.86**|33.42|62.74|**27.74**|54.40|51.26|50.26|
> > > ||✘|Test|**14.84**|28.01|60.76|**22.14**|48.74|47.60|46.48|
> > > ||✘|Test-reviewed|**14.07**|26.79|59.21|18.89|43.46|48.15|44.56|
> > > |MAIRA-2|✘|Valid|9.66|31.50|27.82|62.30|20.37|48.72|57.91|50.36|
> > > ||✘|Test|8.12|27.82|**62.30**|20.37|48.72|57.91|50.36|
> > > ||✘|Test-reviewed|5.28|26.61|60.79|19.08|44.80|57.69|47.97|
> > >
> > > - As shown in `Table TNV1-4 and TNV1-5`, MAIRA-2 achieve better performance on **some metrics**, but not all.
> > > - However, MedGemma-4B and Lingshu-7B **often outperform** CheXpert-Plus and MAIRA-2 **when provided with clinical context**, further demonstrating the effectiveness of clinical context.
> > >
> > > ---

---

> > > ### Author Response · Authors · 2025-11-27
> > > **Response to Reviewer TNV1 (4/4)**
> > >
> > > ---
> > >
> > > ### Reference
> > >
> > > [1] Sellergren, Andrew, et al. "Medgemma technical report." arXiv preprint. 2025.
> > >
> > > [2] Xu, Weiwen, et al. "Lingshu: A Generalist Foundation Model for Unified Multimodal Medical Understanding and Reasoning." arXiv preprint. 2025.
> > >
> > > [3] Chambon, Pierre, et al. "Chexpert plus: Augmenting a large chest x-ray dataset with text radiology reports, patient demographics and additional image formats." arXiv preprint. 2024.
> > >
> > > [4] Bannur, Shruthi, et al. "Maira-2: Grounded radiology report generation." arXiv preprint. 2024.
> > >
> > > [5] Delbrouck, Jean-Benoit, et al. "Automated structured radiology report generation." ACL. 2025.
> > >
> > > [6] Chen, Zhihong, et al. "Chexagent: Towards a foundation model for chest x-ray interpretation." arXiv preprint. 2024.
> > >
> > > [7] Liu, Ze, et al. "Swin Transformer V2: Scaling Up Capacity and Resolution." CVPR. 2022.
> > >
> > > [8] Devlin, Jacob, et al. "BERT: Pre-training of Deep Bidirectional Transformers for Language Understanding." NAACL. 2019.
> > >
> > > ---

---

> ### Author Response · Authors · 2025-11-27
> **Response to Reviewer TNV1 (2/4)**
>
> ---
>
> > **[Follow-Up-Q2-1]** The SRRG project page indicates models such as CheXpert-Plus and MAIRA-2 which might be SoTA for the SRRG task. The current models in the experiments do not represent SoTA in my opinion, so the framing between the contribution and the experiments isn't fully aligned as it is in the current form.
>
> - We sincerely appreciate the reviewer for raising this point and fully agree that CheXpert-Plus [3] and MAIRA-2 [4] are SoTA methods for the SRRG task.
> - First, we would like to clarify that:
>     - Our intention in the experiments is **not to propose new SoTA methods for the SRRG task** or to **compare them against existing SoTA systems** (e.g., CheXpert-Plus and MAIRA-2), but rather to **investigate and demonstrate the effectiveness of clinical context** across **diverse medical MLLMs** on the SRRG task.
>     - Therefore, we respectfully believe that the **current experimental setup fully supports** our main claim and contribution.
>
>
> - Given the above intention, we would like to provide detailed explanations for **why CheXpert-Plus and MAIRA-2 were not included** in our experiments.
>
>     **CheXpert-Plus:**
>
>     - CheXpert-Plus is basically **not a multimodal LLM** but a **SwinV2-based encoder with a 2-layer BERT decoder**, which fundamentally differs from the MLLM architectures we evaluate.
>     - Additionally, the maximum position embedding of BERT is **only 512 tokens** [8], as confirmed in the [public configuration](https://huggingface.co/StanfordAIMI/chexpert-plus-srrg_findings/blob/main/config.json) of model checkpoint used in prior work [5].
>     - With a 256×256 image and encoder stride of 32, the visual representation alone consumes **64 tokens** (8×8 patches), leaving only **448 tokens** for any textual input and output combined, which is far **too limited to accommodate clinical context**, such as multiple images, clinical indications, imaging techniques, and prior study narratives simultaneously.
>
>     **MAIRA-2:**
>
>     - MAIRA-2 is indeed a **multimodal LLM**; however, it has **critical architectural limitations** that prevent its evaluation under our C-SRRG framework:
>
>     - **Fixed input scheme**: As noted in `L157-161` and `Appendix B`, MAIRA-2 is already trained with a **predefined input format** for **free-form report generation**. The model expects specific input modalities in a fixed configuration, which cannot be adapted to accommodate **variable-length clinical history** and **variable-length multi-view images** required by our C-SRRG framework.
>
>     - **Context length limitations**: Similar to the issues we observed with CheXagent-3B [6], MAIRA-2 suffers from **insufficient context capacity**. The base language model is Vicuna 7B v1.5, which has a default context length of **4,096 tokens**. With the 1.5× RoPE scaling factor, the effective context length is extended to approximately **6,144 tokens**.
>
>     - **High-resolution visual tokens**: Each **518×518** image is processed into patches of size **14×14** and encoded by Rad-DINO-MAIRA-2 into a sequence of **1,369 visual tokens**. With up to 3 images (current frontal, current lateral, and prior frontal), this amounts to **3 × 1,369 = 4,107 visual tokens alone**. **Adding text tokens** from the prior report, indication, technique, and comparison sections, plus the generated output, the total sequence length **quickly exceeds the effective context capacity**.
>
>
> - In contrast to CheXpert-Plus and MAIRA-2, we clarify **the rationale behind selecting CheXagent-3B [6], MedGemma-4B [1], and Lingshu-7B [2]**:
>
>     - We include CheXagent-3B because it was evaluated in the original SRRG benchmark [5]. Although it has similar limitations (e.g., context-length constraints), it can be trained with clinical context.
>     - Both MedGemma [1] and Lingshu [2] are **recently proposed general-purpose medical foundation MLLMs** explicitly designed to support **extended context lengths** and **flexible multi-modal inputs**.
>
> ---
>
> - In **summary**:
>
>     - The **current experimental setup fully supports** our main claim and contribution, which is to demonstrate the effectiveness of clinical context using recent medical MLLMs that are feasible to incorporate into C-SRRG (i.e., without architectural or context-length limitations).
>
>     - While the suggested methods, such as CheXpert-Plus and MAIRA-2, are indeed **SoTA for the SRRG task**, they **cannot be incorporated into or evaluated within the C-SRRG framework** due to their inherent limitations.
>
> ---

---

### Official Review · Reviewer_SiHb · 2025-11-04

**Soundness:** 2
**Presentation:** 3
**Contribution:** 3
**Rating:** 6
**Confidence:** 4

**Summary:**

This paper presents C-SRRG, a contextualized framework for structured radiology report generation that explicitly incorporates clinically relevant contextual signals including multi-view chest X-rays, imaging technique, indication, and prior studies. The authors curate a well-aligned dataset combining MIMIC-CXR and CheXpertPlus, and evaluate the proposed framework using advanced medical multimodal large language models (MLLMs), including CheXagent-3B, MedGemma-4B, and Lingshu-7B.

**Strengths:**

* Clinical relevance: The paper is well-motivated from a clinical standpoint. The integration of contextual cues mirrors the diagnostic reasoning processes of radiologists and addresses a gap in current MLLM-based report generation.
* Dataset contribution: The curated dataset, aligned with structured reporting templates, is valuable for the community. The construction pipeline is described clearly and considers temporal alignment and semantic normalization.
* Empirical thoroughness: The experiments are comprehensive and include comparisons across multiple MLLMs, detailed ablations, and organ-level breakdowns. Results show consistent improvements in structured report generation (+2–7% F1-SRR-BERT) and a notable reduction in temporal hallucinations (12–18%), highlighting the utility of contextual signals.
* Transparency and reproducibility: The authors provide clear methodology, ethics considerations, and reproducibility statements, which enhance the credibility of the work.

**Weaknesses:**

* Synthetic supervision: A significant portion of the structured labels is generated using LLMs, which may introduce hallucinations or inaccuracies into the supervision signal. The paper would benefit from a more thorough discussion on how these potential inaccuracies affect downstream training and evaluation.
* Model limitations: The evaluation is limited to models ≤7B parameters. While understandable for resource reasons, it raises questions about scalability and generalization to larger foundation models.
* Limited architectural novelty: The framework primarily relies on contextual data feeding and fine-tuning, rather than proposing new architectural mechanisms for structured generation or hallucination mitigation. This may limit its novelty in some readers’ view.

**Questions:**

N/A

**Details Of Ethics Concerns:**

The authors mention that structured clinical context elements are derived from free-form radiology reports using GPT-4. GPT-4 is not specifically tuned for clinical concept extraction, and may introduce inaccuracies or hallucinated outputs, especially when interpreting nuanced medical language. There is no discussion of error handling, validation, or expert review of the parsed components.

---

> ### Author Response · Authors · 2025-11-23
> **Response to Reviewer SiHb (1/3)**
>
> We sincerly appreciate you for your time and constructive comments which improve our paper. We respectfully address your concerns as following:
>
> ---
>
> >**[Q1]** Synthetic supervision: A significant portion of the structured labels is generated using LLMs, which may introduce hallucinations or inaccuracies into the supervision signal. The paper would benefit from a more thorough discussion on how these potential inaccuracies affect downstream training and evaluation.
>
> - We sincerely appreciate the reviewer for raising this important concern about the reliability of our synthetic structured labels. We fully agree that the significant portion of the structured labels is generated using LLMs, which may introduce hallucinations or inaccuracies as we discussed in `Appendix A.1`.
>
> - However, we repectfully note that the structured report generation task was introduced and **validated by prior work** through a **rigorous reader study** with **five board-certified radiologists** [1].
> - They assessed the clinical validity of GPT-4 generated structured reports on the `test_reviewed` split (`Appendix B` in [1]), demonstrating that this approach maintains **sufficient clinical fidelity** for practical use as follows:
>
>     - First of all, **five board-certified radiologists reviewed 464 reports** sampled from the MIMIC-CXR test set and CheXpert Plus validation set.
>
>     - Their analysis shows the **high average similarity ratios** which is computed as 2 × matches / total tokens in original and edited texts; 0.77 for impressions and 0.88 for findings. It indicates that modifications were **generally minor**, focusing on enhancing clarity and precision rather than altering fundamental diagnostic content.
>
>     - Furthermore, the utterance-level label validation achieved a **72% exact match rate** and **74% average Jaccard similarity** [2] between GPT-generated consensus labels and radiologist-reviewed labels, confirming the reliability of the automated label extraction approach.
>
>     - Lastly, **large-scale data may average out such errors** across diverse examples, and the clinical validation from prior work [1] **suggests the error is manageable**.
>
> - Additionally, our evaluation protocol is designed to assess **clinical validity rather than exact textual matching**:
>
>     - **F1-SRR-BERT captures semantic similarity** appropriate for medical reporting where multiple valid phrasings exist.
>
>     - Organ-level breakdowns provide **fine-grained anatomical assessment**.
>
>     - Temporal hallucination analysis was conducted by **detecting temporal keywords solely from model generated structured report**, not from synthetically generated ground truth structured report.
>
> - In summary, the synthetic supervision used in structured report generation is thoroughly **validated by certified radiologists** [1], and our evaluation protocol naturally **avoids potential bias** when comparing generated outputs with the synthetic targets.
>
> - Nevertheless, as discussed in `Appendix A.1`, we acknowledge that our approach **lacks regulatory clearance** for clinical deployment, and **practitioners must retain final authority** over all diagnostic decisions. Given the inherent risk of errors or hallucinations in any AI system, we strongly advocate for **rigorous clinical validation and expert oversight** when exploring or extending this work.
>
> - We have added a dedicated discussion at `L1052-1072` of `Appendix A.1` in the revised manuscript.
>
> ---

---

> ### Author Response · Authors · 2025-11-23
> **Response to Reviewer SiHb (2/3)**
>
> ---
>
> >**[Q2]** Model limitations: The evaluation is limited to models ≤7B parameters. While understandable for resource reasons, it raises questions about scalability and generalization to larger foundation models.
>
> - We sincerely agree that exploring the effectiveness of C-SRRG across a broader range of model scales, and examining how contextual signals scale with model capacity, would significantly strengthen the contribution and provide valuable insights.
> - However, we kindly note that we **have already acknowledged this limitation** in `L478–479`, and we would also like to appreciate the opportunity to **further discuss its implications and context**:
>
>     - As mentioned in the question, our evaluation is constrained by computational resources and model availability. As noted in `L338-339`, all experiments were conducted on a **single H100 GPU**, which limits the scale of models we could evaluate.
>
>     - More crucially, publicly available medical MLLMs offer only limited size options: MedGemma [3] provides 4B and **27B** variants, while Lingshu [4] provides 7B and **32B** versions. We respectfully note that these larger models (e.g., MedGemma-27B or Lingshu-32B) are not only beyond our computational capacity but also **exceed the model sizes commonly studied in academic fine-tuning and evaluation settings**.
>
>     - Lastly, we acknowledge that our **current evaluation does not demonstrate how our findings generalize to larger MLLMs**.
>
> - We sincerely appreciate your understanding of the above practical limitations.
>
> - Accordingly, we have revised the manuscript to more carefully qualify our claims about the broader applicability of our approach and explicitly discuss this limitation in a dedicated section at `L1082-1093` of `Appendix A.2`.
>
> ---
>
> >**[Q3]** Limited architectural novelty: The framework primarily relies on contextual data feeding and fine-tuning, rather than proposing new architectural mechanisms for structured generation or hallucination mitigation. This may limit its novelty in some readers’ view.
>
> - We sincerely appreciate the reviewer for this valuable feedback regarding the architectural novelty. We first clarify that we **do not claim novelty** in introducing **fundamentally new architectural mechanisms**, and we fully agree that this is a fair assessment.
>
> - However, we respectfully believe that as noted in `L106-117`, our primary contribution lies in **identifying a key limitation of SRRG**, **curating a dataset with essential clinical context**, and **providing empirical evidence** through benchmarking to facilitate future work on SRRG for practitioners:
>
>     - We systematically analyze **the clinical workflow of radiologists** to identify relevant contextual signals that are routinely consulted during report writing, and demonstrates empirically that omitting such context leads to **systematic issues like temporal hallucinations** as depicted in `L187-198` in `§3.1`.
>
>     - A key distinction from prior work is the **comprehensive incorporation of multiple clinical contexts simultaneously**, multi-view chest radiographs, imaging technique specifications, clinical indications, and comparison with prior studies, rather than limiting analysis to isolated contextual elements as depicted in `L200-215` in `§3.1`.
>
>     - To the best of our knowledge, this study is the **first to curate a large-scale structured radiology report generation dataset** that aligns these diverse clinical contexts with structured radiology reports, enabling systematic investigation of their effects as in `Tabs. 5, 6, 7, and 8`.
>
>     - Furthermore, we provide the **first comprehensive empirical study** examining how these clinical contexts influence structured report generation across **multiple state-of-the-art medical MLLMs (e.g., CheXagent-3B, MedGemma-4B, Lingshu-7B)**, revealing consistent improvements (**+2–7% F1-SRR-BERT**) and substantial reductions in temporal hallucinations (**12–18%**).
>
> - Notably, investigating the **effect of clinical contexts in medical MLLMs remains largely underexplored**, despite **the significant importance of report writing in the clinical workflow** described in `L43-47` (i.e. report written by radiologists is read by clinicians to do desicion-making on the patient).
> - This is partly due to the limited capability MLLMs in handling long context, as evidenced by the difficulties of CheXagent-3B with comprehensive contextual information (`L331-333`, `Appendix A.2, A.3`, and `Appendix C`).
>
> - We believe that the above discussion clarifies that we do **not** claim any architectural novelty, and it also clarifies the contributions presented in `L104-115`.
>
> ---

---

> ### Author Response · Authors · 2025-11-23
> **Response to Reviewer SiHb (3/3)**
>
> ---
>
> >**[Q4]** The authors mention that structured clinical context elements are derived from free-form radiology reports using GPT-4. GPT-4 is not specifically tuned for clinical concept extraction, and may introduce inaccuracies or hallucinated outputs, especially when interpreting nuanced medical language. There is no discussion of error handling, validation, or expert review of the parsed components.
>
> - We sincerely appreciate the reviewer for highlighting this important concern regarding GPT-4–based extraction. We respectfully believe that our response to **[Q1]** has addressed the main issue. Here, we would like to summarize our response to **[Q1]**:
>
>     - We first kindly note that the structured report derived from free-form radiology reports are **clinically validated** by prior work [1] through a reader study with **five board-certified radiologists**.
>     - They assessed the clinical validity of GPT-4 generated structured reports on the `test_reviewed` split (`Appendix B` in [1]), demonstrating **sufficient fidelity for research use** despite GPT-4 not being specifically tuned for clinical extraction.
>     - We adopt this validated framework for SRRG and **built upon it to incorporate relevant clinical contexts** (indication, imaging technique, prior studies) into the generation process as in `L248-258`.
>
> - We believe this discussion on error handling, validation, and quality assurance measures for the parsed components, together with our response in **[Q1]** regarding synthetic supervision, comprehensively addresses your concerns about the reliability of our dataset construction process.
>
> ---
>
> ### Reference
> [1] Delbrouck, Jean-Benoit, et al. "Automated structured radiology report generation." ACL 2025.
>
> [2] Jaccard, Paul. "Étude comparative de la distribution florale dans une portion des Alpes et des Jura." Bull Soc Vaudoise Sci Nat 37 (1901): 547-579.
>
> [3] Sellergren, Andrew, et al. "Medgemma technical report." arXiv 2025.
>
> [4] Xu, Weiwen, et al. "Lingshu: A Generalist Foundation Model for Unified Multimodal Medical Understanding and Reasoning." arXiv 2025.
>
> ---

---

> ### Author Response · Authors · 2025-11-27
> **Gentle Reminder**
>
> Dear Reviewer SiHb,
>
> We sincerely appreciate your time and consideration. We respectfully believe that our response has thoroughly addressed the concerns raised. If you have any remaining concerns or questions, please feel free to contact us and we would be happy to discuss and clarify them.
>
> Best,
>
> The Authors

---

### Author Response · Authors · 2025-11-23
**General Response**

We sincerely thank all reviewers for their time and constructive feedback, which have significantly improved our paper.

We are grateful that the reviewers recognize several strengths of our work, including its **strong clinical motivation and relevance** (SiHb, TNV1, 3C92), the **well-curated contextual SRRG dataset** (SiHb, TNV1), and our **comprehensive empirical evaluation with detailed ablations** (SiHb, TNV1, 4zws).

We believe that we have successfully addressed all major concerns in this rebuttal. Below, we summarize **the changes** that have been included in the revised version (shown in blue):

---

- `L161` of `§2`: in response to reviewer **YbDT**, we explain why prior works with **specialized architectures cannot be evaluated on our proposed benchmark**, and we provide comparisons with recent medical MLLMs (with further details in `Appendix B`).

- `L200-201` in `§3.1`: In response to reviewer **3C92**, we discussed how **context omission affects clinical interpretability**, explaining the distinct **role of each context component**.

- `L376-377` in `§4.2`: In response to reviewer **TNV1**, we **toned down scaling claim**, acknowledging confounding factors when comparing different model families (MedGemma vs Lingshu).

- `L451-454` in `§4.2`: In response to reviewer **TNV1**, we added **temporal hallucination analysis for all three models**, demonstrating widespread rates.

- `L1052–1075` in `Appendix A.1`: In response to reviewers **SiHb**, **3C92**, and **YbDT**, we discuss the **risks of GPT-4–based synthetic supervision** and provide additional details on the **clinical validation conducted by five radiologists**.

- `L1082-1093` in `Appendix A.2`: In response to reviewers **SiHb** and **TNV1**, we discussed **the limitations of model scale** (smaller than 7B parameters), **computational constraints** (single H100 GPU), and **public medical MLLM options**.

- `L1119–1135` and `L1144–1158` in `Appendix A.3`: In response to reviewers **3C92** and **YbDT**, we discuss the limitations regarding the **range of clinical context**, clarifying **our choice** to include only the two most recent prior studies and to focus on imaging-centric contextual information.

- `Appendix A.5`: In response to reviewer **3C92**, we discuss the limitations of our **radiologist-in-the-loop evaluation**, noting the practical resource constraints and emphasizing that dataset validation was conducted through expert assessment.

- `Appendix A.6`: In response to reviewer **3C92**, we discussed **generalization to other imaging modalities** (CT, MRI, ultrasound, mammography) while acknowledging need for empirical validation.

- `Appendix B`: In response to reviewer **YbDT**, we expanded the discussion in `L161` of `§2`.

- `Appendix C`: In response to reviewer **TNV1**, we added the analysis on the **temporal hallucination behavior** and the **training failure** of **CheXagent-3B**.

- `Appendix D`: In response to reviewer **3C92**, we added **clinical interpretation of each context component** (multi-view images, indication, technique, prior studies) explaining how omission affects diagnostic accuracy and clinical utility.

- `Appendix E`: In response to reviewer **TNV1**, we include results of medical MLLMs **without fine-tuning**, along with a **hallucination analysis** examining temporal statements in both the dataset ground truth and model outputs.


---

---

### Author Response · Authors · 2025-11-29
**Letter to AC and Reviewers (2/2)**

---

### Reviewer 3C92


> **[Q1 & Q3 & Q4]** The reviewer raised a concern about **synthetic supervision**.

- We clarify that the structured report generation task is validated by prior work [1] with **five board-certified radiologists**, and the evaluation protocol naturally **avoids potential bias** when comparing generated outputs with the synthetic targets.

> **[Q2]** The reviewer raised a concern about **truncated longitudinal contexts**.

- We clarify that our design choice is based on both clinical practice [4] (radiologists consult an average of **3.2 prior imaging studies per report**) and technical constraints (**the context limits of medical LLMs**).


> **[Q5]** The reviewer questioned the absence of a **radiologist-in-the-loop** evaluation.

- We discuss that conducting radiologist-in-the-loop evaluation presents **significant resource challenges**, and we clarify that this remains an important direction for future work.


> **[Q6]** The reviewer questioned the **generalization to other imaging modalities**.

- We clarify that this is beyond the scope of our current study. However, we provide the hypothesis that contextual information would play a **crucial role in report generation across various imaging modalities** (e.g., CT, MRI, ultrasound, and mammography).


> **[Q7]** The reviewer questioned the **impact of context omission from a clinical perspective**.

- We clarify that omitting multi-view images, indication, imaging technique, and prior studies not only reduces model performance quantitatively (`Tabs. 7 and 8`), but also **fundamentally limits clinical utility and safety**.


---

### Reviewer YbDT

> **[Q1]** The reviewer raised a concern about the **novelty of curating the C-SRRG dataset** on top of prior work [1].

- We clarify that while the C-SRRG dataset is built upon prior work [1], our contribution goes far beyond simple data processing. It addresses a **fundamental limitation in the current structured radiology report generation framework** by incorporating comprehensive and clinically essential contextual information.


> **[Q2]** The reviewer raised a concern about the **novelty of incorporating clinical context in RRG**.

- We clarify that previous work has considered only 1) **partial contexts**, 2) **free-form generation**, or 3) models that are **not medical MLLMs**.
- Our study is the first to investigate the effectiveness of **extensive clinical context** on **structured** radiology report generation using **medical MLLMs**.
- We also clarify that previous work cannot be evaluated on our proposed benchmark due to their **specialized architectural components**, which motivates us to investigate the effectiveness of clinical context **on medical MLLMs**.

> **[Q3]** The reviewer raised a concern about **unexplored clinical context**.

- We clarify that we follow **best practice guidelines** [5, 6], which establish indication, technique, and comparison as standard components of high-quality radiology reports.
- However, we acknowledge that additional clinical context may be helpful and consider this an **important direction for future work**.

> **[Q4 & Q5 & Q6 & Q7]** The reviewer raised concerns about **synthetic supervision** and **prompts**.

- We clarify that the structured report generation task is validated by prior work [1] with **five board-certified radiologists**, and the evaluation protocol naturally **avoids potential bias** when comparing generated outputs with the synthetic targets.
- We also clarify the prompts used for generating the synthetic targets and provide a pointer to the prompt details in [1].

---

### Reference

[1] Delbrouck, Jean-Benoit, et al. "Automated structured radiology report generation." ACL 2025.

[2] Chambon, Pierre, et al. "Chexpert plus: Augmenting a large chest x-ray dataset with text radiology reports, patient demographics and additional image formats." arXiv preprint. 2024.

[3] Bannur, Shruthi, et al. "Maira-2: Grounded radiology report generation." arXiv preprint. 2024.

[4] Haygood, Tamara Miner, et al. "Consultation and citation rates for prior imaging studies and documents in radiology." Journal of Medical Imaging. 2018.

[5] Kahn Jr, Charles E., et al. "Toward best practices in radiology reporting." Radiology. 2009.

[6] European Society of Radiology (ESR). "Good practice for radiological reporting. Guidelines from the European Society of Radiology (ESR)." Insights into imaging. 2011.

---

---

### Author Response · Authors · 2025-11-29
**Letter to AC and Reviewers (1/2)**

Dear AC and Reviewers,

We sincerely appreciate your time and efforts in reviewing our paper. Due to the unfortunate incident during the rebuttal period, ICLR decided to halt the discussion between reviewers and authors and to reassign the corresponding AC. Accordingly, we would like to provide a summary of our rebuttal for the new AC.

We first summarize **several strengths** recognized by reviewers:
- **strong clinical motivation and relevance** (SiHb, TNV1, 3C92),
- the **well-curated contextual SRRG dataset** (SiHb, TNV1), and
- **comprehensive empirical evaluation with detailed ablations** (SiHb, TNV1, 4zws).

We next **summarize below how we addressed each reviewer’s concerns**, which we believe have been successfully resolved. Furthermore, if the new AC has any remaining concerns or questions raised by the reviewers, we would be happy to address them.

---

### Reviewer SiHb

>**[Q1 & Q4]** The reviewer raised a concern about **synthetic supervision**.

- We clarify that the SRRG task is validated by prior work [1] with **five board-certified radiologists**, and the evaluation protocol naturally **avoids potential bias** when comparing generated outputs with the synthetic targets.

---

>**[Q2]** The reviewer raised a concern about **scale of the models**.

- We clarify that due to **computational constraints** and the fact that publicly available medical MLLMs **offer only limited size options** (e.g., 4B vs. 27B or 7B vs. 32B), our experimental choices were necessarily restricted.


>**[Q3]** The reviewer raised a concern about **limited architectural novelty**.

- We clarify that we **do not claim architectural novelty**, and our contributions lie in **identifying a key limitation of SRRG**, **curating a dataset with essential clinical context**, and **providing comprehensive empirical evidence**.

---

### Reviewer TNV1

> **[Rating]** The reviewer raised the rating from **2 to 4** with two follow-up questions.

>**[Q1]** The reviewer raised a concern about **limited technical novelty**.

- We clarify that our contributions lie in **identifying a key limitation of SRRG**, **curating a dataset with essential clinical context**, and **providing comprehensive empirical evidence**.

>**[Q2]** The reviewer raised a concern about a **misclaim regarding scaling behavior with clinical context**.

- We agree with this concern and have toned down the claim to: "We observe that performance increases as the number of parameters grows from 3B to 7B; however, this does not imply that clinical context is always helpful, as there may be confounding factors."

>**[Q3]** The reviewer raised a concern about the **limited investigation of temporal hallucination across diverse models**.

- We conducted an additional analysis on temporal hallucination using CheXagent-3B and Lingshu-7B:
    - For CheXagent-3B, temporal hallucination increases with clinical context due to the presence of a few key words (“stable” and “unchanged”), but decreases when these words are excluded.
    - For Lingshu-7B, temporal hallucination consistently decreases with clinical context.

>**[Q4]** The reviewer questioned that the C-SRRG dataset is curated on top of prior work [1].

- We clarify that while the C-SRRG dataset is built upon [1], our contribution goes far beyond simple data processing. It addresses a **fundamental limitation in the current SRRG framework** by incorporating comprehensive and clinically essential context.

>**[Q5]** The reviewer questioned whether the training failure behavior of CheXagent-3B exists prior to training.

- We clarify what we mean by training failure behavior: the model **fails to follow the expected output structure** required for SRRG tasks. This behavior indeed exists by definition, as the model was **never trained on these tasks**, and therefore naturally does not produce outputs aligned with the required structure.

>**[Q6]** The reviewer questioned the baseline performance without any fine-tuning.

- We clarify that evaluating the **baseline without any fine-tuning on SRRG-Findings is not meaningful**.
- We report the performance without any fine-tuning on **SRRG-Impression**, which shows a significant degradation, highlighting the importance of fine-tuning.

>**[Follow-Up-Q1]** The reviewer raised a concern about a misclaim regarding **referring to smaller MLLMs as SoTA**.

- We clarify that the use of SoTA refers to the **state-of-the-art within each model family**.

>**[Follow-Up-Q2]** The reviewer raised a concern about the **comparison to CheXpert-Plus [2] and MAIRA-2 [3]**.

- We clarify that we exclude CheXpert-Plus [2] and MAIRA-2 [3] from the benchmarking because their **architectural limitations** prevent us from incorporating clinical context.
- Instead, we report their performance **without clinical context**, and observe that MedGemma-4B and Lingshu-7B **outperform both when clinical context is included**, demonstrating the effectiveness of clinical context in SRRG tasks.

---

---

### Meta-Review · Area_Chair_ucYG · 2026-01-11

**Summary:**

Three out of the four reviewers believe that the manuscript does not meet the standards of ICLR or is slightly below them. Therefore, the manuscript will be rejected.

**Reviewer Concerns:**

Concerns addressed by the rebuttal:

- Synthetic labels / GPT-4 supervision: Authors clarified that the structured report format and labels come from a prior ACL 2025 study clinically validated by five board-certified radiologists (high similarity ratios, 72% exact match). Evaluation uses semantic metrics (F1-SRR-BERT), not verbatim matching.
- Limited model scale (≤7B): Due to hardware limits (single H100) and lack of publicly available larger medical MLLMs suitable for fine-tuning. Claims were appropriately qualified.
- Temporal hallucination analysis: Initially only on MedGemma-4B; authors added new results for CheXagent-3B and Lingshu-7B, showing model-dependent behaviors and root causes (e.g., overuse of “stable/unchanged” in CheXagent).
- Baseline without fine-tuning: Authors showed unfine-tuned models cannot generate structured “Findings,” but can produce “Impression”; clinical context still helps even without fine-tuning.
- Exclusion of SOTA baselines (CheXpert-Plus, MAIRA-2): These models have architectural or context-length limitations that prevent integration of rich clinical context. Authors reported their zero-context scores and showed MedGemma/Lingshu with context often outperform them.
- Scope of clinical context: Authors justified focusing on imaging-centric contexts (indication, technique, prior studies, multi-view) per radiology best practices; acknowledged broader EHR/lab data as future work.


Concerns partially addressed / still outstanding:

- Novelty: All reviewers noted limited technical or architectural novelty. Authors clarified they do not claim algorithmic novelty—contribution is dataset + benchmark + empirical analysis. This framing may still be seen as incremental.
- Dataset contribution: Reviewer 4zws argued C-SRRG is mostly reformatting existing data. Authors countered that it’s the first unified benchmark enabling systematic study of rich context in structured report generation with medical MLLMs—a perspective that remains subjective.

**Reviewer Scores:**

Reviewer SiHb Likely would maintain or slightly improve to a 4 (Weak Accept) after rebuttal, as major concerns (label quality, model scale) were well addressed. However, may still have reservations about novelty.

Reviewer TNV1: Likely would increase to a 3 or 4, given the thorough responses, new experiments (e.g., temporal hallucination on more models), and clarifications on baselines and fine-tuning. Original score appeared based on misunderstandings now resolved.

Reviewer 3C92: Already positive; likely would stay at 5 or move to 6 (Accept) after seeing additional validation and acknowledgment of limitations (e.g., radiologist evaluation as future work).

Reviewer 4zws: Main objection was perceived lack of novelty/data contribution. Rebuttal clarified the benchmarking focus, but this is subjective. Might remain at 3 or shift to 4, depending on how they weigh empirical rigor vs. novelty.

---

### Decision · Program_Chairs · 2026-01-26

Reject